# Different responses of cold-air outbreak clouds to aerosol and ice production depending on cloud temperature

Xinyi Huang[1], Paul R. Field[1,2], Benjamin J. Murray[1], Daniel P. Grosvenor[1,2], Floortje van den Heuvel[3], and Kenneth S. Carslaw[1]

[1]Institute for Climate and Atmospheric Science, School of Earth and Environment, University of Leeds, Leeds LS2 9JT, United Kingdom
[2]Met Office, Exeter EX1 3PB, United Kingdom
[3]British Antarctic Survey, Cambridge CE3 0ET, United Kingdom

**Correspondence:** Xinyi Huang (ee21xh@leeds.ac.uk)

**Abstract.** Aerosol-cloud interactions and ice production processes are important factors that influence mixed-phase cold-air outbreak (CAO) clouds and their contribution to cloud-phase feedback. Recent case studies of CAO events suggest that increases in ice-nucleating particle (INP) concentrations cause a reduction in cloud total water content and albedo at the top of the atmosphere. However, no study has compared the sensitivities of the CAO cloud to these processes under different environmental conditions. Here, we use a high-resolution nested model to quantify and compare the responses of cloud microphysics and dynamics in cloud droplet number concentration ($N_d$), INP concentration and efficiency of the Hallet-Mossop (HM) secondary ice production process in two CAO events over the Labrador Sea, representing intense (cold, March) and weaker (warmer, October) mixed-phase conditions. Our results show that variations in INP concentrations strongly influence both cases, while changing $N_d$ and the HM process efficiency affect only the warmer October case. With a higher INP concentration, cloud cover and albedo at the top of the atmosphere increase in the cold March case, while the opposite responses were found in the warm October case. We suggest that the CAO cloud response to the parameters is different in ice-dominated and liquid-dominated regimes, and the determination of the regime is strongly controlled by the cloud temperature and the characteristics of ambient INP, which both control the glaciation of clouds. This study provides an instructive perspective to understand how these cloud microphysics affect CAO clouds under different environmental conditions and serves as an important basis for future exploration of cloud microphysics parameter space.

## 1  Introduction

During cold-air outbreak (CAO) events, cold and dry air masses are drawn from high-latitude continental or sea-ice-covered regions to the warm and open ocean, leading to extensive formation of boundary layer clouds (Brümmer, 1996; Brümmer, 1999; Renfrew and Moore, 1999; Kolstad and Bracegirdle, 2008; Kolstad et al., 2009; Fletcher et al., 2016a, b). CAO clouds, which form mainly over extra-tropical regions and are generally in a mixed-phase state, play an important role in cloud feedback under a warming climate (Ceppi et al., 2017; Sherwood et al., 2020; Zelinka et al., 2020; Murray et al., 2021) and different

physical representations of clouds are a key reason why models in CMIP6 (Coupled Model Intercomparison Project phase 6) have a higher climate sensitivity compared to models in CMIP5 (Zelinka et al., 2020).

Poor representation of mixed-phase CAO clouds is one of the major reasons for radiative flux biases in global climate models (GCMs) compared to observations, especially in the Southern Ocean (Bodas-Salcedo et al., 2014, 2016). As CAO clouds are often in a mixed-phase state, where both cloud liquid and ice are present at the same time, cloud liquid can be rapidly removed by ice through the Wegener-Bergeron-Findeisen (WBF) process (Wegener, 1911; Bergeron, 1935; Findeisen, 1938; Findeisen and Findeisen, 1943) and accretion processes. Therefore, the interactions between liquid and ice hydrometeors as well as their properties are important for mixed-phase clouds, which are strongly controlled by cloud microphysics processes. However, large uncertainties still exist when simulating the behaviour of these mixed-phase clouds under a warming climate because of the poorly represented cloud microphysics in GCMs (Bodas-Salcedo et al., 2019; Sherwood et al., 2020). Recent studies show that using satellite observations of mixed-phase clouds to constrain GCMs result in a higher climate sensitivity (Tan et al., 2016; Hofer et al., 2024), suggesting the importance of having a good representation of mixed-phase clouds in GCMs for future climate prediction. Even within cloud-resolving models, cloud microphysical processes have large uncertainties due to their complicated and highly parameterised nature (Morrison et al., 2020). Aerosol-cloud interactions and ice production processes are the main sources of these uncertainties (Khain et al., 2015; Morrison et al., 2020), as demonstrated in simulations of cloud properties and cloud field development using high-resolution models (Field et al., 2014; Abel et al., 2017; Vergara-Temprado et al., 2018; de Roode et al., 2019; Tornow et al., 2021; Karalis et al., 2022).

Adjustment of various microphysical processes in models has been shown to improve agreement with observations for CAO clouds. Field et al. (2014) found that an improvement of LWP (liquid water path) and radiation bias can be achieved by modifying the boundary layer parameterization and by inhibiting heterogeneous ice formation in CAO clouds. It is also found by Vergara-Temprado et al. (2018) that changes in the INP (ice-nucleating particle) concentration can strongly modulate the freezing behaviour of cloud droplets and the albedo of CAO clouds through changing the liquid-ice partitioning in mixed-phase CAO clouds, and hence affects the cloud-phase feedback (Storelvmo et al., 2015; Murray et al., 2021). Stratocumulus-to-cumulus transition (SCT) in CAOs have an important influence on the radiative properties of CAO clouds, and recent studies have shown that SCT in CAO events is sensitive to aerosol loadings including CCN (cloud condensation nuclei) (de Roode et al., 2019; Tornow et al., 2021), INP concentrations (Tornow et al., 2021) and secondary ice production (SIP) (Karalis et al., 2022), which influence precipitation (Abel et al., 2017) and hence affect the radiative properties of the CAO clouds. These studies highlight the importance of cloud microphysical processes for the modelling of mixed-phase CAO clouds. However, they were mainly focused on the sensitivity of single CAO cases to these uncertain cloud microphysical properties and processes, with limited work on understanding the role of environmental conditions.

Our study aims to improve our understanding of the responses of mixed-phase CAO clouds to CCNs (through changing the droplet number concentration), INPs and the secondary ice production process. We use a convection-permitting numerical weather prediction model with a horizontal grid spacing of 1.5 km over a 1500 km domain and compare two CAO cases over the Labrador Sea that occurred under different environmental conditions – one in spring that was cold and one in autumn that was comparatively warm, with the one in Autumn corresponding to the period of the M-Phase field campaign (Murray and the

MPhase Team, 2024; Tarn et al., 2025). The selected cases also have different marine CAO strengths, which have been found to affect the CAO cloud properties and cloud field morphology by previous studies using satellite and reanalysis data (Fletcher et al., 2016b; McCoy et al., 2017; Wu and Ovchinnikov, 2022; Murray-Watson et al., 2023).

This paper is structured as follows. In Section 2, we describe the two CAO cases, the default model setup, the selection of model parameters including their values for each sensitivity test, and the satellite data used for model-observation comparison. In Section 3, we present the results showing how these parameters affect the CAO cloud properties differently in each case, as well as the comparison between model output and satellite retrievals. In Section 4, we discuss the reasons behind the responses of two CAO events to these tested cloud microphysical processes, along with the limitations and future work.

## 2    Methods

The overall approach of this study is using high-resolution, convection-permitting regional model simulations to understand and compare the sensitivity of mixed-phase CAO cloud properties in two CAO cloud events over the Labrador Sea to droplet number concentration, INP concentration and efficiency of the Hallet-Mossop secondary ice production process.

### 2.1    Case description

Two CAO events were selected over the Labrador Sea: 15 March 2022 and 24 October 2022, with the latter one coinciding with the M-Phase aircraft campaign (Murray and the MPhase Team, 2024; Tarn et al., 2025). Figure 1 shows the UK Met Office surface analysis charts for both cases. There were strong north-westerly flows over the Labrador Sea region during both cases, which is a typical feature during CAO events in this region. A low-pressure system was located to the south-east of Greenland in March, drawing the CAO system around the Greenland. Compared with the March case, the October CAO event 75 was at an earlier stage, generally weaker and only for approximately 2 days (compared to approximately 4 days of the March CAO event). The October case was also accompanied with warmer environmental conditions (see Section 2.6 for cloud top temperatures from satellite).

### 2.2    Model set-up

The Met Office Unified Model (UM) (Brown et al., 2012) version 13.0 with Regional Atmosphere and Land (RAL) 3.2 80 configuration (Bush et al., 2020, 2022) was used in this study. A 1500 km by 1500 km regional domain with 1.5 km grid spacing and centred at 59 °N, 52 °W was nested within a global model (N216, $\cong$ 60 km grid spacing near the mid-latitudes) with Global Atmosphere and Land (GAL) 6.1 configuration (Walters et al., 2017). The nested model domains are shown in Appendix C. Using a 1.5 km grid spacing has shown a good ability to reproduce the general features of the CAO cloud system (e.g., the stratus and cumulus regions) from Field et al. (2017a). Such design of the regional domain balances sufficient 85 coverage of the CAO cloud system and the computational cost. There were 70 vertical levels in the nested region up to 40 km (28 model levels below 3 km where most of the cloud are in both cases) and the time step is 60 seconds for the regional model. The lateral boundary is provided to the regional model from the global model every hour. The simulations were initialized from

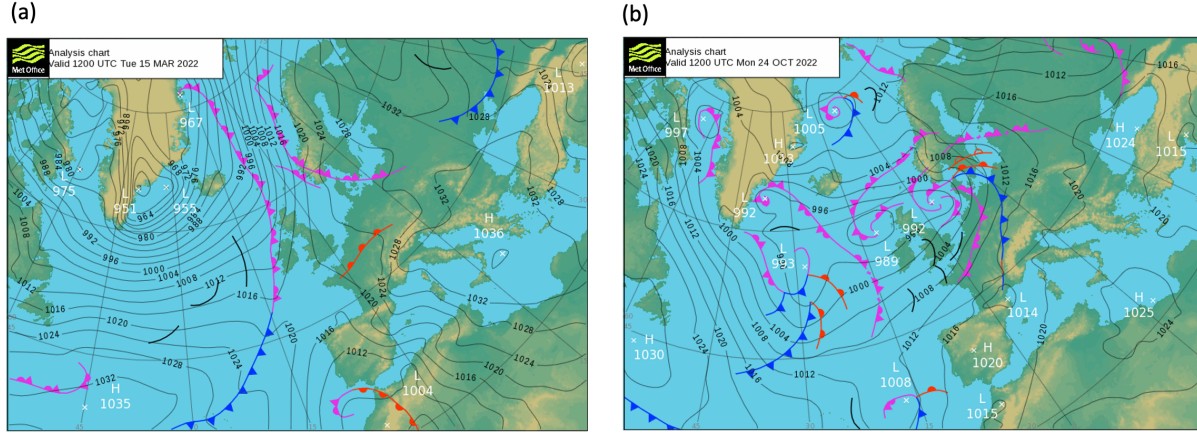

**Figure 1.** The UK Met Office surface analysis charts at 1200 UTC on (a) 15 March 2022 and (b) 24 October 2022.

archived global model analysis at 0000 UTC on the case date and run for 24 hours. The first 12 hours were excluded from the analysis due to model spin-up.

Cloud microphysics are parametrized with the double-moment bulk Cloud AeroSol Interacting Microphysics (CASIM) scheme (Shipway and Hill, 2012; Grosvenor et al., 2017; Field et al., 2023). There are five hydrometeor species in CASIM: cloud liquid, rain, cloud ice, snow and graupel, with a generalized gamma distribution for the particle size distributions (PSD). CASIM provides two options to calculate the droplet number concentration ($N_d$), prescribing a fixed in-cloud $N_d$ or deriving $N_d$ from the background aerosol. The prescribed fixed $N_d$ option was selected in this study for an easier perturbation of $N_d$

and interpretation of the results. However, it is worth noting that using fixed in-cloud $N_d$ instead of having aerosols involved can remove potential feedbacks between aerosols and clouds, for example, precipitation formed in clouds can remove aerosols/CCNs which can enhance the precipitation and further remove the aerosols. Details on the selection of parameter values are shown in Section 2.3 below. For heterogeneous ice nucleation on INPs (primary ice production, PIP), we use the parameterization of Cooper (1986). The Cooper approach is a parameterization for ice crystal number concentration ($N_{ice}$), but because

we assume that one INP can produce one ice crystal, the Cooper approach is treated as an INP parameterization in this study. Heterogeneous ice nucleation is assumed to occur in grid boxes with temperatures lower than -8 °C, and higher than -38 °C when homogeneous ice nucleation can occur. Bigg's parameterization for rain freezing (Bigg, 1953) was switched off in this study to avoid potentially unrealistic formation of graupel in convective clouds. The secondary ice production (SIP) process implemented in CASIM is the Hallet-Mossop (HM) process (Hallett and Mossop, 1974), which produces ice splinters through

riming between -2.5 °C and -7.5 °C with a peak efficiency at -5 °C. The rates are calculated from cloud liquid accreted by graupel and snow with a default efficiency of 350 ice splinters produced per milligram-rimed cloud liquid. Other SIP mechanisms (e.g., collision fragmentation (Vardiman, 1978; Takahashi et al., 1995) and droplet shattering (Latham et al., 1961)) are currently in development and hence not available for use in this study. When both ice and liquid exist in the same grid box, the overlap mixed-phase fraction is calculated, with a fixed mixed-phase overlap factor of 0.5 (Field et al., 2023). The mixed-

phase overlap factor controls a function that quantifies the overlap during the run time of the model rather than being a fixed mixed-phase overlap in a model grid box. If the overlap factor is set to 1, then the subgrid liquid and ice cloud are maximally overlapped. If the overlap factor is 0, then the subgrid liquid and ice are not overlapped as long as $CF_{\text{liq}} + CF_{\text{ice}} < 1$, where $CF$ refers to cloud volume fraction. Once the combined cloud fraction goes above 1, there will be overlap. For an overlap factor of 0, the overlap is minimised. Values of the overlap factor in between lead to increasing overlap, but once either the

liquid or ice cloud fraction reaches 1, then mixed-phase overlap is maximum whatever the overlap factor is set to. See section A.6 in the documentation of CASIM implementation in UM from Field et al. (2023) for more information.

The cloud parameterization in the nested UM is a diagnostic cloud scheme that uses skewed and bi-modal probability density function for sub-grid saturation departure. This Bimodal Cloud scheme diagnoses the cloud volume fractions and condensed liquid water amounts in each grid box and passes them to CASIM. It is the Bimodal Cloud scheme that handles

the condensation and evaporation between water vapour and cloud liquid via a saturation adjustment approach justified by the long model timestep (approx. 60 s) compared to the timescales of saturation adjustment (approx. 1 s). This process happens at a different point of the time step from CASIM. A detailed technical explanation of coupling CASIM to the cloud schemes in UM is discussed in Field et al. (2023).

The radiative processes in the simulations are represented by SOCRATES (Suite Of Community Radiative Transfer codes

based on Edwards and Slingo) (Edwards and Slingo, 1996; Manners et al., 2023), which calculates radiative fluxes using the two-stream method and radiance using spherical harmonics. The single-scattering properties of water droplets are dependent on the mass mixing ratio of liquid water and the effective radius of the droplets (Slingo and Schrecker, 1982). In this study, the single-scattering properties of ice crystals are calculated using an equivalent mass spherical radius with both the ice water mass mixing ratio and ice hydrometeor number concentration ($N_{\text{ice}}$) from CASIM. This allows a Twomey-like effect to be included

from changes in $N_{\text{ice}}$.

### 2.3 Perturbed parameters and the selection of their values

Three model input parameters are perturbed in this study: the prescribed fixed in-cloud droplet number concentration ($N_{\text{d}}$), the scale factor of INP concentration ($S_{\text{INP}}$), and the ice multiplication efficiency of the Hallet-Mossop process ($E_{\text{HM}}$). Table 1 shows the values used for each simulation and the selection of parameter values are explained below.

**2.3.1 Droplet number concentration ($N_{\text{d}}$)**

CASIM provides two options for calculating $N_{\text{d}}$. Here we use fixed in-cloud $N_{\text{d}}$ to allow an easier interpretation of results instead of deriving $N_{\text{d}}$ from background aerosol. The grid-box mean $N_{\text{d}}$ is calculated by multiplying the fixed in-cloud $N_{\text{d}}$ with the liquid cloud fraction in the grid box from the Bimodal Cloud scheme. The default value of the fixed in-cloud $N_{\text{d}}$ is 150 $\text{cm}^{-3}$, and we selected 10 $\text{cm}^{-3}$ for low $N_{\text{d}}$ and 500 $\text{cm}^{-3}$ for high $N_{\text{d}}$ simulations based on values from (Wood, 2012)

for general stratocumulus clouds. This range also covers the observations from the M-Phase measurements (Murray and the MPhase Team, 2024; Tarn et al., 2025) and warm cloud $N_{\text{d}}$ derived from satellite retrievals (Grosvenor et al., 2018) in this region.

### 2.3.2 Scale factor of INP concentration ($S_{INP}$)

The INP parameterization used in this study is the default heterogeneous ice nucleation parameterization from Cooper (1986) assuming that one INP produces one ice crystal. Here we use a scale factor $S_{INP}$ (unitless) to change the INP concentration from the default Cooper parameterization:

$$N_{INP}(T) = S_{INP}(5e^{-0.304(T_0-T)}), \tag{1}$$

The unit of $N_{INP}(T)$ is m$^{-3}$. The default value of $S_{INP}$ is 1.0. $T_0$ is 273.15 K and $T$ is the ambient temperature (K). 265.15 K (-8 °C) was chosen as the warmest condition for ice nucleation, meaning there are no INPs at temperature higher than -8 °C. We selected 0.0001 for low $S_{INP}$ and 100 for high $S_{INP}$ simulations to cover the majority of the INP measurements at high latitude regions of the Northern Hemisphere (Figure 2a) and the INP measurements from the M-Phase aircraft campaign (Figure 2b, Murray and the MPhase Team (2024); Tarn et al. (2025)). There is also a parameter in the Cooper parameterization that defines the dependence of $N_{INP}$ on temperature. Although it has been shown important for deep convective anvil cirrus (Hawker et al., 2021a, b), it plays a secondary role in the CAO clouds of interest here as the CAO clouds are generally thin (Fletcher et al., 2016b) and the slope in the default Cooper approach matches reasonably well with most of the M-Phase measurements in Figure 2b.

### 2.3.3 Efficiency of the Hallet-Mossop Process ($E_{HM}$)

The Hallet-Mossop (HM) process is included as the only SIP process in this study with a default efficiency of 350 ice splinters produced per milligram of rimed cloud liquid (Hallett and Mossop, 1974; Field et al., 2023):

$$P_{HM} = E_{HM}(P_{gacw} + P_{sacw})f(T)M_{I0}, \tag{2}$$

where $P_{HM}$ is the mass of ice produced from the HM process, $E_{HM}$ is the HM process efficiency perturbed in this study with a default value of 350 mg, $P_{gacw}$ is the rate of graupel accretes cloud water, $P_{sacw}$ is the rate of snow accretes cloud water, $M_{I0}$ is the produced splinter mass of $10^{-18}$ kg, and $f(T)$ is a triangular function between -2.5 °C and -7.5 °C with peak at -5 °C when $f(T) = 1$. Clouds formed in the October case span this range of temperatures, while cloud temperature in the March case are much colder with little clouds formed in this range of temperatures (Section 3.1).

We selected 10 mg for low $E_{HM}$ and 7000 mg for high $E_{HM}$ simulations. The high $E_{HM}$ value was selected following the studies by Young et al. (2019) and Sotiropoulou et al. (2020) to show good agreement with observed $N_{ice}$ when only the HM process is implemented in the model. The low $E_{HM}$ was selected to test the effects of reducing the HM process but not completely removing it. It is worth noticing here that self-limiting feedback may exist when using high $E_{HM}$ (Field et al., 2017b) which can potentially limit the increase of ice splinters produced by increasing the $E_{HM}$ through stronger removal of liquid for riming.

**Table 1.** Configurations of simulations for both case studies. Cell content marked by "-" means that the value used for the parameter is the same as the configuration in the control simulation.

| Model Configuration | Fixed $N_d$ (cm$^{-3}$) | $S_{INP}$ | $E_{HM}$ (mg$^{-1}$) |
|:---:|:---:|:---:|:---:|
| Control | 150 | 1 | 350 |
| low $N_d$ | 10 | - | - |
| high $N_d$ | 500 | - | - |
| low $S_{INP}$ | - | 0.0001 | - |
| high $S_{INP}$ | - | 100 | - |
| low $E_{HM}$ | - | - | 10 |
| high $E_{HM}$ | - | - | 7000 |

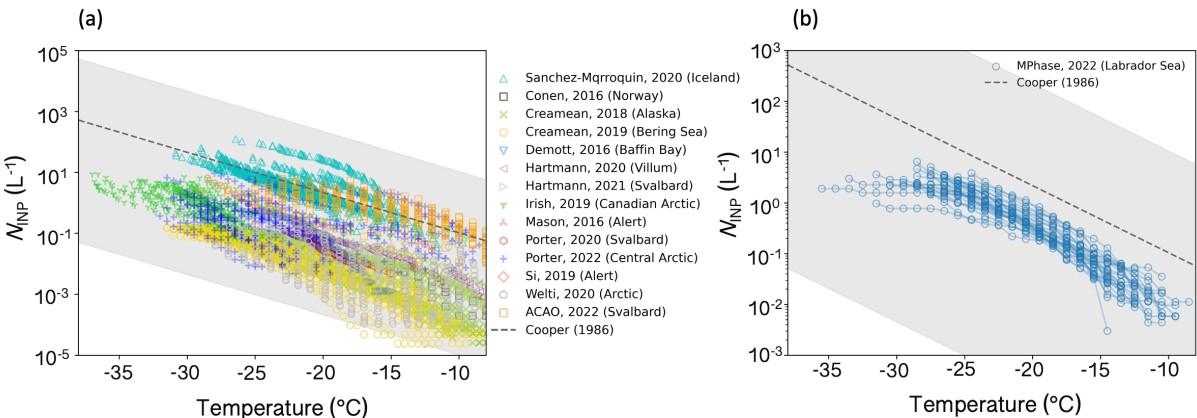

**Figure 2.** Perturbed INP range in this study compared to INP measurements at high-latitude regions in the Northern Hemisphere from (a) literature data (Sanchez-Marroquin et al., 2020; Franz Conen, 2016; Creamean et al., 2018, 2019; DeMott et al., 2016; Hartmann et al., 2020, 2021; Irish et al., 2019; Mason et al., 2016; Porter et al., 2020, 2022; Si et al., 2019; Welti et al., 2020; Raif et al., 2024) and (b) M-Phase aircraft campaign (Tarn et al., 2025). INP measurements from each flight during the M-Phase aircraft campaign are connected with lines to compare the INP concentration slope with the default Cooper parameterization slope. The top and bottom boundary of the shaded area are the upper and lower perturbed range of INP concentration in the sensitivity test.

## 2.4 Satellite data

Multiple satellite data products were used in this study to compare model output with observations. Figure 3 shows the satellite data products for the two CAO cases including RGB composites (a and e) using bands 1 (620-670 nm), 3 (459-479 nm) and 4 (545-565 nm) from MODIS (Moderate Resolution Imaging Spectroradiometer) Level 1B Calibrated Radiances Product (Collection 6.1) (MODIS Characterization Support Team (MCST), 2017); the Single Scanner Footprint (SSF) of top-of-atmosphere shortwave flux ($F_{SW}^{TOA}$, b and f) and longwave flux ($F_{LW}^{TOA}$, c and g) from the CERES (Clouds and the Earth's Radiant Energy System) instrument (Edition 4A) (Su et al., 2015a, b); and the cloud top temperature (d and h) from the MODIS Atmosphere Level 2 Cloud Product (Collection 6.1) (Platnick et al., 2015). All-sky liquid water path (LWP) with 0.25° spatial resolution retrieved from the AMSR-2 (Advanced Microwave Scanning Radiometer) columnar cloud liquid water product (version 8.2) (Wentz et al., 2014) and cloud water path (CWP) for both liquid and ice and cloud cover from MODIS Atmosphere Level 2 Cloud Product (Collection 6.1) (Platnick et al., 2015) were also used for model-observation comparison, shown in the Results Section below. A table of retrieval time and selected model time points for each satellite product is shown in Appendix A.

The MODIS and CERES instruments are onboard NASA's Aqua satellite and the AMSR-2 instrument is onboard the JAXA's (The Japan Aerospace Exploration Agency) GCOM-W (Global Change Observation Mission – Water) satellite. Both polar-orbiting satellites have the same equator crossing time of 1:30 p.m. while ascending, and similar altitudes for their orbits. This means satellite retrievals can be made close to each other in time. Geostationary satellite products are not used in this study due to large uncertainties in retrievals for high-latitude regions (Seethala and Horváth, 2010).

Although the two selected CAO events shared similar synoptic situations as shown in Figure 1, the cloud top temperatures of them were strongly different. The March case had much colder cloud top temperatures with peak around -30 °C, compared to the ones in the October case with peak around -10 °C. Monthly distributions for cloud top temperature of low-level, mixed-phase clouds during CAO events over the Labrador Sea in 2022 are shown in Appendix B using the ERA5 (ECMWF ReAnalysis version 5) (Hersbach et al., 2020) dataset and CTT retrieved from MODIS. The CTTs in the March case are near the colder end of the shown distributions, while the ones in the October case are more close to the warmer end, which suggests that these two CAO cases are nicely contrasting from each other in terms of CTTs and sit near the boundaries of CTT ranges in CAO clouds over the Labrador Sea. These two cases were chosen on the basis that they represent end members of the temperature range of mixed-phase CAO clouds. Detailed information for the method of analysis is shown in Appendix B.

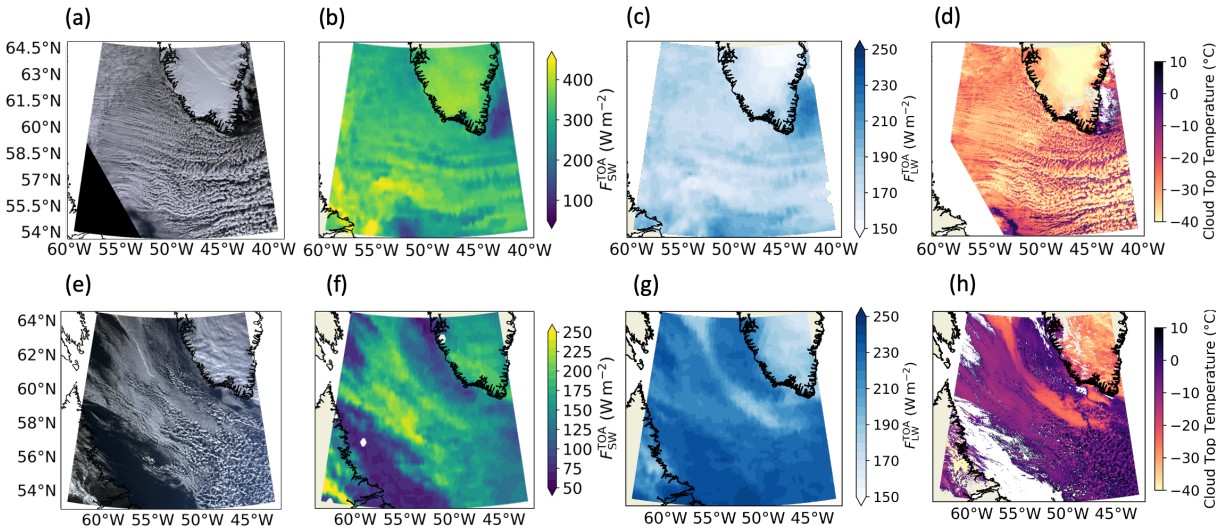

**Figure 3.** Satellite retrievals for CAO events over the Labrador Sea on 15 March 2022 (top) and 24 October 2022 (bottom): RGB imagery (a, e), top-of-atmosphere shortwave flux (b, f), top-of-atmosphere longwave flux (c, g), and cloud top temperature (d, h). Note that the scales of the colour bars for the shortwave radiation flux are different for these two cases due to different satellite retrieval times.

## 3 Results

### 3.1 Control simulations

Control simulations with the default model set-up are introduced and compared in this section. Figure 4 shows the modelled in-cloud cloud water path (CWP), which is the sum of the in-cloud liquid water path (LWP) and the in-cloud ice water path (IWP), for both cases compared with MODIS retrieved in-cloud CWP. This comparison acts as a qualitative check of whether our model can simulate the main synoptic features of the CAO cloud system. MODIS-retrieved CWP data were re-gridded to the same spatial resolution as the modelled CWP (1.5 km) using the nearest-neighbour method. The satellite retrieval time and selected model output time point are shown in Appendix A.

In general, the March CAO event has a less broken cloud field with higher CWP compared to the October CAO event. For both cases, the control simulations capture the cloud regimes (e.g., stratus and open cells) during the CAO event, and reproduce the large-scale synoptic structures and the locations of the CAO event well compared with the MODIS retrievals. In the March case, the model struggles with reproducing the fine structures of cloud streets to the east of 54 °W in the sub-domain. This is because the effective horizontal resolution (between 5 times to 10 times of the grid spacing) in our model cannot fully resolve the cloud streets at the beginning of the CAO event. With the clouds moving into the convective region and the boundary layer getting depended (to the west of 54 °W), the scales of cloud street grow and can then be resolved in the model. A better representation of the cloud streets at the beginning of the CAO event requires higher model resolution (Field et al., 2017a) and therefore much higher computational resources to conduct the sensitivity test, hence are not further investigated here. Our control simulations generally have higher in-cloud CWP compared with the MODIS, which may because of different definitions of cloudy pixels between model output and satellite products, uncertainties in the MODIS-retrieved CWP for high-latitude mixed-phase clouds (Khanal and Wang, 2018), as well as potential overestimation of CWP from our model.

Note that the CWP comparison is only qualitative here, and quantitative model-observation comparisons of the control simulations with satellite retrievals of cloud top temperature from MODIS, temperature and IWC (ice water content) from CALIOP for the March case, and MPhase aircraft measurements of temperature, cloud water content and liquid water fraction for the October case are shown and discussed in Appendix D. Quantitative comparisons of all-sky LWP, shortwave (SW) and longwave (LW) fluxes between model output from all simulations (control and sensitivity test simulations) and satellite retrievals are shown later in Section 3.5.

Cross-section mean cloud properties within the highlighted sub-domain (yellow parallelogram) in Figure 4 of both cases are presented and compared in Figure 5, with supplementary information shown in Appendix E. The sub-domain was selected to be aligned with directions of wind and cloud movements, and the cross-section mean is calculated by averaging along the y-axis of the sub-domain parallelogram. The locations of the sub-domains in the nested model domains are shown in Appendix C. The whole sub-domain in the March case and most of the sub-domain (except the small northwestern part) of the October case are sufficiently distant from the boundaries of the nested model domain, hence not affected by the boundary effects from fields entering from the global model. A detailed discussion on the boundary effects is shown in Appendix C.

Both cases experience a general west-to-east reducing trend of cloud cover (Figure 5a) in the sub-domain along the direction of cloud movements to the open ocean, with the March case having a generally higher cloud cover (>0.9 for most of the cross-section) compared to the one in October. The in-cloud LWP (Figure 5b) is much higher in October, with the peak LWP happening around 150 g m$^{-2}$ near 56 °W in October and less than 20 g m$^{-2}$ near the eastern boundary of the sub-domain in March. The trend of LWP changing from the west to the east of the sub-domain is different in these two cases: a general increasing trend in March while the LWP first increases to the peak value and then reduces in October. The in-cloud IWP (Figure 5c) is much higher in the March case with the peak around 750 g m$^{-2}$ near 50 °W. The in-cloud IWP starts at a low level in October and generally increases from west to east with the peak value slightly lower than 300 g m$^{-2}$ near the the eastern boundary of the sub-domain. Liquid water fraction, the ratio of LWP to CWP, is calculated to show the liquid-ice partitioning in both cases (Appendix E). The dominant cloud water are in ice-phase in the March case, while the liquid-phase dominates at the western region in the October and later ice-phase dominates when clouds move towards east. Both cases experience little precipitation in the western region and enhanced precipitation when clouds move to further east (Figure 5d), with the dominant type of precipitation in March being snow and rain in October (Appendix E).

As the SW radiation dominates cloud radiative effects in shallow mixed-phase clouds, here we use albedo at the top-of-the-atmosphere (TOA) in Figure 5e to investigate the CAO cloud radiative properties. The outgoing shortwave and longwave flux are shown in Appendix E. The overall trend of albedo changing from the west to the east of the sub-domain is very similar in these two cases, with the albedo slightly higher in October. By comparing the trend with other cloud properties mentioned above, it is shown that albedo is strongly affected by the cloud cover in both of the two cases, but influenced more from in-cloud IWP in March, while more from in-cloud LWP in October. This is due to the liquid-ice partitioning in their control simulations varies and the liquid-ice partitioning is strongly controlled by the cloud temperature with a same temperature-dependent INP parameterization (as well $N_d$ and $E_{HM}$). Cross-section mean for cloud profiles (cloud volume fraction and total water content) with ambient temperature are shown in Appendix E. Most of the clouds in the March case in the sub-domain are between -15 °C and -35 °C, while the ones in the October case having a much warmer ambient temperature most between 0 °C and -15 °C. Such temperature difference can directly lead to different efficiencies of many temperature-dependent cloud microphysics processes including the INP concentration and HM efficiency perturbed in this study during these two cases.

Meteorological variables and environmental conditions of boundary layer are also shown and compared for the control simulations of these cases as shown in Figure E2 of Appendix E. The SST (sea surface temperature) increases as clouds move eastward, with the SST temperature lower in the March case (2-3 °C) compared to the one in the October case (4-7 °C). Note that the SST was prescribed based on daily forecasting analyses in all our model simulations. The March case is a much stronger CAO event with the highest CAO index at 800 hPa ($M_{800}$) almost reaching 20 K at the western boundary of the sub-domain, while the highest $M_{800}$ is around 1.5 K in the October case. The more unstable boundary layer in the March case is consistent with a lower LTS (lower tropospheric stability) compared to the one in the October case. Both cases experience an EIS (estimated inversion strength) over 5 K at the western boundary of their sub-domain with a decreasing trend to the east.

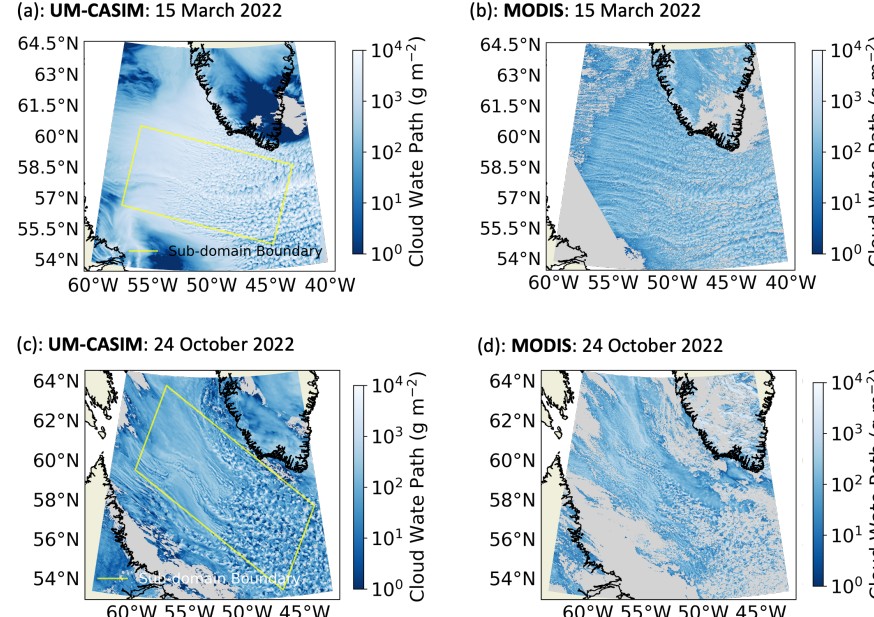

**Figure 4.** In-cloud cloud water path (CWP) from the control UM-CASIM simulations and MODIS retrievals on 15 March 2022 (a,b) and 24 October 2022 (c,d). The model reproduces the general CAO cloud system in both case studies well when compared to satellite retrievals. In-cloud cloud water path retrieved from MODIS is not used for a quantitative comparison due to its large uncertainties. Sub-domains of interest for both cases are highlighted in yellow. Model output pixels with less than 20% cloud cover are excluded before calculating the in-cloud values. Times of model output and satellite retrieval time are shown in Appendix A.

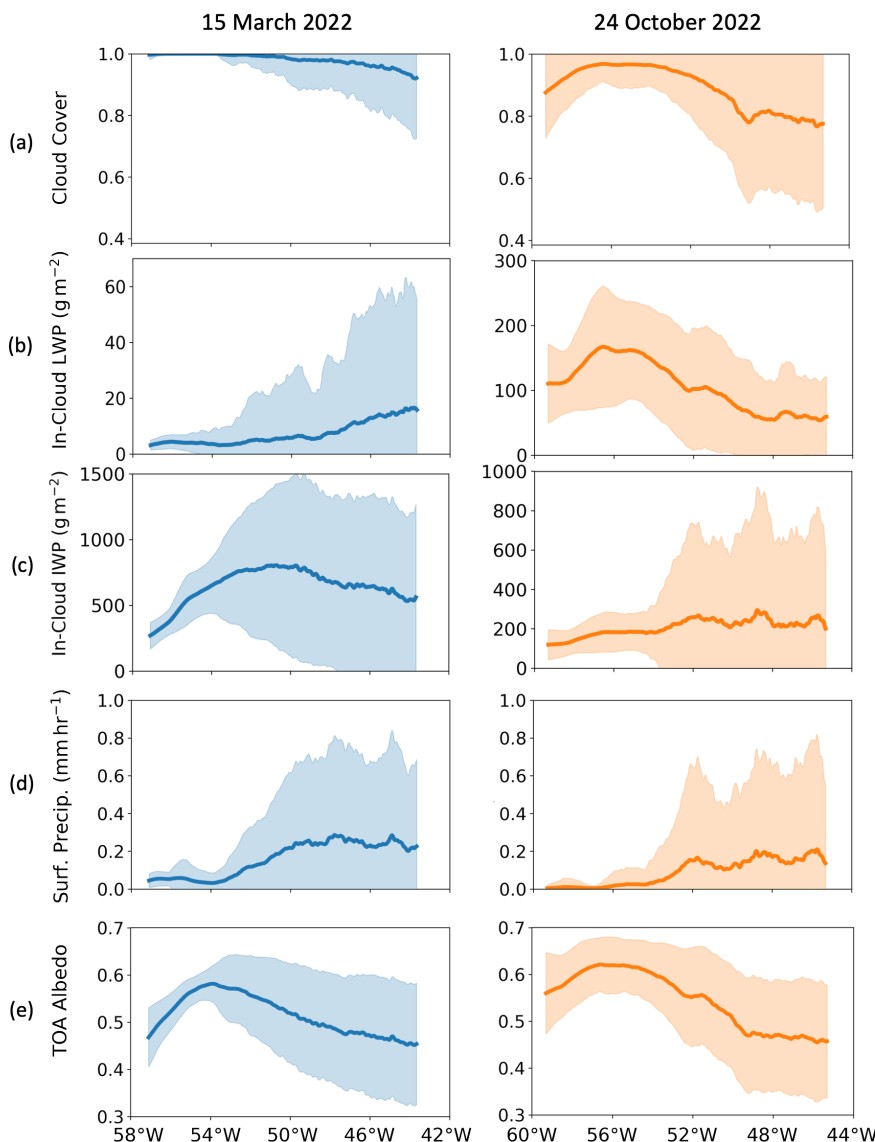

**Figure 5.** Cross-section mean (averaging along y-axis of the sub-domain parallelogram shown in Figure 4) cloud properties from the March (left panel) and the October (right panel) control simulations: (a) cloud cover, (b) in-cloud liquid water path (LWP), (c) in-cloud ice water path (IWP), (d) surface precipitation, and (e) albedo at the top-of-the-atmosphere (TOA albedo). The time points selected are 16:45 UTC for the March case and 17:00 UTC for the October case, which are consistent with the corresponding CERES measurement times. Grid boxes with cloud cover smaller than 20% were removed before averaging for calculation of the in-cloud LWP and in-cloud IWP. The shaded area indicates the range of +/- 1 standard deviation.

## 3.2 Responses of cloud properties to perturbed parameters

In this section, the responses of the cloud properties to the perturbations in droplet number concentration ($N_d$), the INP concentration ($S_{INP}$) and the efficiency of the HM process ($E_{HM}$) are compared between the two cases. An overall comparison between the two cases as well as the cloud profiles are shown later in Section 3.4.

### 3.2.1 15 March 2022

Here we first present the responses of cloud cover, in-cloud LWP, in-cloud IWP, surface precipitation rate, and TOA albedo in Figure 6 for the March case, as well as other properties shown in Appendix F (Figure F1). There is limited influence from perturbing $N_d$ (left panel) and $E_{HM}$ (right panel) on the cloud properties in March. Effects from these parameters are small due to little water in the control simulation and most of clouds are out of the active temperature range for the HM process in March as shown in Figure E3 of Appendix E.

Perturbing $S_{INP}$ (middle column of panels) has a much stronger influence on all the column cloud properties shown here than perturbing $N_d$ or $E_{HM}$. With a higher INP concentration, the modelled CAO clouds experience a higher cloud cover from around 54 °W to the eastern boundary of the sub-domain and a higher in-cloud IWP throughout the sub-domain (stronger increase in the eastern region) along with a lower surface precipitation. The in-cloud LWP slightly decreases only slightly because there is so little liquid water in the control simulation, with a similar change for the liquid cloud fraction (Appendix E). The limited influence from lower in-cloud LWP is then offset by the higher cloud cover and higher in-cloud IWP, resulting in a general higher TOA albedo in the high INP concentration simulation.

The responses of in-cloud LWP, IWP and liquid water fraction are consistent with previous studies (Abel et al., 2017; Tornow et al., 2021), but the responses of cloud cover, surface precipitation and TOA albedo differ from the previous studies, where the CAO cloud cover becomes smaller with a higher INP concentration (Tornow et al., 2021). Similarly, Abel et al. (2017) found that cloud cover was increased when they prevented ice formation. In this March case, the increase in cloud cover with higher INP concentration is due to the small amount of liquid water in the control simulation so that the increase in ice concentrations has only a small effect on the further removal of liquid water. In addition, with a higher INP concentration, there is a higher ice number concentration and the autoconversion from ice crystals to snow becomes slower, as well as the mean ice hydrometeor particle size becomes smaller resulting in a lower mean fallspeed and reduced sedimentation flux. These effects lead to less precipitation and slower removal of cloud water, resulting in a greater cloud cover. The responses here to a higher INP concentration are similar to what we would expect in cirrus clouds.

With a lower INP concentration, a strong increase in the in-cloud LWP is seen in the western sub-domain, with the peak value over $300\,\mathrm{g\,m^{-2}}$. However, a sharp reduction of in-cloud LWP follows around 53 °W to 52 °W as well as a strong surface precipitation at the same location. This strong removal of liquid water from clouds limits the influence of decreasing INP on increasing the in-cloud LWP in the rest of the sub-domain. Compared to the control simulation, lower INP concentration has limited influence on the cloud cover and leads to a generally lower in-cloud IWP throughout the sub-domain. The change of TOA albedo is slightly complex: in the western sub-domain before the liquid water being rapidly removed, a higher albedo is

seen by having a much higher LWP; after the strong removal of LWP, the albedo reduces quickly and becomes lower than the one in the control simulation, which is a result of the lower in-cloud IWP. While the enhanced reflectivity with decreased INP is confined to the beginning of the sub-domain, the response to INP in the rest of the sub-domain extends over a massive area
stretching out into the Atlantic and dominates the radiative effect of the INP over the sub-domain region as a whole.

### 3.2.2  24 October 2022

A similar analysis for the October CAO case is shown both in Figure 7 and Appendix F (Figure F2). Unlike the March case where only the $S_{INP}$ simulation strongly influences the cloud properties, all three perturbed parameters show clear and various influences in this October case, and some responses of cloud properties vary in the CAO development from west to east.

Perturbing $N_d$ now has a strong influence on CAO cloud cover, in-cloud LWP and TOA albedo in this October case. A low $N_d$ leads to lower cloud cover and LWP along with a higher surface precipitation, which is consistent with the Albrecht effect (Albrecht, 1989). With a high $N_d$, there is limited influence on cloud cover at the beginning of the CAO cloud system, and this is because the precipitation rate is very small in the control simulation at this location, hence cannot be further suppressed with a higher $N_d$. The responses of the TOA albedo to both low and high $N_d$ are the strongest among all the sensitivity test
simulations in October, and is consistent with the Twomey effect (Twomey, 1977).

    The responses of cloud properties to the perturbation of $S_{INP}$ from the western boundary to around 50 °W are similar to the CAO cloud responses to INP concentration or ice in previous studies (Abel et al., 2017; Tornow et al., 2021). The responses become complex and some even non-monotonic near the eastern end of the sub-domain. Until around 50 °W, a higher INP concentration results in lower cloud cover, in-cloud LWP and TOA albedo, with higher in-cloud IWP and surface precipitation,
vice versa.

    Various responses of cloud properties are also seen when perturbing $E_{HM}$. With a high $E_{HM}$, which means a more efficient HM process in the simulation, the cloud cover, surface precipitation and TOA albedo are only affected strongly close to the eastern end of the sub-domain. This is because the HM process in the model is dependent on the processes of cloud water accretion onto graupel and snow, while ice is very limited in the stratocumulus-dominated region but a higher IWP is seen
after SCT. A high $E_{HM}$ leads to a higher cloud cover, a lower surface precipitation rate and a higher TOA albedo. This is because although the HM process is the source of ice crystals, it is also the sink for graupel and snow which can accrete and remove water through precipitation in the model. A high $E_{HM}$ results in a lower amount of graupel and snow, hence leading to a smaller precipitation. The responses of in-cloud LWP and in-cloud IWP are consistent throughout the sub-domain, with a high $E_{HM}$ resulting in lower in-cloud LWP (through riming) and higher in-cloud IWP (slow snow autoconversion with a high
ice crystal number concentration). These influences become stronger in the eastern region. Although low $E_{HM}$ has a limited influence compared to the ones from high $E_{HM}$, one might notice that the responses of surface precipitation to $E_{HM}$ become complicated and non-monotonic (e.g., the default model output is outside the low and high model output range) near the end of the sub-domain where cumulus clouds dominate. For example, low $E_{HM}$ results in a stronger surface precipitation from 52 °W to 50 °W, but a weaker surface precipitation around 46 °W, compared to the precipitation from the control simulation.
This may because that precipitation rate is a state-dependent variable, and a low $E_{HM}$ leads to an earlier peak of precipitation

when clouds move eastward compared to the one from the control simulation, followed by a lower precipitation rate later as less cloud water exist.

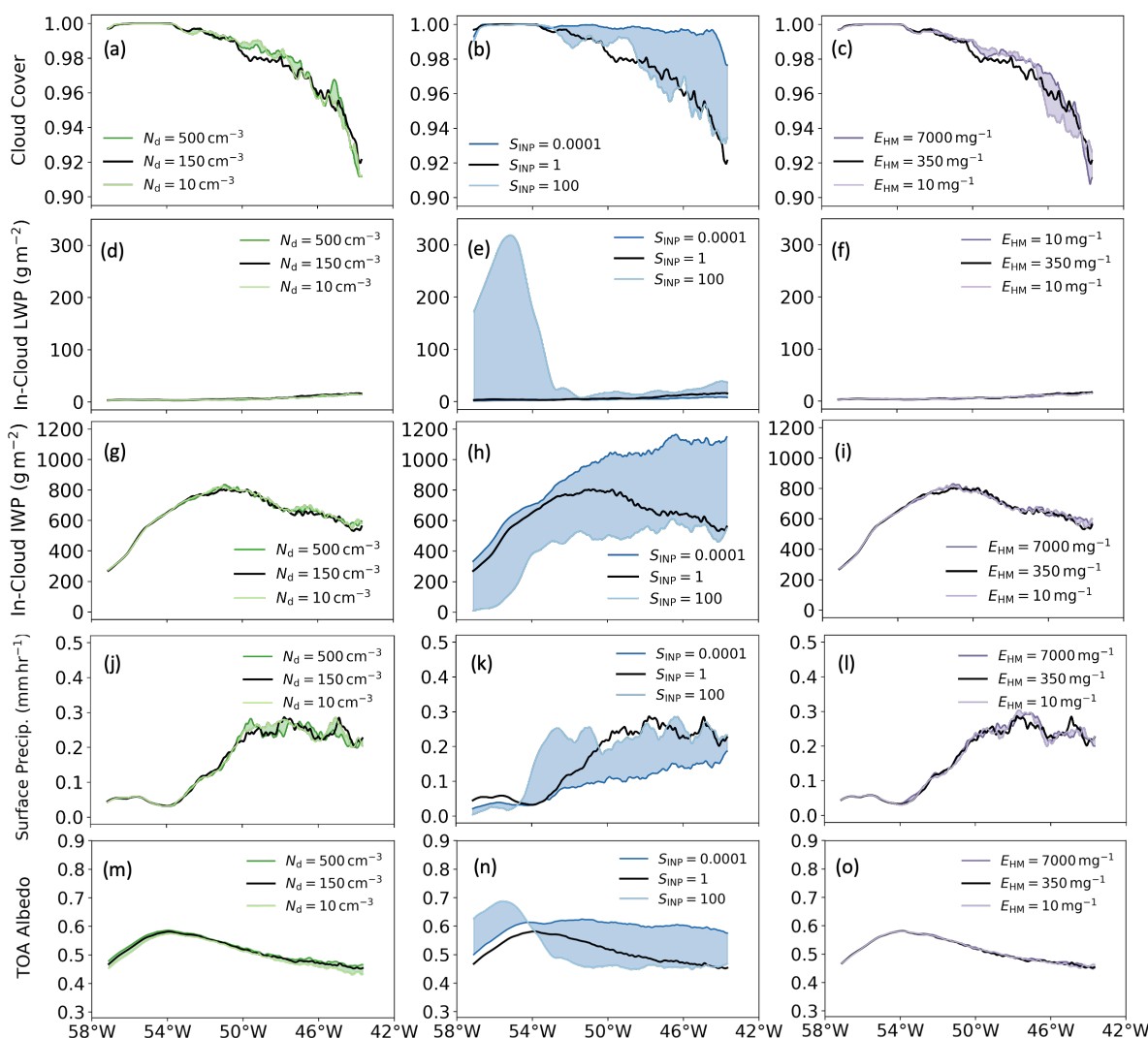

**Figure 6.** Responses of cross-section mean CAO cloud properties to the three perturbed parameters on 15 March 2022 at 16:45 UTC: (a)-(c) cloud cover, (d)-(f) in-cloud liquid water path (LWP), (g)-(i) in-cloud ice water path (IWP), (j)-(i) surface precipitation, and (m)-(o) albedo at the top-of-the-atmosphere (TOA albedo). Grid boxes with cloud cover smaller than 20% were removed before calculating in-cloud LWP and IWP. 16:45 UTC was chosen for the corresponding CERES measurements of radiation on 15 March 2022. Space between the variable data from the high and low simulations is filled to highlight the range of variables and identify non-monotonic behaviours (e.g., data from the control simulation are not in the shaded space).

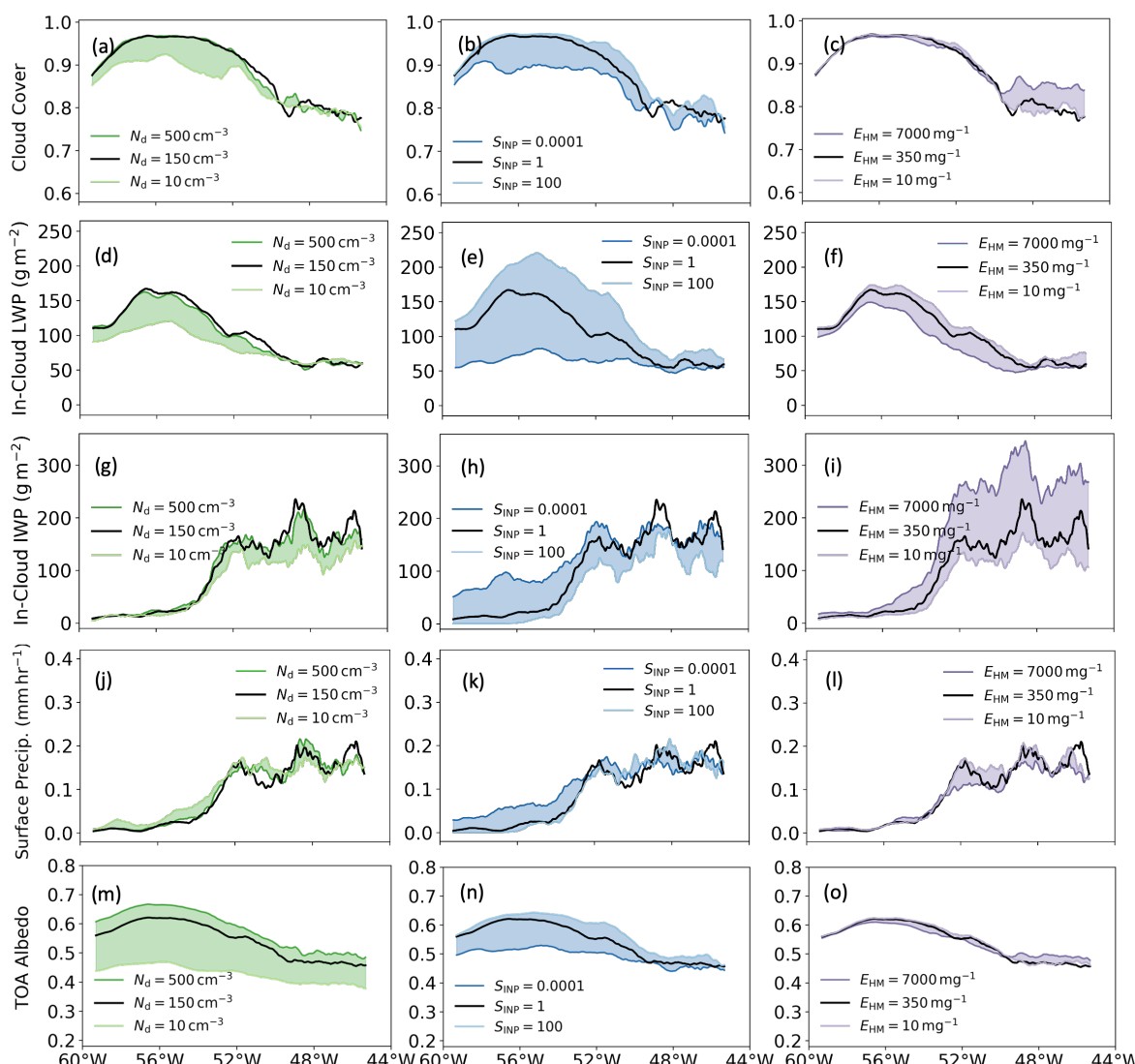

**Figure 7.** Responses of cross-section mean CAO cloud properties to the three perturbed parameters on 24 October 2022 at 17:00 UTC: (a)-(c) cloud cover, (d)-(f) in-cloud liquid water path (LWP), (g)-(i) in-cloud ice water path (IWP), (j)-(l) surface precipitation, and (m)-(o) albedo at the top-of-the-atmosphere (TOA albedo). Grid boxes with cloud cover smaller than 20% were removed before calculating in-cloud LWP and IWP. 17:00 UTC was chosen for the corresponding CERES measurements of radiation on 24 October 2022. Space between the variable data from the high and low simulations is filled to highlight the range of variables and identify non-monotonic behaviours (e.g., data from the control simulation are not in the shaded space).

### 3.3 Responses of CAO cloud field development to perturbed parameters

Cloud field development including stratocumulus-to-cumulus transition (SCT) is important for the radiative properties of CAO clouds. In this section, we use the cloud field homogeneity parameter ($\nu$) calculated using in-cloud cloud water path (CWP), which has been successfully used for identifying cloud field transition from satellite retrievals (Wood, 2012; Wu and Ovchinnikov, 2022), to understand how the perturbed parameters affect the cloud field development in the two selected cases. The cloud field homogeneity parameter ($\nu$) is calculated using the squared ratio of the mean ($\bar{x}$) to the standard deviation ($\sigma$) for CWP over 20 by 20 grids: $\nu = (\frac{\bar{x}}{\sigma})^2$.

Figures shown in this section are the cross-section mean of the cloud field homogeneity parameter in the selected subdomain. A sharp decrease in the cloud field homogeneity parameter generally implies a transition from stratocumulus clouds (Sc) to cumulus clouds (Cu). We also qualitatively determine the stratocumulus-dominated, transition and cumulus-dominated regions by using the trends of the cloud field homogeneity parameter and the fraction of the 20 by 20 grids that are of the cumulus-capped boundary layer type. The methods for determining boundary layer type in the UM are shown in Lock (2001). A quantitative determination of where the transition happens is not conducted in this study and requires further research on using the cloud field homogeneity parameter for model output. Note that we did not define the regions for different simulations individually to avoid any location effects. A detailed description of how we qualitatively define these regions can be found in the captions of Figure 8 and Figure 9 .

Figure 8 shows the evolution of cloud morphology (in terms of the spatial distribution of CWP) together with the homogeneity parameter and the cumulus-capped boundary layer fraction in the March case. The contribution of other boundary layer types for all the simulations is shown in Appendix G. Figure 8d shows the CWP field with different $S_{INP}$ and the stratocumulus-dominated, transition and cumulus-dominated regions are separated by dashed grey lines.

Similar to the cloud properties shown in Figure 6, the overall development of the cloud field in March is only strongly influenced by $S_{INP}$. With a higher INP concentration, the CAO cloud field begins with a more heterogeneous stratocumulus-dominated region a higher cumulus-capped boundary layer fraction, and a slightly earlier transition to cumulus-dominated region, indicated by an earlier sharp decrease of the cloud field homogeneity parameter and increase of the cumulus-capped boundary layer fraction.

To test whether the effect of INP on cloud field homogeneity and boundary layer types is caused by modifications to precipitation through the precipitation-induced SCT mechanism from Abel et al. (2017), we performed additional simulations in which precipitation, or evaporation and sublimation of precipitation were turned off, and a simulation without precipitation, evaporation or sublimation (denoted "no-all") shown in Appendix H1 and H2. Most of the difference in cloud field development is removed in the no-precipitation and no-all simulations, with limited influence from the no-evaporation-and-sublimation simulations. This shows that the influence of INP on the March case's cloud field morphology and boundary layer structure is mainly through precipitation evolution that acts as a sink of moisture from the cloud layer.

Figure 9 shows a similar analysis for the October case. Compared to the cloud field in the March case above, the CAO cloud field in the October case is more heterogeneous, and the cumulus clouds begin to show even at the western boundary

of the sub-domain (Appendix G). Perturbing both $N_d$ and $S_{INP}$ now have strong influences on the cloud field. Despite the dependence of cloud properties on $E_{HM}$ in the October case discussed above (Figure 7), there are limited effects on the cloud field development from perturbing $E_{HM}$.

With a higher $S_{INP}$, the October CAO cloud field also has an earlier transition to cumulus-dominated region, a more heterogeneous cloud field across all of the CAO domain, and a higher surface precipitation rate (shown in Figure 7k). An earlier transition to cumulus-dominated region is also seen with low $N_d$ in the October case here. Both high $S_{INP}$ and low $N_d$ simulations experience earlier and more intense precipitation at the early stage of the CAO cloud shown in Figure 7, and this is consistent with the precipitation-induced SCT mechanism in CAO clouds from Abel et al. (2017). Such influence of low $N_d$ on

SCT is not seen in the March case, as there is limited influence of changing $N_d$ on precipitation due to the very limited amount of liquid cloud in the control simulation. Not that the stratocumulus-dominated region in the October case is very limited here due to the cumulus starts to show up at very early stage of the sub-domain.

Several other factors may influence SCT during CAO events but they were not examined in this study. These include sea surface temperature (SST), which can impact convection and turbulence, as well as boundary layer stability, inversion layer

strength and humidity. Additionally, this work focused on only three cloud microphysics parameters, whereas other microphysical processes could also potentially affect SCT. Future research should aim to explore the influence of these additional factors to gain a more comprehensive understanding of factors controlling SCT in CAO events.

The regions identified here are used for the overall comparison of cloud properties in the sub-domain as well as in different regions between the March and the October case in the next section.

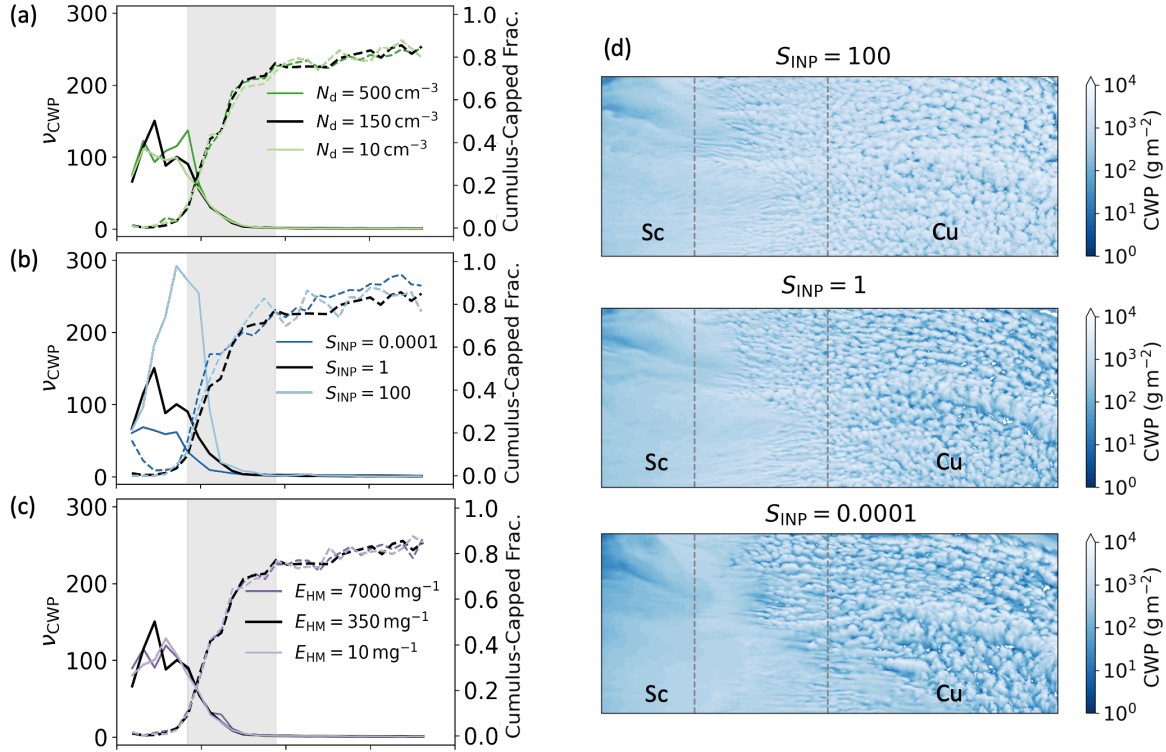

**Figure 8.** The cloud field homogeneity parameter ($\nu$, solid lines) and the fraction of cumulus-capped boundary layer (dashed lines) in cloudy pixels (cloud cover >= 20%) for simulations with perturbed $N_d$ (a), $S_{INP}$ (b) and $E_{HM}$ (c) on 15 March 2022, and (d) shows the 2D fields of cloud water path for simulations with perturbed $S_{INP}$. Grey-shaded areas in (a)-(c) as well as the region between grey dashed lines in (d) are the general stratocumulus to cumulus transition regions selected using both the cloud field homogeneity parameter ($\nu$) and the fraction of cumulus-topped boundary layer. Note that we did not define the regions for different simulations individually to avoid any location effects. The stratocumulus-dominated region is determined to be located before the sharp decrease of the cloud homogeneity parameter and before the sharp increase of cumulus-capped boundary layer fraction; the cumulus-dominated region is determined to be where both the cloud homogeneity parameter and the cumulus-capped boundary layer fraction become stable; and the rest is determined as the transition region.

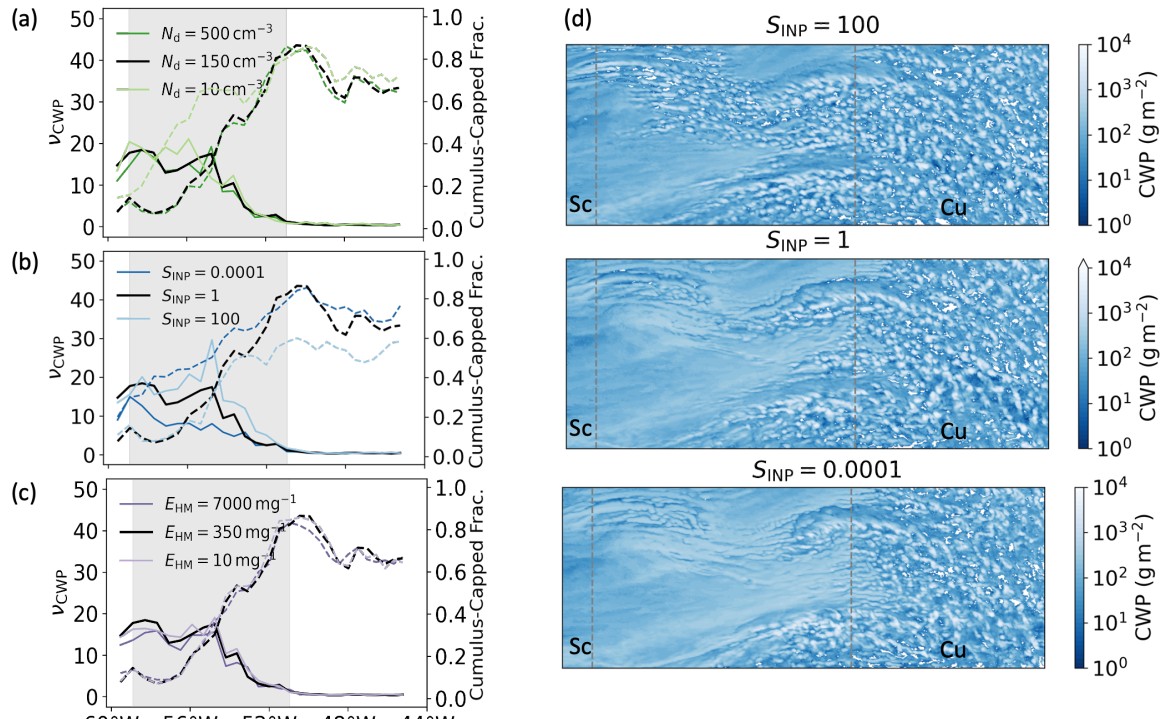

**Figure 9.** The cloud field homogeneity parameter ($\nu$, solid lines) and the fraction of cumulus-capped boundary layer (dashed lines) in cloudy pixels (cloud cover >= 20%) for simulations with perturbed $N_d$ (a), $S_{INP}$ (b) and $E_{HM}$ (c) on 24 October 2022, and (d) shows the 2D fields of cloud water path for simulations with perturbed $S_{INP}$. Grey-shaded areas in (a)-(d) as well as the region between grey dashed lines are the general stratocumulus to cumulus transition regions selected using both the cloud field homogeneity parameter ($\nu$) and the fraction of cumulus-topped boundary layer. Note that we did not define the regions for different simulations individually to avoid the location effect. The stratocumulus-dominated region is determined as before the sharp decrease of the cloud homogeneity parameter and the sharp increase of cumulus-capped boundary layer fraction, the cumulus-dominated region is determined by the trend of cloud homogeneity parameter and the overall cumulus-capped boundary layer fraction becoming stable, and the rest is determined as the transition region.

## 3.4 Overall comparison of the cloud responses to perturbed parameters between the cases

In this section, the responses of cloud properties are compared in terms of the fractional change relative to in the default simulation (Figure 10). We compared the fractional changes between the stratocumulus-dominated region (Figure 10a), cumulus-dominated region (Figure 10b), and the overall domain that includes the previous two regions and the transition region (Figure 10c). Mean cloud profiles (grid-box with in-cloud total water content $> 10^{-6}$ kg kg$^{-1}$) for the stratocumulus-dominated region and the cumulus-dominated region for each case are shown in Figure 11 and Figure 12 to illustrate the responses of in-cloud properties to the perturbed $S_{INP}$. The responses of in-cloud properties to $N_d$ and $E_{HM}$ are shown in Appendix I and will not be further discussed in detail here.

The strengths of the cloud responses to high $S_{INP}$ and low $S_{INP}$ are different in the two cases in the stratocumulus-dominated regions. Low $S_{INP}$ has the strongest effect in March, while high $S_{INP}$ has the strongest effect in October. This is because the March control simulation has low liquid water (Figure 11b and c) to be further removed when $S_{INP}$ increases , while the October control simulation has a very high liquid fraction (Figure 12b and c), so a high $S_{INP}$ and can strongly convert the liquid to ice with subsequent ice removal through accretion. This is similar when considering the influence of low $S_{INP}$ for the two cases.

Responses of cloud cover to $S_{INP}$ perturbations are opposite in these two cases in the cumulus-dominated region: a high $S_{INP}$ results in a higher cloud cover in March, but a lower one in October. The response in October is similar to the previous studies, hence not further discussed here. The higher cloud cover in March from a higher $S_{INP}$ is the result of a slower snow autoconversion and smaller ice hydrometeor size (Figure 11j) for lower fallspeed hence a lower precipitation rate. As there is very limited liquid in the control simulation already, and the dominant precipitation type (the main way to remove cloud water) in March is snow. Therefore, the impact of having more ice to remove more liquid is very limited in March, and instead, we see a similar influence from having more INPs in precipitating mixed-phase clouds to the one from having more CCNs in precipitation liquid-only clouds (the Albrecht effect). Such response is also expected to see in cirrus clouds with more INPs. This also explains why there is no such influence in the stratocumulus-dominated region as the precipitation rate is very low.

The influence of $S_{INP}$ on in-cloud LWP in the March cumulus-dominated region is also suppressed when compared to the one in the stratocumulus-dominated region. This is due to the liquid water being rapidly removed during SCT as shown in Figure 6h. We also find lower $S_{INP}$ leads to higher LWP but lower albedo at the top of the atmosphere in the March cumulus-dominated region. This is the result of the compensation between a slightly increased LWC (Figure 11g) near cloud top and a decrease in the albedo of ice through increasing ice size (Figure 11j) - a "Twomey-like" effect from INPs.

Most responses of the cloud properties to $N_d$ in both cases and all regions have the same direction (same sign of the fractional changes), but the influence of perturbing $N_d$ on the October CAO clouds are much stronger. This can also be explained by the different liquid-ice partitioning in control simulations of the two cases, with the October case has a much higher in-cloud LWP.

The strongest influence of perturbing $E_{HM}$ on cloud proprieties is in the cumulus-dominated region of the October case. There are limited effects of perturbing $E_{HM}$ in March and this is because the ambient temperature for the March case is too cold and outside of the HM active range (11a and f) . In the cumulus-dominated region of the October case, both high and low $E_{HM}$ simulations result in a higher reflected radiation, but the underlying reasons are different. For the simulation with

high $E_{HM}$, the higher albedo comes from a higher cloud cover, while for the simulation with low $E_{HM}$, it comes from a higher in-cloud LWP.

The responses of cloud properties in the overall domain are not just determined by the responses in stratocumulus- and cumulus-dominated regions, but also the size of these two regions and the SCT region. As the selected domains of CAO clouds in two cases both have bigger cumulus-dominated regions than the stratocumulus-dominated regions, the overall responses of cloud properties shown here are more similar to the ones in cumulus-dominated regions, with some influences from the stratocumulus-dominated regions.

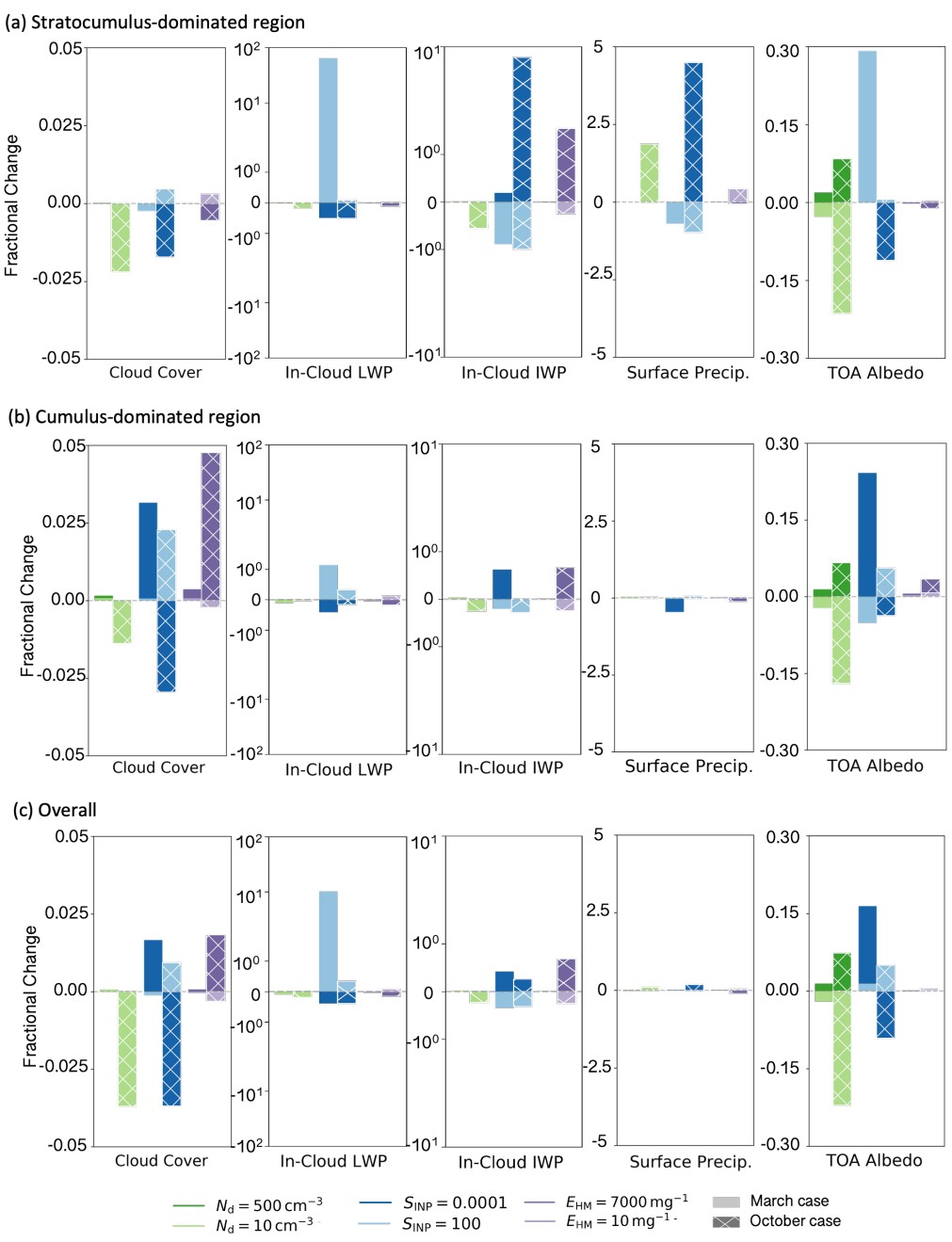

**Figure 10.** Fractional changes of cloud properties in the perturbed parameter simulations relative to the control simulations for the 15 March 2022 (solid shading) and the 24 October 2022 (hatched shading) cases, separated into the (a) stratocumulus-dominated domain, (b) cumulus-dominated domain and (c) overall domain. Cross-section means of the sub-domain are shown in Figures 6 and 7, and the determination of stratocumulus- and cumulus-dominated domains are discussed in Section 3.3. Note that the fractional change in the sub-domain is not just influenced by the fraction changes in stratocumulus-dominated and cumulus-dominated regions, but also the proportion of each region.

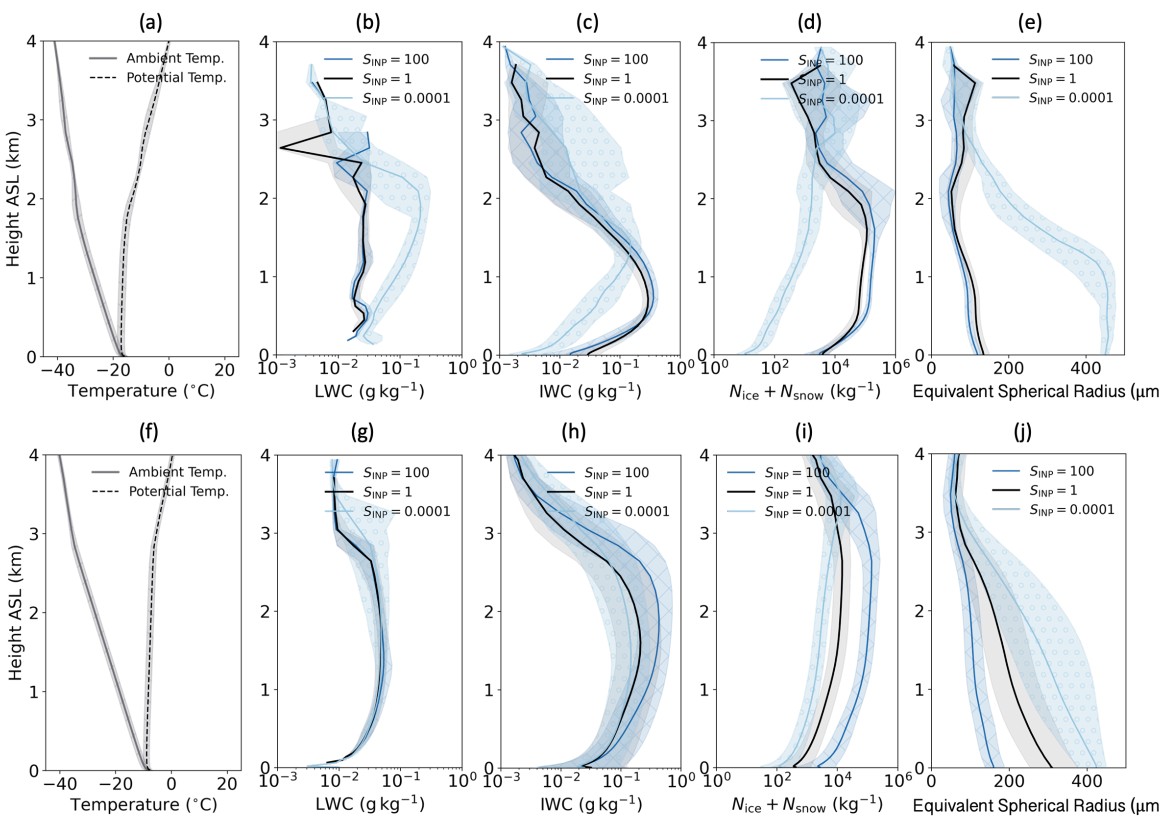

**Figure 11.** Vertical profiles for in-cloud properties in the March case: ambient temperature (default configuration), potential temperature (default configuration), in-cloud liquid water content (LWC), in-cloud ice water content (IWC), $N_{ice} + N_{snow}$ and equivalent spherical radius for stratocumulus-dominated (a-e) and cumulus-dominated (f-j) regions in the 15 March 2022 CAO case with different $S_{INP}$. The solid lines are medians, and the shaded areas were values between 25% quantile and 75% quantile. For the cloud properties plots, grids with lower than $10^{-6}$ kg kg$^{-1}$ total water content are removed. For hydrometeor number concentrations, cloudy grids with lower than 1 m$^{-3}$ are removed.

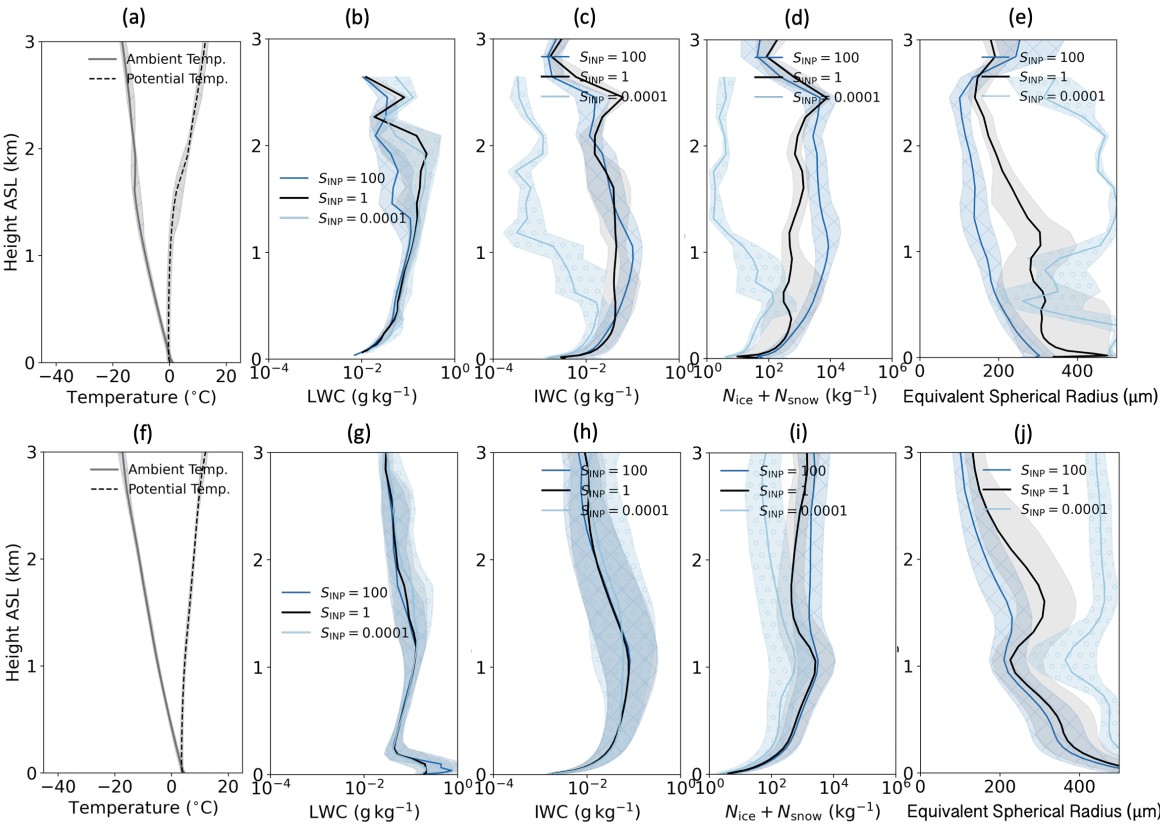

**Figure 12.** Vertical profiles for in-cloud properties in the October case: ambient temperature (default configuration), potential temperature (default configuration), in-cloud liquid water content (LWC), in-cloud ice water content (IWC), $N_{ice} + N_{snow}$ and equivalent spherical radius for stratocumulus-dominated (a-e) and cumulus-dominated (f-j) regions in the 24 October 2022 CAO case with different $S_{INP}$. The solid lines are medians, and the shaded areas were values between 25% quantile and 75% quantile. For the cloud properties plots, grids with lower than $10^{-6}$ kg kg$^{-1}$ total water content are removed. For hydrometeor number concentrations, cloudy grids with lower than 1 m$^{-3}$ are removed.

## 3.5 Comparison with satellite retrievals

In this section, we explore the extent to which the changes in INP, droplet number and secondary ice production alter the comparison with multiple satellite-retrieved cloud properties (all-sky LWP, top-of-atmosphere shortwave flux, and top-of-atmosphere longwave flux) in Figure 13 for March and Figure 14 for October. The satellite retrieval time and the selected corresponding model output time are concluded in Appendix A. Figures shown in the main text only include model output with different $S_{INP}$, comparison to simulations with $N_d$ and $E_{HM}$ are shown in Figures J1 and J2 of Appendix J. Cloud fractions retrieved from MODIS were also used for model-observation comparison shown in Figures J3 and J4. As the cloud fractions from MODIS were calculated by the percentage of cloudy pixel from finer resolution in a 5 km resolution pixel, we created a cloud mask from our model (1.5 km grid spacing) using 20% cloud cover as a threshold and then derive the cloud fraction using the percentage of cloudy grids in each corresponding MODIS cloud fraction grid.

For all the comparisons, our model output was first regridded to the same spatial resolution as the satellite retrieval and we only focused on the data within the sub-domain as shown in Figure 4(a) and Figure 4(c) for model-observation comparison. The normalized frequency and cross-section mean (+/- 1 standard deviation) are used for the comparison. MODIS-retrieved LWP is not used for quantitative comparison here due to its high mixed-phase cloud bias (Khanal and Wang, 2018).

In the March case, the control simulation shows reasonably good agreements of SW and LW fluxes at the top-of-the-atmosphere compared with other sensitivity test simulations, however underestimates the all-sky LWP from approximately 54 °W to 46 °W (approx. 10 $\mathrm{g\,m^{-2}}$ lower for the domain-mean LWP compared to the AMSR-2 retrievals (Figure 13)). With low $S_{INP}$, a higher all-sky LWP is produced but leads to a very large overestimation of all-sky LWP at the beginning of the CAO cloud field. Small underestimation of SW flux and overestimation of LW flux are also seen near the eastern boundary of the sub-domain in the control simulation, which may due to the cloud cover and IWP in the control simulation being slightly lower than the observed.

Although our model underestimates the LWP, the LWP from AMSR-2 retrievals suggests that the liquid water was very small in this March case (domain mean around 17 $\mathrm{g m^{-2}}$). Clouds in the March case were dominated by ice with a high IWP (modelled domain mean around 632 $\mathrm{g m^{-2}}$) and the control simulation shows good agreement of IWC against CALIOP retrievals (Appendix D). Therefore, we suggest that our model agrees with the observations on the liquid-ice partitioning and that the CAO clouds in this March case were dominated by ice, on which we based our conclusions.

The simulation with a high $S_{INP}$ in the October case agrees with all satellite retrievals from approximately 56 °W to 46 °W(Figure 14), but strongly underestimates the LWP for the region from 60 °W to 56 °W and overestimates the LWP for the eastern end of the region. The simulations with default and low $S_{INP}$ reproduce the LWP for the region from 60 °W to 56 °W, but overestimates the LWP for the rest of the region. Such overestimation of LWP in the cumulus-dominated region for the control simulation may come from the overestimation of LWC and underestimation of IWC in Figure D5 when model output are compared to MPhase C323 measurements. Similar biases are seen for SW flux which may be the results of LWP bias from the simulations. The overestimation of SW flux can be reduced by using a low $N_d$ as shown in Figure J2 of Appendix J, but such change has limited influence on the all-sky LWP bias. Based on the INP measurements from the M-Phase aircraft campaign,

it is known that the measured INP concentrations in this October case are within the range of INP concentrations from default $S_{INP}$ and low $S_{INP}$, but both simulations show clear overestimation of LWP and SW flux here. This inconsistency may come from the fact that we are only doing a sensitivity test here instead of exploring the whole parameter space, missing the output from different combinations of the parameter values; there are other processes (e.g., mixing and other cloud microphysics processes) which are not investigated in this study; the INP concentrations are temperature dependent but not directly derived from the background aerosols, and potentially missing variations of INPs through CAO cloud development.

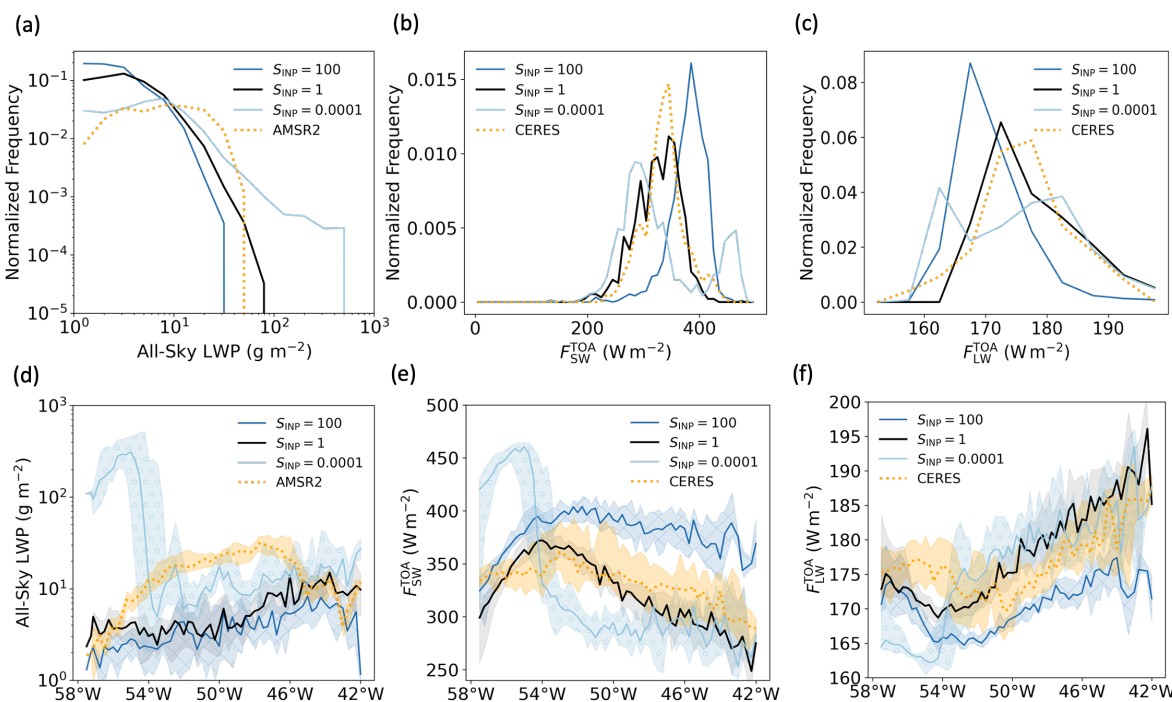

**Figure 13.** Model output compared with satellite retrievals of all-sky liquid water path (LWP) from AMSR-2, shortwave radiation and longwave radiation at the top-of-atmosphere from CERES for simulations with different $S_{INP}$ on 15 March 2022 : (a)-(c) are the normalized frequency, and (d)-(f) are the cross-section median and quantile comparisons. All the comparisons were done within the selected sub-domain with model output and satellite retrievals regridded to the same resolution. The times of model output were selected as the closest quarter to the satellite retrieval times.

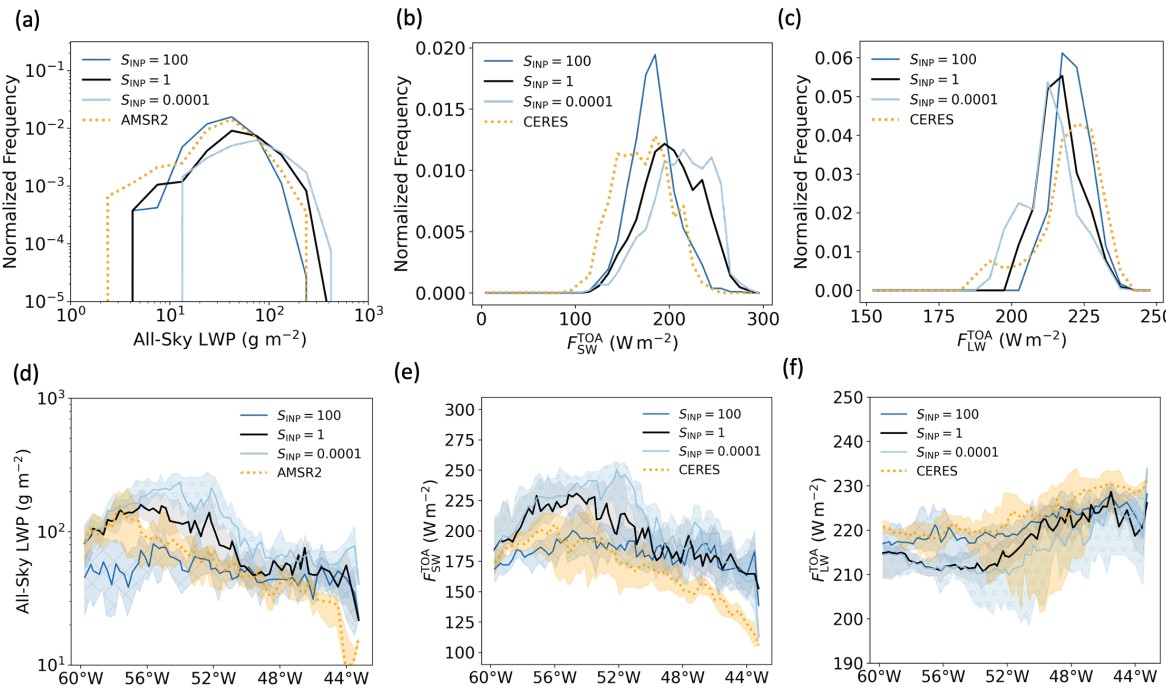

**Figure 14.** Model output compared with satellite retrievals of all-sky liquid water path (LWP) from AMSR-2, shortwave radiation and longwave radiation at the top-of-atmosphere from CERES for simulations with different $S_{INP}$ on 24 October 2022 : (a)-(c) are the normalized frequency, and (d)-(f) are the cross-section median and quantile comparisons. All the comparisons were done within the selected sub-domain with model output and satellite retrievals regridded to the same resolution. The times of model output were selected as the closest quarter to the satellite retrieval times.

## 4 Discussion and Conclusions

We illustrate in the Results section above that the responses of modelled CAO cloud properties to the perturbations of $N_{\mathrm{d}}$, $S_{\mathrm{INP}}$ and $E_{\mathrm{HM}}$ are different or even opposite in the two selected CAO cases over the Labrador Sea. Clouds in the October case respond similarly to increases in INP concentration (or ice concentration) compared to previous studies, which is a reduction in reflected SW flux and albedo at the top-of-the-atmosphere (Vergara-Temprado et al., 2018) and an earlier transition from stratocumulus to cumulus (Abel et al., 2017; Tornow et al., 2021). However, the March case differs strongly from the existing literature. We explain this difference in behaviour by categorising the March case as an ice-dominated regime and the October case as a liquid-dominated regime (Figure 15).

Cloud temperatures are very different in the March and the October CAO cases, with the mixed-phase CAO clouds in March being in a much colder environment (approximately 15 °C to 20 °C lower). Such temperature difference leads to a strong difference in primary ice production through INP as the INP concentration increases exponentially with decreasing temperature using the same parameterization (approximately 2 orders of magnitude for 20 °C difference). A higher primary ice production (colder cloud temperatures) means a greater portion of the condensed cloud water is converted to ice, resulting in a lower liquid water fraction in March, and vice versa for lower primary ice production (warmer cloud temperatures) in October.

The March CAO event is in an ice-dominated regime with a low liquid water fraction. In such an ice-dominated regime, increasing INP concentration leads to a higher number concentration of ice ($N_{\mathrm{ice}}$), slows down the snow autoconversion rate and reduces the ice hydrometeor size and fallspeed, which then reduces the precipitation and restricts the removal of cloud water. This is more obvious in the cumulus-dominated region as it experiences stronger precipitation than the stratocumulus-dominated region. Such influence leads to higher cloud cover and IWP in March, and consequently a higher TOA albedo and SW flux, which is further enhanced by the higher single-scattering albedo from high $N_{\mathrm{ice}}$ (Twomey-like effect). These behaviours are similar to the aerosol first (Twomey, 1977) and second (Albrecht, 1989) indirect effects on liquid clouds through changes in cloud condensation nuclei concentrations, but in this March case acting through INP concentrations. As the clouds are dominated by ice, there is also very little water available for liquid-phase processes, therefore changes in $N_{\mathrm{d}}$ have only a small influence on the cloud. Furthermore, because the cloud temperatures are cold (approx. -15 °C to -35 °C for cloudy grids) and the Hallet-Mossop process is assumed to occur only in the temperature range from -2.5 to -7.5 °C, changing $E_{\mathrm{HM}}$ has only a small influence on the clouds.

The response of TOA albedo to increased $S_{\mathrm{INP}}$ simulation in March identifies a possible new mechanism of negative cloud-phase feedback if INP concentrations increase in the future, besides the original three mechanisms suggested in Murray et al. (2021). For ice-dominated cloud with increasing INP concentrations from the warming climate, it will respond to the higher INP concentration in a similar radiative responses that seen in liquid clouds when CCN increases (i.e., an INP-driven first and second indirect effect), leading to a higher SW flux at the top of the atmosphere and negative cloud-phase feedback, competing with the effect of warming these cloud systems.

With a low INP concentrations in the March CAO event, different responses are seen in stratocumulus-dominated and cumulus-dominated regions. In the stratocumulus-dominated region, a higher in-cloud LWP and a lower surface precipitation with no obvious change in cloud cover result in a higher reflected radiation. In the cumulus-dominated region, as the cloud liquid is rapidly removed during SCT, the increase of in-cloud LWP from low INP concentration is therefore very limited. Instead, lower IWP and lower ice albedo from the "Twomey-like" effect result in a lower SW reflection, compensating the limited increase of LWP.

Contrary to the March case, the warmer October case is in a liquid-dominated regime with a high liquid water fraction in general (apart from the end of the CAO cloud system). In this liquid-dominated regime, increases in INP concentration lead to higher $N_{ice}$ and therefore a higher collection of liquid water from ice hydrometeors, and consequently more precipitation, stronger removal of cloud water and lower cloud cover, opposite to the March case. Together with lower LWP, increasing INP concentration in such a liquid-dominated regime leads to a lower SW flux at the top-of-atmosphere. As the liquid water fraction is high, there is also a strong influence from changing $N_d$ and consequently a larger effect on the SW flux from changes in liquid water than in March. Because the temperatures are relatively warm, more clouds are in the active temperature range for the HM process and there is enough liquid water available for riming, hence we see a strong influence from changing $E_{HM}$.

The occurrence of liquid- and ice-dominated clouds, which controls their response to INP, $N_d$ and $E_{HM}$, is not controlled only by temperature, but also by the ambient INP concentration: for example, the cloud is liquid-dominated at the beginning of the March CAO cloud system in the low $S_{INP}$ simulation. This suggests that there could be an interaction between INP concentrations and other cloud properties, such as a strong effect of $N_d$ on cloud properties at very low INP concentrations but a weak dependence at high INP concentrations when the cloud is ice-dominated. This illustrates one of the limitations of this study since we have only explored the effects of individual parameters. A full PPE (perturbed parameter ensemble) that explores potential co-variations in inputs and interactive effects would be needed to explore this further. In addition, we only compared two cases and their environmental conditions in this study, and a more robust and systematic investigation of the influence of environmental conditions can also be carried out using a PPE in which environmental/initial conditions are varied using idealised simulations.

Other secondary ice production (SIP) mechanisms are not included in our model when this work is conducted, while the non-included SIP mechanisms such as droplet shattering and ice-ice collision have been shown important to SCT in CAO events (Karalis et al., 2022). These two SIP mechanisms can take place at a colder temperature than the existing HM process, which could have some impacts on the CAO cloud properties and responses in the cold March case (e.g., higher ice number concentrations and smaller ice hydrometeor sizes). Future modelling work will include other SIP mechanisms when these become available in the model. In addition, fixed in-cloud $N_d$ was used in this study for the easier interpretation of sensitivity test results, but such a setup can lead to the neglect of potential feedbacks between cloud and aerosols/CCNs. Future work will also include aerosol-derived $N_d$ and cloud processing of aerosols where possible for a better representation of aerosol-cloud interactions.

In general, this comparative sensitivity study reveals different or even opposite responses of the CAO cloud properties to aerosols including CCNs (through changing $N_d$) and INPs and SIP (the Hallet-Mossop process) when the cloud temperatures

are different by comparing two CAO events over the Labrador Sea. The main findings and conclusions drawn from this study are shown below.

1. Cloud temperature and INP concentrations control the liquid-ice partitioning in the control simulations and thereby affect their responses to the perturbed parameters. The two cases have different liquid-ice partitioning and hence are categorized into ice-dominated (the cold March case) and liquid-dominated (the warm October case) regimes.

2. In the liquid-dominated, warm October case, increasing INP concentration leads to lower cloud cover and in-cloud LWP, hence a lower albedo, consistent with findings from previous studies.

3. In the ice-dominated, cold March case, increasing INP concentration leads to higher cloud cover and in-cloud IWP, hence resulting in a higher albedo and SW flux at the top-of-atmosphere. Such response is more prominent in the cumulus clouds. This influence of increasing INP concentration is opposite to the one in the liquid-dominated October case, and is potentially a new mechanism of negative cloud-phase feedback in ice-dominated CAO clouds if INP concentrations increase in the future due to the warming climate, additional to the original three cloud-phase feedback mechanisms suggested by Murray et al. (2021).

4. Stronger influences from changing $N_{\rm d}$ and the Hallet-Mossop efficiency are seen in the liquid-dominated October case as more liquid is available and cloud temperature in October spans the HM active temperature range (-2.5 to -7.5 °C).

Future work with a full exploration of the parameter space (including other SIP mechanisms), systematically perturbed environmental conditions or other important cloud microphysics parameters will be beneficial to our understanding and modelling of these mixed-phase CAO clouds and their responses to the warming climate.

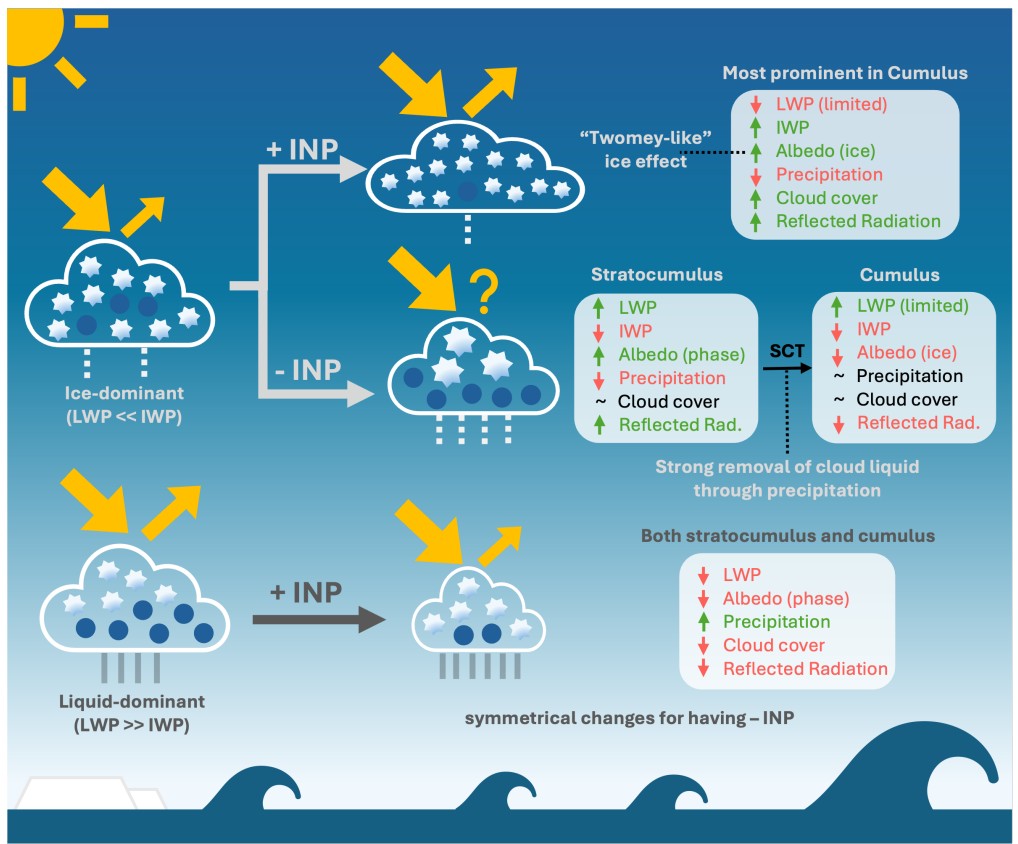

**Figure 15.** A schematic diagram for explaining the different sensitivities of CAO cloud properties to perturbations of INPs in ice-dominant and liquid-dominant clouds.

*Data availability.* The satellite data products used in this study include: MODIS Level 1B Calibrated Radiances Product (Collection 6.1) onboard the Aqua satellite for RGB composites with bands 1,3 and 4 (https://ladsweb.modaps.eosdis.nasa.gov/missions-and-measurements/ products/MYD021KM) with geolocation data from the Geolocation 1km (https://ladsweb.modaps.eosdis.nasa.gov/missions-and-measurements/ products/MYD03); MODIS Atmosphere Level 2 Cloud Production (Collection 6.1) onboard the Aqua satellite for cloud water path, cloud cover and cloud-top temperature (https://ladsweb.modaps.eosdis.nasa.gov/missions-and-measurements/products/MYD06_L2); CERES SSF Level 2 product (Edition 4A) onboard the Aqua satellite for TOA shortwave flux and longwave flux (https://ceres.larc.nasa.gov/data/); columnar cloud liquid water (version 8.2) for all-sky liquid water path from AMSR-2 onboard GCOM-W (https://www.remss.com/missions/amsr/); Temperature from Level-2 1km Cloud Layer Data (version 4-51, https://asdc.larc.nasa.gov/project/CALIPSO/CAL_LID_L2_01kmCLay-Standard-V4-51_ V4-51) and ice water content from Level-2 5km Cloud Profile Data (version 4-51, https://asdc.larc.nasa.gov/project/CALIPSO/CAL_LID_ L2_05kmCPro-Standard-V4-51_V4-51) from CALIPSO. The ERA5 data used include the surface skin and surface pressure from ERA5 hourly data on single levels (https://cds.climate.copernicus.eu/datasets/reanalysis-era5-single-levels), and temperature and pressure at 800 hPa from ERA5 hourly data on pressure levels (https://cds.climate.copernicus.eu/datasets/reanalysis-era5-pressure-levels). The INP data

from the MPhase aircraft campaign were obtained from: https://zenodo.org/records/14781199. The measurements during the MPhase aircraft campaign can be obtained from https://catalogue.ceda.ac.uk/uuid/6d7971a92d154bb29af3167dfb6f5a7e/. The model data used for analysis can be found from https://zenodo.org/records/14536461.

**Appendix A:  Retrieval time and selected model timepoint for satellite data used in this study**

| | | 15 March 2022 | | 24 October 2022 | |
|---|---|---|---|---|---|
| Instrument | Satellite Products | Retrieval time (UTC) | Selected model timepoint (UTC) | Retrieval time (UTC) | Selected model timepoint (UTC) |
| AMSR-2 | LWP | 16:48 | 16:45 | 16:18 | 16:15 |
| CERES | SW,LW Fluxes | 16:45 | 16:45 | 17:00 | 17:00 |
| MODIS | Band1,3,4, CTT, CWP | 15:15 | 15:15 | 17:00 | 17:00 |
| CALIOP | IWC, temperature | 16:13 | 16:15 | N/A | N/A |

**Table A1.** The satellite products used in this study, their retrieval times and selected model time points for comparison in the sub-domains.

LWP: liquid water path, SW: shortwave, LW: longwave, CTT: cloud top temperature, CWP: cloud water path, IWC: ice water content.

## Appendix B: Monthly distribution of cloud top temperature over the Labrador Sea in 2022

Monthly distribution of cloud top temperature of low-level (cloud top pressure > 700 hPa) and mixed-phase (cloud top temper-
ature ranging from -40 °C to 0 °C) CAO cloud over the Labrador Sea in 2022 are shown here in Figure B1.

Daily-mean CAO index at 800 hPa ($M_{800}$) (Kolstad and Bracegirdle, 2008; Fletcher et al., 2016a), which is the difference
between potential temperature at surface skin ($\theta_{skin}$) and potential temperature at 800 hPa ($\theta_{800}$): $M_{800} = \theta_{skin} - \theta_{800}$, was
calculated using the ERA5 dataset (Hersbach et al., 2020). Grids with $M_{800} > 0$ K, which is a compulsory condition for CAO
identification and an indicator of unstable atmosphere, were defined as CAO grids. We use $\theta_{800}$ in this study because more
high-latitude CAOs can be identified by using $\theta_{800}$ compared with using $\theta_{700}$ (potential temperature at 700 hPa) (Fletcher
et al., 2016a). Cloud top pressure (CTP) and cloud top temperature (CTH) from MODIS were used to filter low-level (CTP >
700 hPa), mixed-phase (-40 °C < CTP < 0°C) clouds.

CTTs of low-level, mixed-phase CAO clouds in January, February and March are generally colder with CTT peaks between
-20 °C to -25 °C, while the ones in October, November and December are warmer with CTT peaks between -10 °C and
-15 °C. Other months were not included due to low density of CAO events in those time of the year. The cold March case
in this study located near the colder end of the CTT climatology while the warm October case located near the warmer end,
providing contrasting CTT conditions for this sensitivity test and a study range covering most of the range from the shown
CTT climatology.

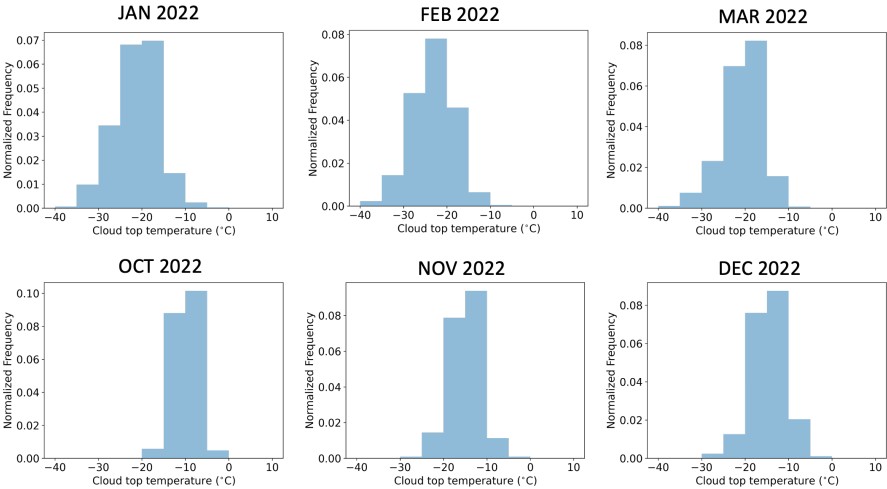

**Figure B1.** Monthly distribution for cloud top temperature of low-level, mixed-phase CAO clouds over the Labrador Sea in January, February,
March, October, November and December 2022. Clouds with cloud top pressure smaller than 700 hPa and cloud top temperature warmer
than 0 °C and colder than -40 °C were excluded.

## Appendix C:  Maps for the nested model domains and the sub-domains for analysis

Figure C1 shows the nested model domains and the sub-domains for analysis of both cases. The sub-domains were chosen to be away from the boundaries of the nested model domains to avoid the boundary effects. In both cases, due to the winds and airmasses from north-west direction, the fields generated from the coarser global model require a few hours to spin-up once entering the nested model domains.

The timescales for spinning up the boundary layer structure in the nested model domain is $\frac{L}{\sigma_w}$, where $L$ is the boundary
layer depth ( 1000 m at the beginning of the CAO events) and $\sigma_w$ is the standard deviation of the w component of wind. For the March case, the mean $\sigma_w$ between the western boundaries of the nested model domain and the sub-domain is around 0.83 ms$^{-1}$, which requires about 20 minutes to spin up. For the October case, the mean $\sigma_w$ between the northwestern ends of the nested model domain and the sub-domain is around 0.39 ms$^{-1}$, which requires about 45 minutes to spin up.

We examined whether the boundary effects can reach the sub-domains by using simple calculations of how long it takes the
airmasses to reach the western boundary of each sub-domain. The distances were estimated based on the mean direction of wind. For the March case, the distance between the middle of the western boundary of the nested model domain to the middle of the western boundary of the sub-domain is around 430 km. The mean wind speed from surface to 2 km height above sea level is 13.0 ms$^{-1}$ (46.8 kmhr$^{-1}$). Therefore, it takes around 9 hours for the airmasses to travel in the March case. For the October case, the distance between the north-west point of the nested model domain to the middle of the western boundary
of the sub-domain is around 500 km. The mean wind speed from surface to 2 km height above sea level is 16.5 ms$^{-1}$ (59.4 kmhr$^{-1}$), so it takes around 8.5 hours for the airmasses to travel in the October case. Some airmass travelling into northwestern part of the sub-domain may have less time to spin up and be affected by the boundary effects, however, we choose to keep this part of the sub-domain for capturing more earlier stage CAO clouds as the clouds broke up into cumulus clouds very soon in the October case.

For the whole sub-domain in the March case and most of the sub-domain (except the small northwestern part) of the October case, the time and distance required for the airmasses to reach the sub-domains are sufficient to avoid the boundary effects propagate into the sub-domain.

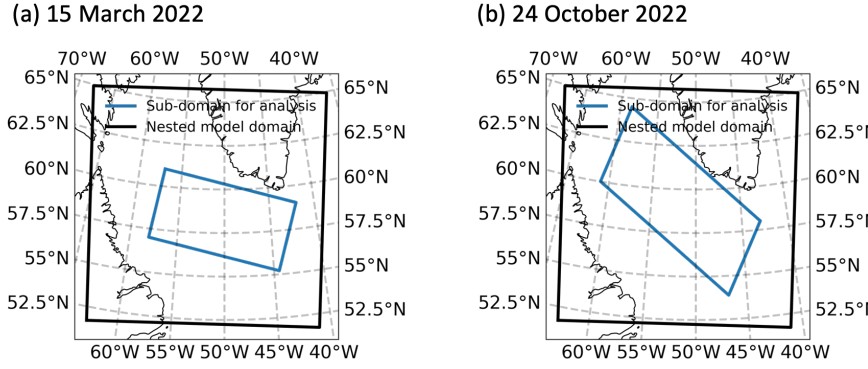

**Figure C1.** Nested model domains (black) and the selected sub-domains for analysis (blue) for (a) 15 March 2022, (b) 24 October 2022.

**Appendix D: Model-observation comparison for control simulations**

In this appendix, we validate our control simulations using satellite retrievals for the March case (as there is no aircraft campaign for this case) shown in Figures D1, D2 and D3, and MPhase aircraft measurements for the October case shown in Figure D5 from two flights (MPhase C322 and C323) with the flight tracks shown in Figure D4. The results of model-observation comparison for the March case are shown first, followed by the results for the October case.

**D1    15 March 2022**

For the March case studied in this work, we evaluate our model against cloud top temperature (CTT) from MODIS onboard the Aqua satellite, as well as temperature and ice water content (IWC) from CALIOP onboard the CALIPSO satellite (Level-2 1 km Cloud Layer Data for temperature and Level-2 5 km Cloud Layer Data for IWC, version 4-51).

Figure D1 shows the CTT from model output and MODIS retrievals. As satellite simulator was not available for regional simulations in our model, the extraction of modelled CTT relied on the definition of cloudy grids. Here we used two thresholds of grid-box mean TWC (total water content) for determining whether a grid in our model output is cloudy or non-cloudy. The threshold $10^{-5}$ kgkg$^{-1}$ has been used before for comparing model water content with *in-situ* aircraft measurements (e.g., Abel et al. (2017)), and the threshold $10^{-4}$ kgkg$^{-1}$ was selected for a stricter definition of cloudy grids as passive satellite retrievals can be less sensitive compared to aircraft measurements. As there is no high clouds in the domain of interest, the cloud top for each column of model grids is defined as the highest grid passing the selected threshold of TWC. The MODIS data (1 km) were regridded to model resolution (1.5 km) using the nearest-neighbour method.

The cloud top height (CTH) retrieved from MODIS is around 5 km for low-level clouds (not shown), which is unreasonable and therefore not further used for validating CTH in the control simulation. This may because that the MODIS CTH over the sea level is not directly retrieved but calculated using observed SST (sea surface temperature), CTT, and a zonal mean lapse rate (Platnick et al., 2016), but the CTT in the March case is much colder compared to climatology.

The distributions of CTTs within the sub-domain shown in Figures C1(a),(b) and (c) are compared in Figure D2. Using a threshold of $10^{-5}$ kgkg$^{-1}$ resulted in a much colder CTT (peak near -40 $^\circ$C) compared to the MODIS-retrieved CTT (peak near -30 $^\circ$C). Using a threshold of $10^{-4}$ kgkg$^{-1}$ reduces the cold bias strongly and leads to a similar peak temperature of CTT (near -30 $^\circ$C) to the one from MODIS. However, small cold bias still exists with the modelled CTT using the $10^{-4}$ kgkg$^{-1}$ threshold having higher frequency at colder temperatures and lower frequency at warmed temperatures compared to satellite retrieved CTT. Note that the grid-to-grid bias was not calculated here due to potential double-penalty problem when compared model data with observations.

Temperature and IWC profiles from CALIOP onboard CALIPSO were also used to evaluate the control simulation of the March case. Figure D3 shows the selected CALIPSO track (a), comparison of the temperature and IWC distributions (b,c) as well as the along-track profiles (d-g). The modelled profile was extracted by finding the closest model grid for each footprint from the CALIPSO data. The CALIPSO temperature has a higher resolution (1 km) and therefore was regridded to the model resolution (1.5 km), while the modelled IWC has a higher resolution compared to the CALIPSO IWC resolution (5 km)

therefore the modelled IWC was regridded to the CALIPSO IWC resolution. Due to the nature of lidar attenuation for clouds at lower levels, grids which had no CALIPSO data or model data were excluded when comparing the distributions and plotting the profiles.

There is also a small cold bias for the temperature but the overall temperature distribution from our model agrees well with the one from CALIPSO. This is also shown for the whole profile statistics: the median temperature from the modelled profile is -30.9 °C (interquartile range (IQR): -34.3 °C to -25.8 °C) and the median temperature from the CALIPSO profile is -30.2 °C (IRQ: -33.7 °C to -25.1 °C).

The distributions of IWC from model and CALIPSO have similar peak IWC values. However, the modelled IWC has higher frequencies both at the low (0.01 to 0.02 $\text{gm}^{-3}$) and high (0.5 to 1 $\text{gm}^{-3}$) ends of the IWC bins, and lower frequencies for the IWC bins in the middle. Based on the profile statistics, the modelled slightly overestimates the IWC (median: 0.19 $\text{gm}^{-3}$, IQR:0.08 $\text{gm}^{-3}$ to 0.39 $\text{gm}^{-3}$) compared to the CALIPSO IWC (median: 0.17 $\text{gm}^{-3}$, IQR:0.08 $\text{gm}^{-3}$ to 0.33 $\text{gm}^{-3}$). However, such bias (0.02 $\text{gm}^{-3}$) is considered low and acceptable.

To summarize the model-observation comparison above for the March case, our modelled CTTs are slightly colder compared to the MODIS-retrieved CTTs when using the $10^{-4}$ $\text{kgkg}^{-1}$ threshold. Similarly, the temperature comparison between modelled profile and CALIPSO profile shows that our model has small cold bias of the temperature (-0.7 °C), but such bias is acceptable and suggests temperatures from our model are reasonably comparable with the satellite retrievals. Our model overestimates the IWC by 0.02 $\text{gm}^{-3}$ compared to the CALIPSO IWC, but such bias is also relatively small and acceptable, suggesting the control simulation can also reproduce the IWC reasonably well.

### D2 24 October 2022

For the October case, we evaluate the control simulations by comparing the temperature, grid-box mean TWC (total water content), liquid water fraction (LWC / TWC), grid-box mean LWC (liquid water content) and grid-box mean IWC (ice water content) with the aircraft measurements from two flights (C322 and C323) during the MPhase aircraft campaign. The flight tracks and time-series plots of temperature and height for each flight are shown in Figure D4. Results of comparison for the variables mentioned above are shown in Figure D5. The water content measurements were from Nevzorov probes onboard the aircraft.

Preprocessing of the aircraft data (1 Hz resolution) and the model data was performed for a like-for-like comparison. Each point of the observation data was assigned to the nearest model grid (longitude,latitude and altitude) and the mean of all the observed data points within each model grid was calculated for the later comparison with model data. The model data were extracted using the flight track with grids within 10.5 km around each point of the flight tracks also included. Considering the issues including a model may not simulate the same cloud at a same location as the real world, and the aircraft measurements may not be representative enough for comparison with grid-box mean value from the model, we did not calculate the grid-by-grid bias directly, but compared them using the composited vertical profile of each flight.

For the model-observation comparison with measurements from MPhase C322, where the dominant type of clouds are stratocumulus clouds, our modelled temperature are colder than the observed temperature above 1000 m (Figure D5(a)) and

has a higher inversion layer. This may be the reason why that the locations of the clouds simulated from our control simulation are higher compared to the observed clouds (Figure D5(b)). Before the calculation of bias, we shifted the modelled clouds lower to match the peaks of the observed cloud profile and modelled cloud profile. The control simulation slightly overestimates the TWC, LWC and IWC with the profile mean bias of 0.048 $\mathrm{gm}^{-3}$, 0.047 $\mathrm{gm}^{-3}$ and 0.0003 $\mathrm{gm}^{-3}$, but is consistent with the observations on the liquid water fraction that most of the clouds were dominated by liquid. Some ice-free conditions were

not simulated in our model (Figure D5(e)), but such bias needs further model-observation comparison with more cases to investigate and is beyond the scope of this work.

     For the model-observation comparison with measurements from MPhase C323, where the dominant type of clouds were cumulus clouds, our model shows good agreement of the temperature profile with the observed one (Figure D5(g)). Our model did not simulate some high clouds (around 2500 m), however these clouds were very thin with the majority of their TWC

around 0.001 $\mathrm{gm}^{-3}$ (Figure D5(i)). Our model reproduced the grid-box mean TWC with a profile mean bias of 0.004 $\mathrm{gm}^{-3}$. The LWC is overestimated with a bias of 0.018 $\mathrm{gm}^{-3}$, while the IWC is underestimated with a bias of -0.003 $\mathrm{gm}^{-3}$, resulting in higher liquid water fraction in our model. For clouds ranging from 1500 m to 2000 m, our model captured the TWC but making too much liquid and too little ice, resulting a much higher liquid water fraction. The overestimation of LWC and liquid water fraction here may be the reason for the overestimation of all-sky LWP shown in Figure 14 in the October case.

To summarize the model-observation comparison of the October case, in the C322 region where stratocumulus clouds dominated, the heights of our modelled clouds were higher for around 500 m, potentially due to the cold bias of temperature above 1000 m and a higher modelled inversion layer. The grid-box mean TWC, LWC and liquid water fraction generally agreed with the observation if the modelled clouds shifted to the heights matching with the observed clouds. The modelled IWC biased high compared to the observed IWC, but remained as a very small part of the cloud water similar to the observation. In the

C323 region where cumulus clouds dominated, our model captured the temperature profile but missed some thin (TWC around 0.001 $\mathrm{gm}^{-3}$), high (around 2500 m) clouds. For clouds ranging from 1500 m to 2000 m, our model overestimates the LWC and underestimates the IWC, this is also seen for clouds below 1500 m but with much smaller bias, which can lead to the overestimation of all-sky LWP shown in Figure 14 in the October case.

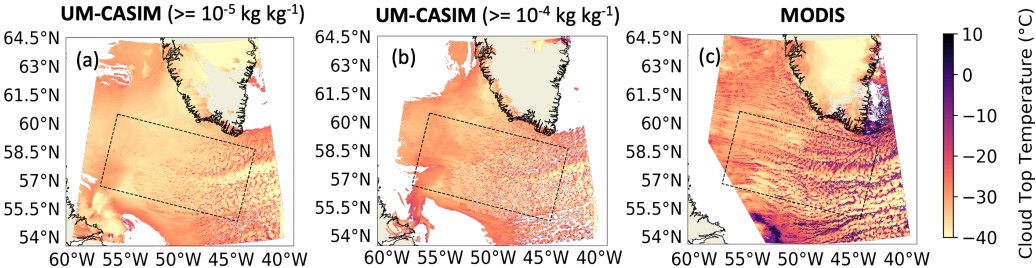

**Figure D1.** Cloud top temperature (CTT, a-c) from the control simulation of the March case and the MODIS onboard Aqua satellite. Two TWC (total water content) thresholds were used for determine whether a model grid is cloudy or not which are TWC >= $10^{-5}$ $\mathrm{kgkg}^{-1}$ (a) and TWC >= $10^{-4}$ $\mathrm{kgkg}^{-1}$ (b). Regions marked with grey dashed lines in (a) - (c) are the sub-domains (same as the sub-domains shown in the main content) for the comparison of CTT distributions in Figure D2.

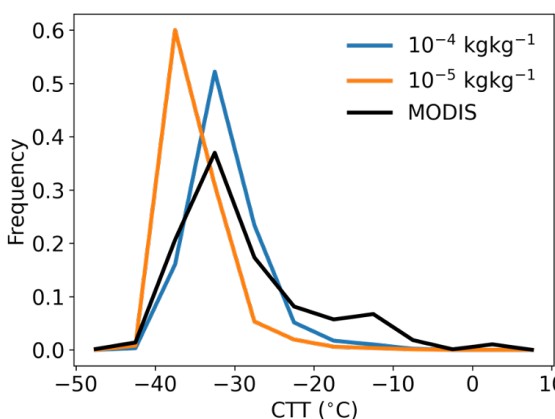

**Figure D2.** Comparison of the cloud top temperature (CTT) distributions of 15 March 2022 between the CTT retrieved from the MODIS onboard the Aqua satellite (black) and the CTT extracted from the control simulation of the March case using different total water content thresholds: TWC >= $10^{-5}$ kgkg$^{-1}$ (orange) and TWC >= $10^{-4}$ kgkg$^{-1}$ (blue).

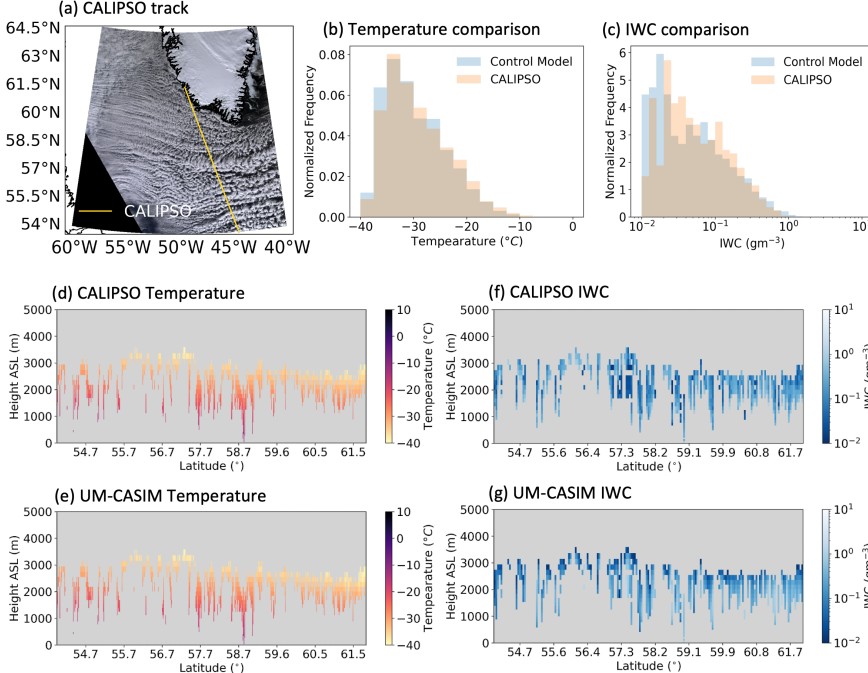

**Figure D3.** Comparison of temperature and IWC (ice water content) profiles from CALIOP onboard CALIPSO satellite and the control simulation of 15 March 2022: (a) the selected CALIPSO track, (b) distributions of temperature, (c) distributions of IWC, (d) regridded CALIPSO temperature profile. (f) CALIPSO IWC profile, (e) modelled temperature profile, (g) regridded modelled IWC profile. Note that for the comparison and individual profiles, grids without valid CALIOP data or model data are removed.

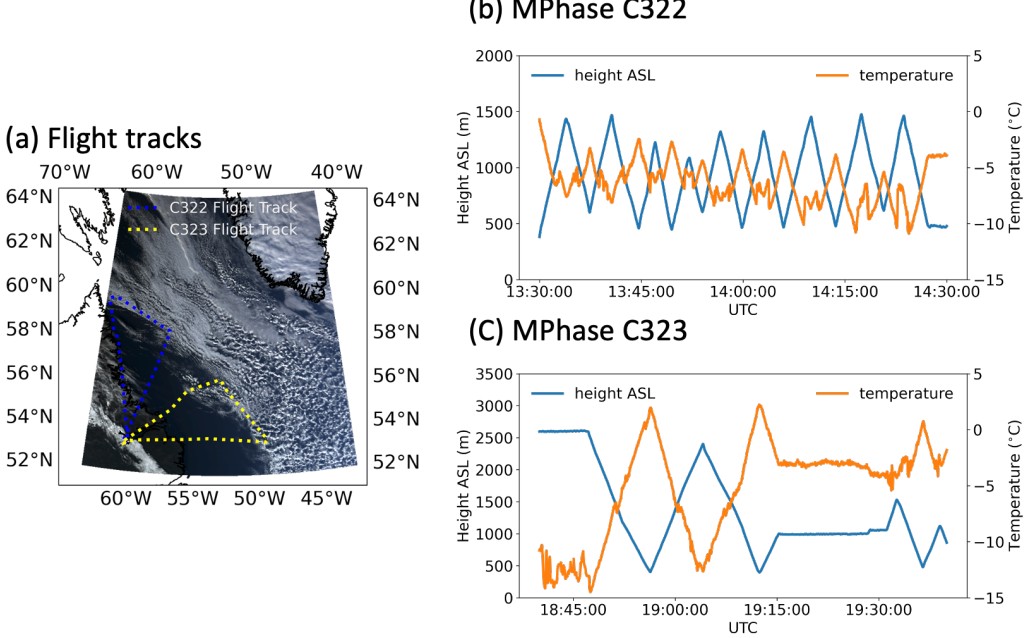

**Figure D4.** Information of the MPhase C322 and C323 flights used for the validation of the control simulation on 24 October 2022: (a) flight tracks, (b) height and temperature profiles for cloud measurements during MPhase C322, (c) height and temperature profiles for cloud measurements during MPhase C323.

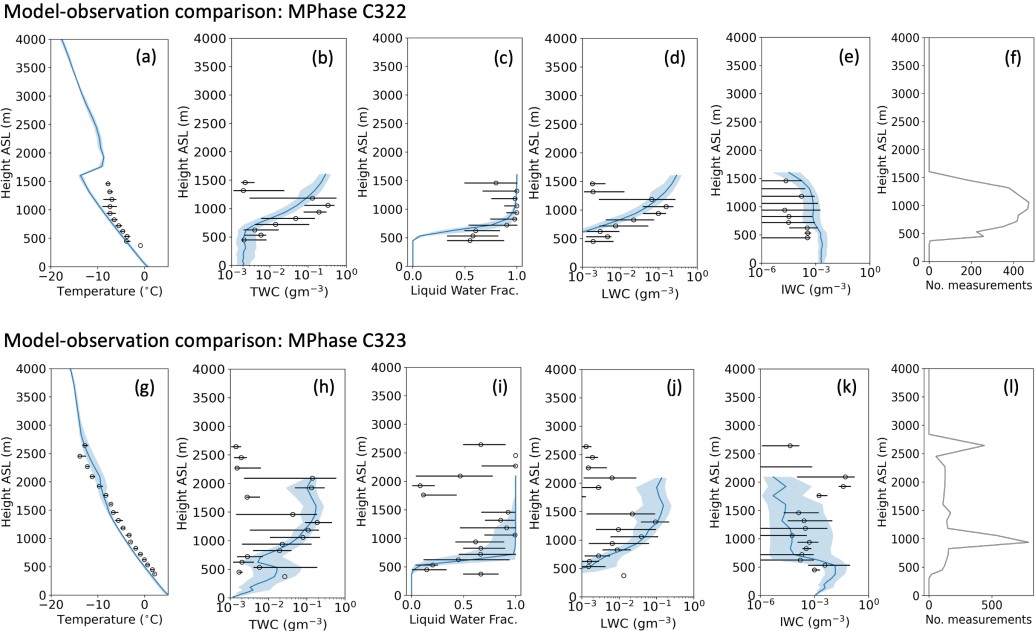

**Figure D5.** Model-observation comparison of the control simulation against MPhase C322 and C323 measurements on 24 October 2022: (a,g) temperature, (b,h) grid-box mean TWC (total water content), (c-i) liquid water fraction (LWC / TWC), (d-j) grid-box mean LWC (liquid water content), (e-k) grid-box mean IWC (ice water content). Figures (f) and (l) show the number of measurements from each flight. Model data are shown in blue lines (medians) with the IQR (interquartile) ranges coloured in light blue. Observation data are shown in black unfilled circles (medians) with the IQR ranges shown in black solid lines.

## Appendix E: Supplementary figures for Figure 5

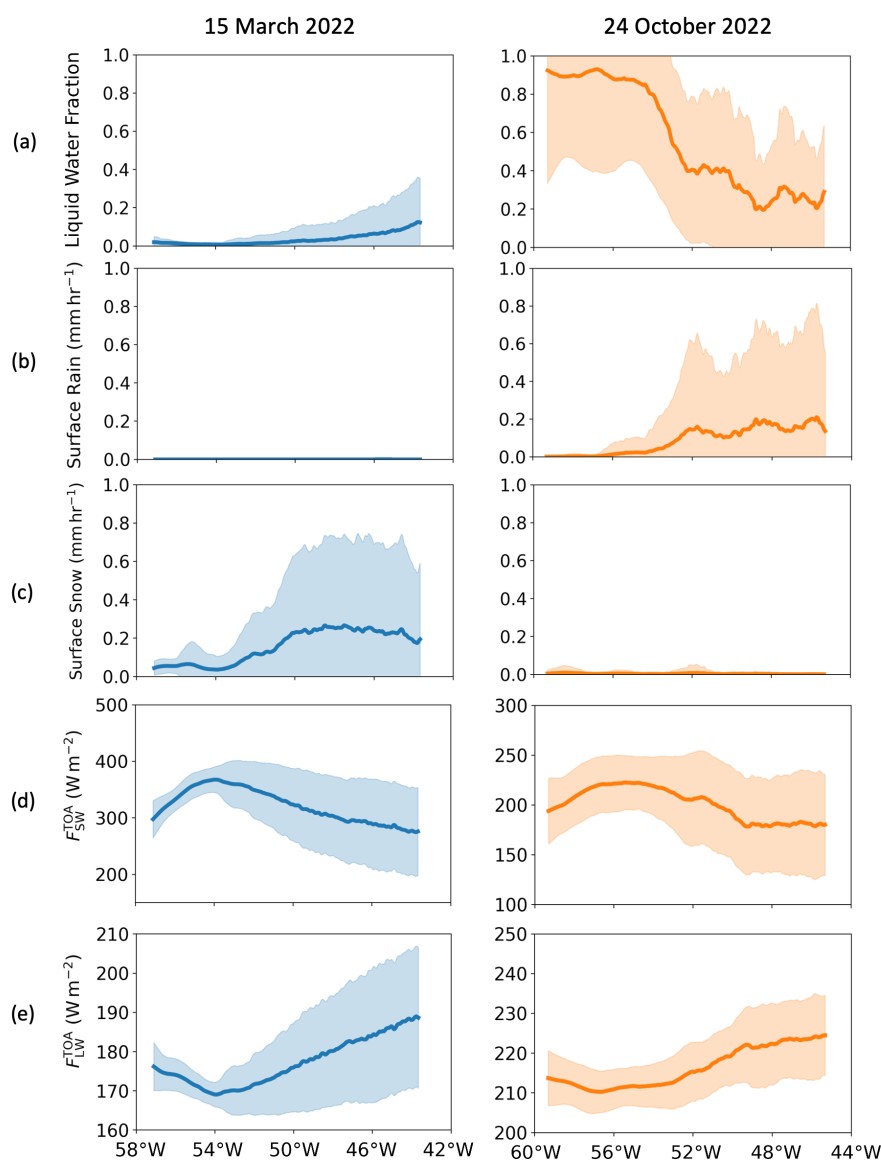

**Figure E1.** Cross-section mean cloud properties from the March (left panel) and the October (right panel) control simulations in the sub-domain: (a) liquid water fraction (LWP/CWP), (b) surface rain rate, (c) surface snow rate, (d) shortwave (SW) radiation flux at the top of atmosphere, and (e) longwave radiation (SW) flux at the top of atmosphere. The shaded area indicates the range of +/- 1 standard deviation. The time points selected are 16:45 UTC for the March case and 17:00 UTC for the October case, which are consistent with the corresponding CERES measurement times.

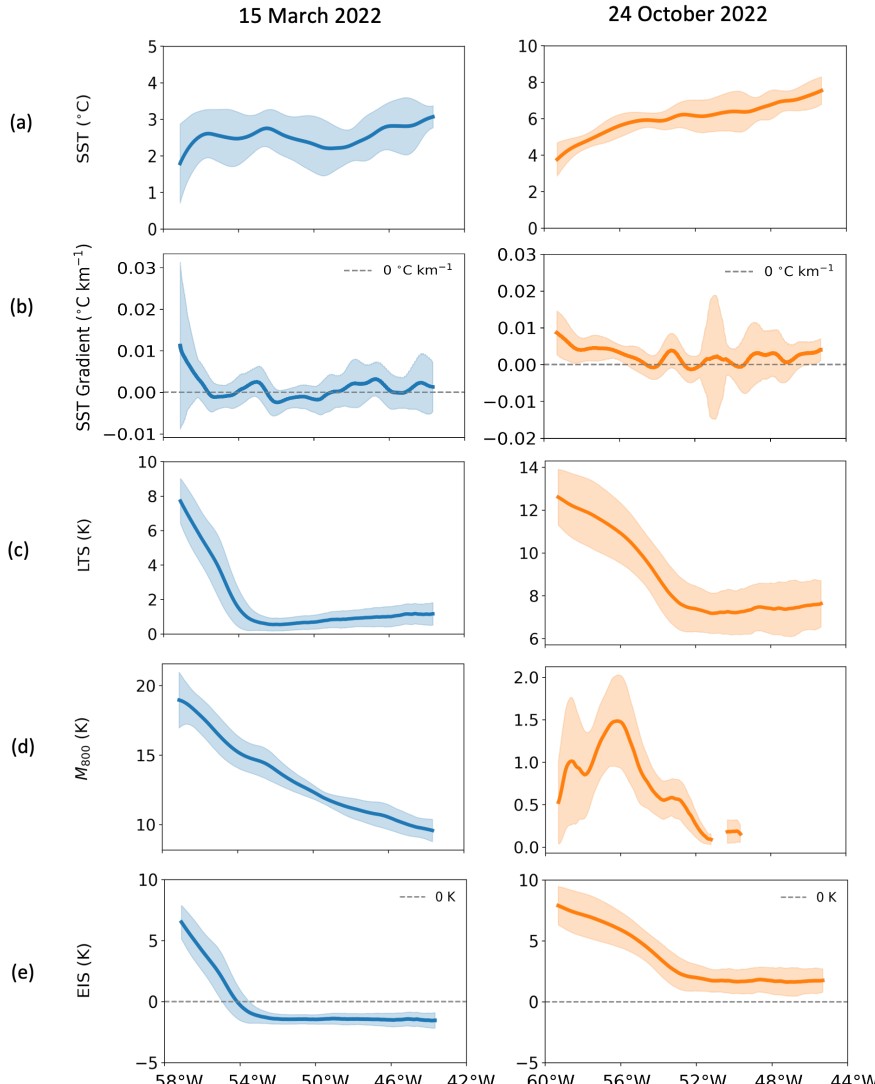

**Figure E2.** Cross-section mean cloud properties from the March (left panel) and the October (right panel) control simulations in the subdomain: (a) sea surface temperature (SST), (b) SST gradient, (c) lower tropospheric stability (LTS), (d) CAO index at 800 hPa ($M_{800}$), and (e) estimated inversion layer strength (EIS). The shaded area indicates the range of +/- 1 standard deviation. The time points selected are 16:45 UTC for the March case and 17:00 UTC for the October case, which are consistent with the corresponding CERES measurement times.

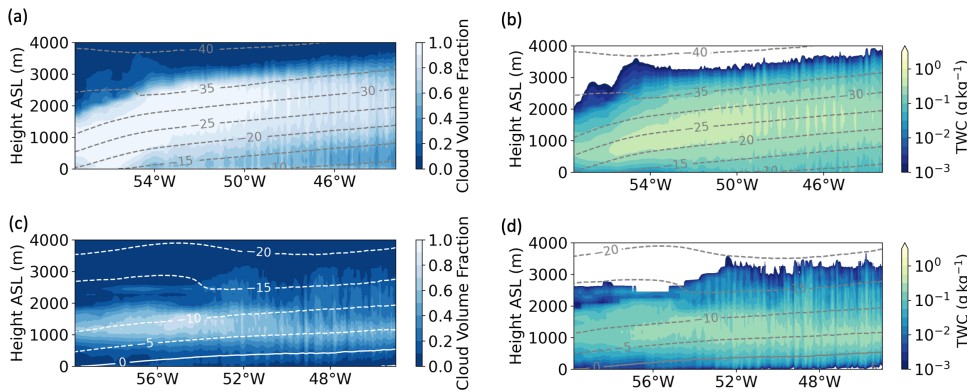

**Figure E3.** Cross-section mean vertical profiles from the sub-domain of bulk cloud volume fraction and in-cloud total water content (TWC) for the March case (a,b) and the October case (c,d) in the control simulations. Dashed lines in all figures are the ambient temperature contour lines in °C. To calculate the mean of the in-cloud total TWC, grid boxes with cloud volume fractions lower than 5% and total in-cloud TWC less than 1 gkg$^{-1}$ are excluded. The time points selected are 16:45 UTC for the March case and 17:00 UTC for the October case, which are consistent with the corresponding CERES measurement times.

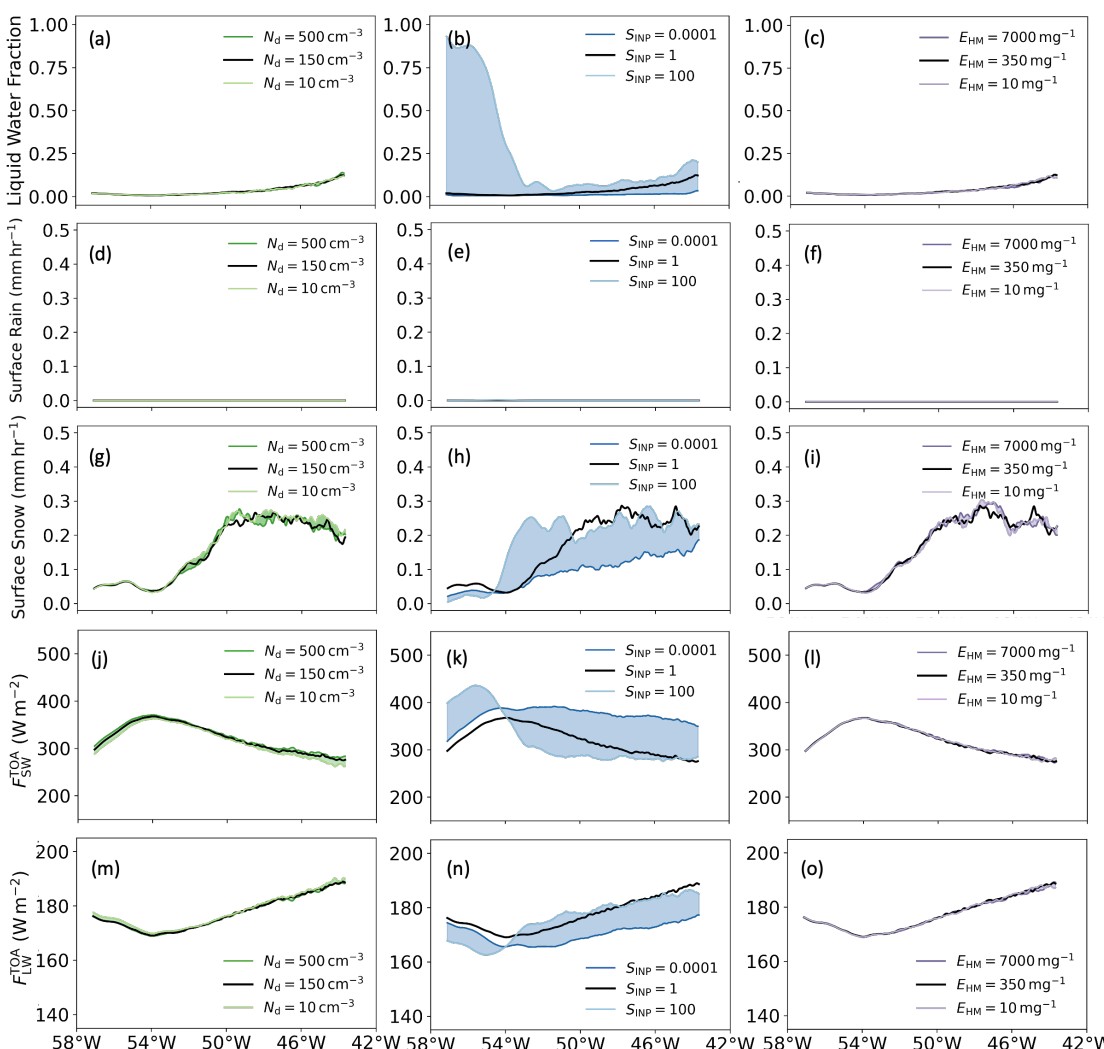

**Figure F1.** Sub-domain mean CAO cloud properties from the beginning of the cloud system to the end of the cloud system for 15 March 2022 at 16:45 UTC: (a)-(c) liquid water fraction (LWP/CWP), (d)-(f) surface rain, (g)-(i) surface snow, (j)-(l) shortwave radiation flux at the top of atmosphere, and (m)-(o) longwave radiation flux at the top of atmosphere. Grid boxes with cloud cover smaller than 20% were moved before averaging. 16:45 UTC was chosen for the relative CERES (onboard the Aqua satellite) measurement time.

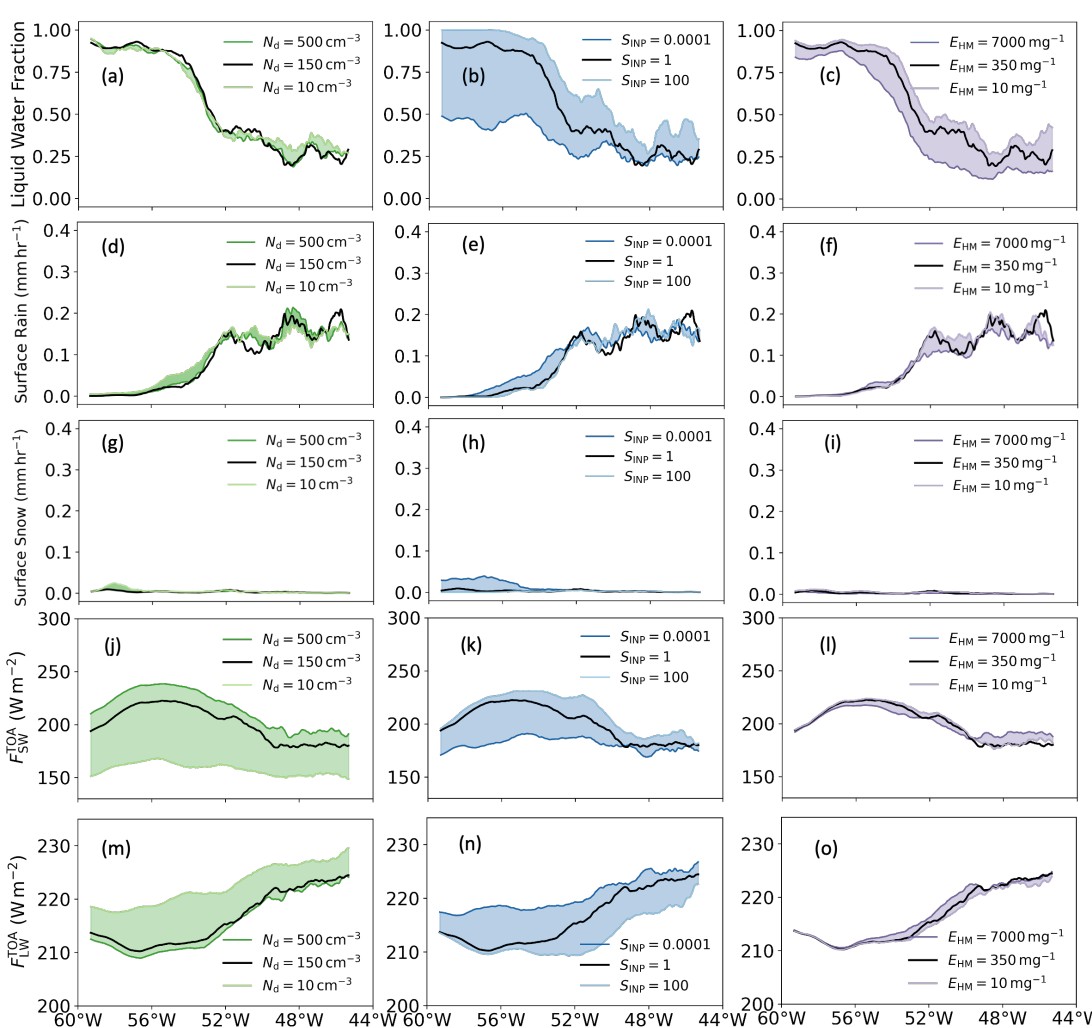

**Figure F2.** Sub-domain mean CAO cloud properties from the beginning of the cloud system to the end of the cloud system for 24 October 2022 at 17:00 UTC: (a)-(c) liquid water fraction (LWP/CWP), (d)-(f) surface rain, (g)-(i) surface snow, (j)-(i) shortwave radiation flux at the top of atmosphere, and (m)-(o) longwave radiation flux at the top of atmosphere. Grid boxes with cloud cover smaller than 20% were moved before averaging. 17:00 UTC was chosen for the relative CERES (onboard the Aqua satellite) measurement time.

## Appendix G:  Boundary layer type fractions for all the simulations

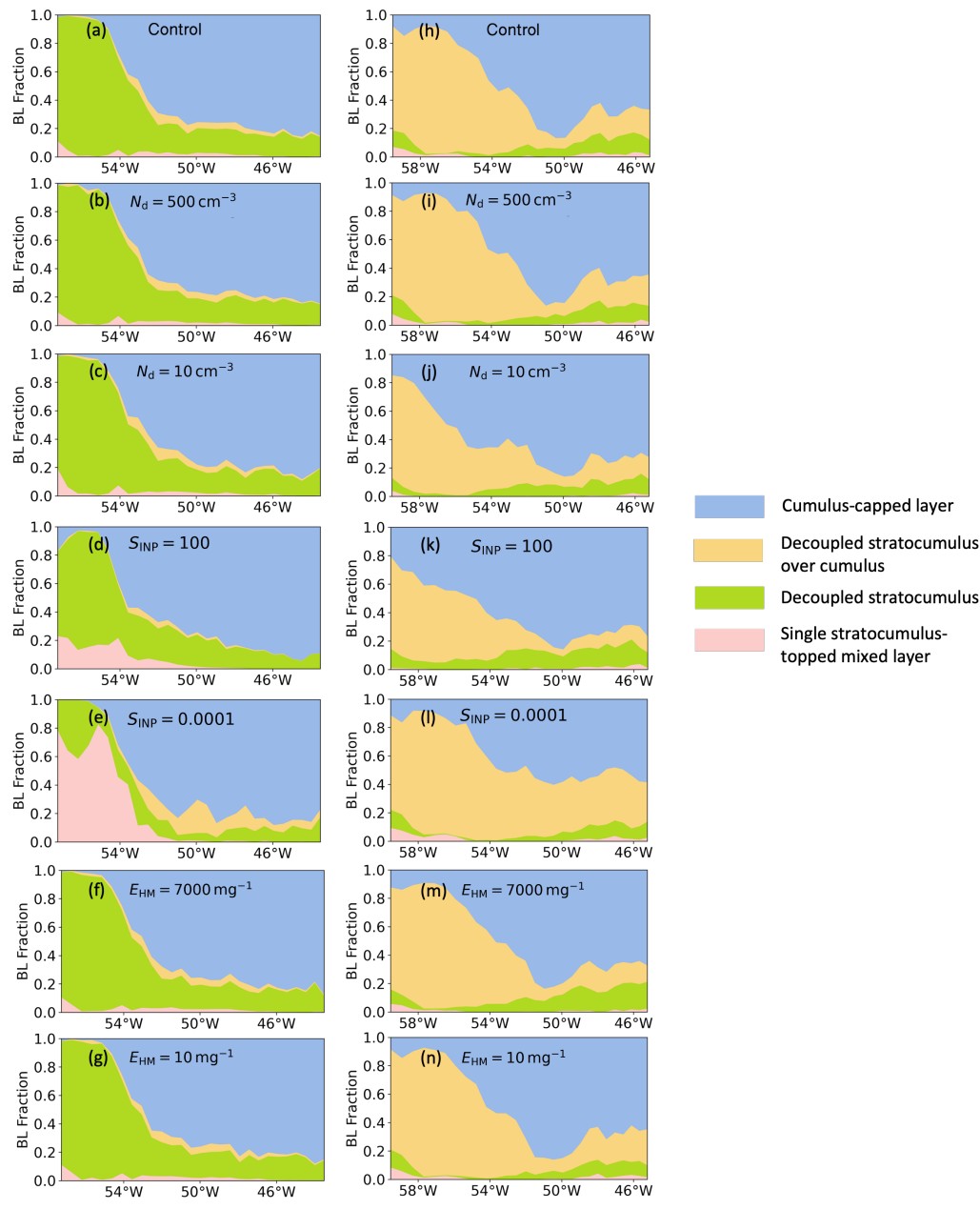

**Figure G1.** Boundary layer type fractions for all the simulations of 15 March 2022 (a-g) and 24 October 2022 (h-n). Four major boundary layer types are shown here: pink – single stratocumulus-topped mixed-payer, green - decoupled stratocumulus, orange – decoupled stratocumulus over cumulus, and blue – cumulus-capped layer. Only pixels with cloud cover higher than or equal to 20% are included for analysis.

# Appendix H: Results of model simulations with different precipitation, evaporation and sublimation setup for the March case

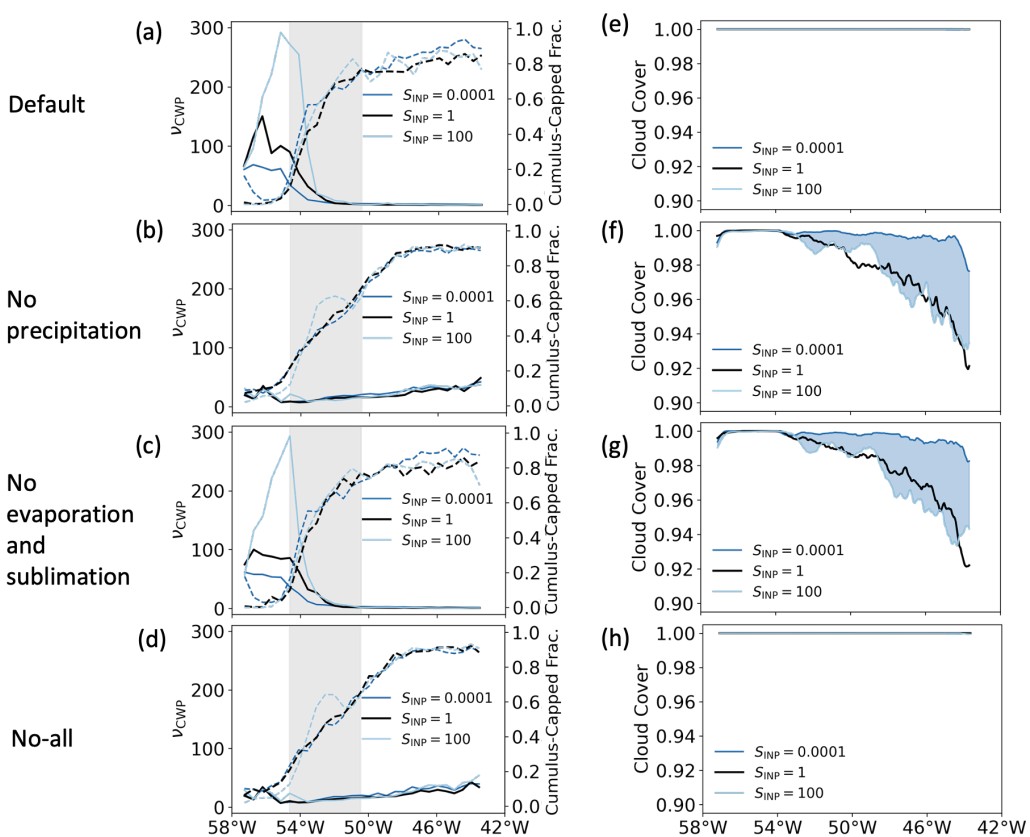

**Figure H1.** The cloud field homogeneity parameter ($\nu$, solid lines) with the fraction of cumulus-capped boundary layer (dashed lines) in cloudy pixels (cloud cover >= 20%) (a-d) and cloud cover (e-h) from simulations with perturbed $S_{INP}$ for (a,e) default setup, (b,f) no precipitation (sedimentation of all hydrometeors) setup, (c,g) no evaporation and sublimation setup, and (d) no-all (no precipitation, evaporation and sublimation) for the March case.

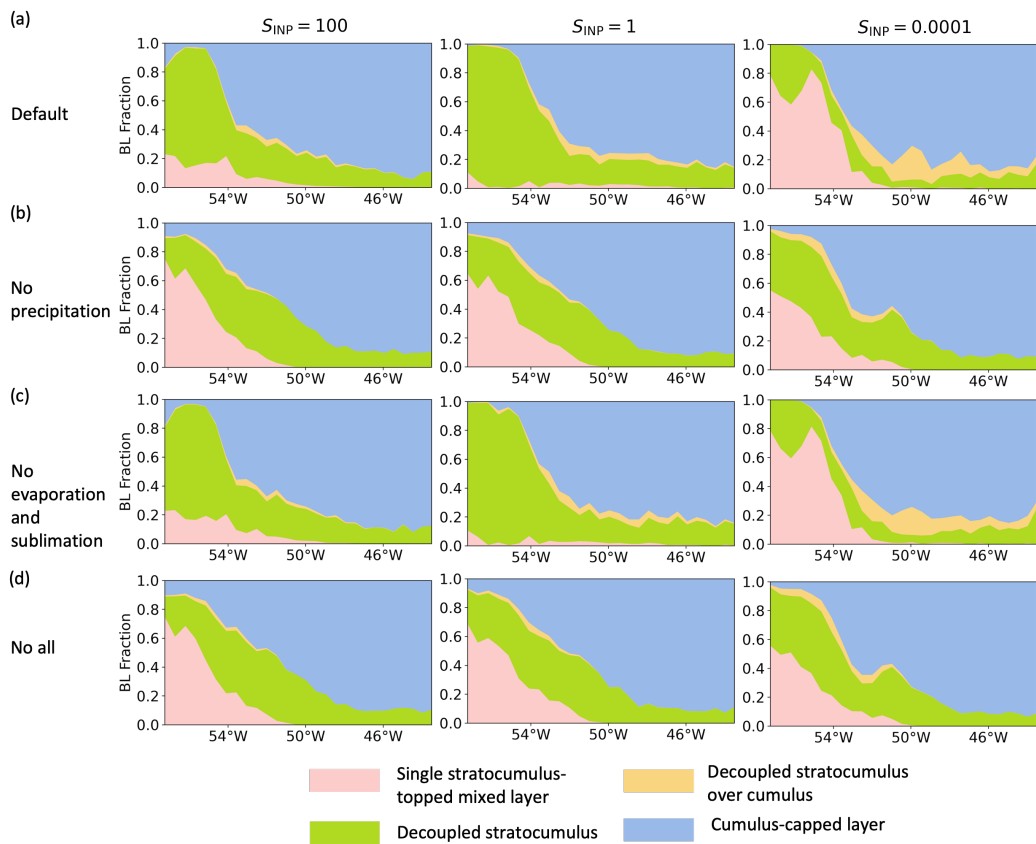

**Figure H2.** Boundary layer type fractions from simulations with perturbed $S_{INP}$ for (a) default setup, (b) no precipitation (sedimentation of all hydrometeors) setup, (c) no evaporation and sublimation setup, and (d) no-all (no precipitation, evaporation and sublimation) for March case.

## Appendix I: Vertical profiles of in-cloud properties for simulations with different $N_d$ and $E_{HM}$

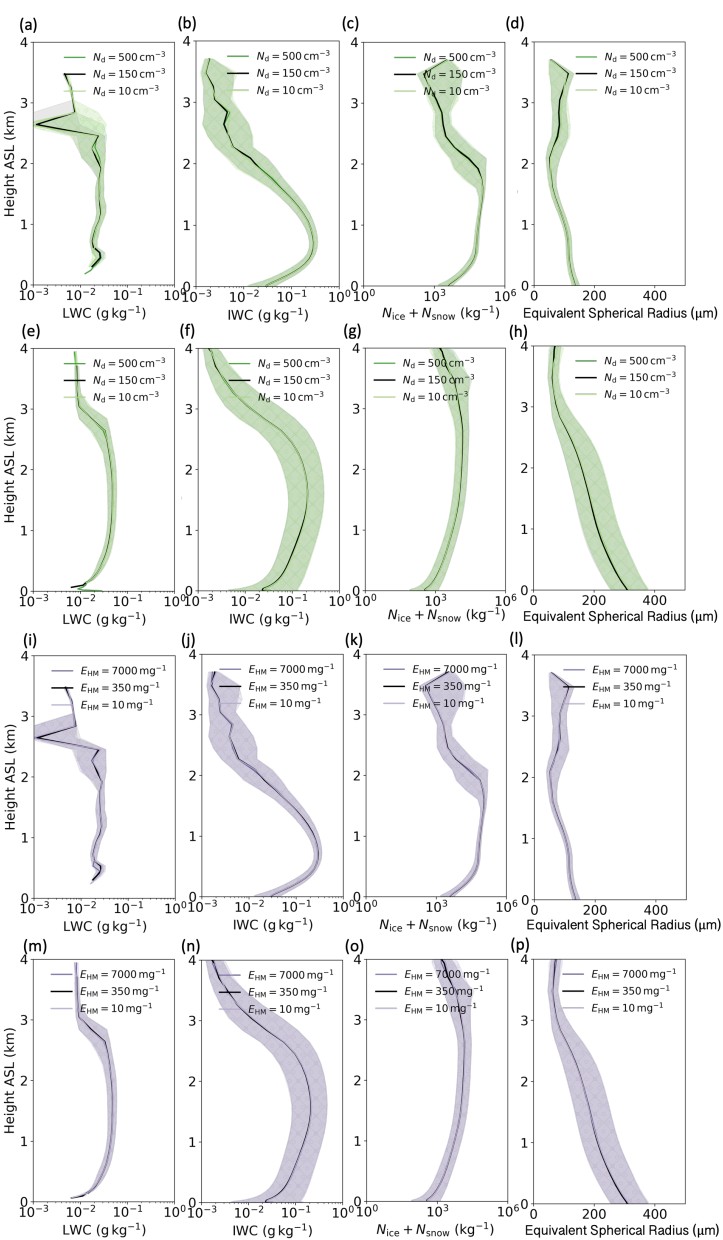

**Figure I1.** Vertical profiles of in-cloud LWC (liquid water content), in-cloud IWC (ice water content), $N_{ice} + N_{snow}$ and equivalent spherical radius for stratocumulus-dominated (a-d, i-l) and cumulus-dominated (e-h, m-p) regions in the 15 March 2022 CAO case with different $N_d$ (a-f) and $E_{HM}$ (g-i).

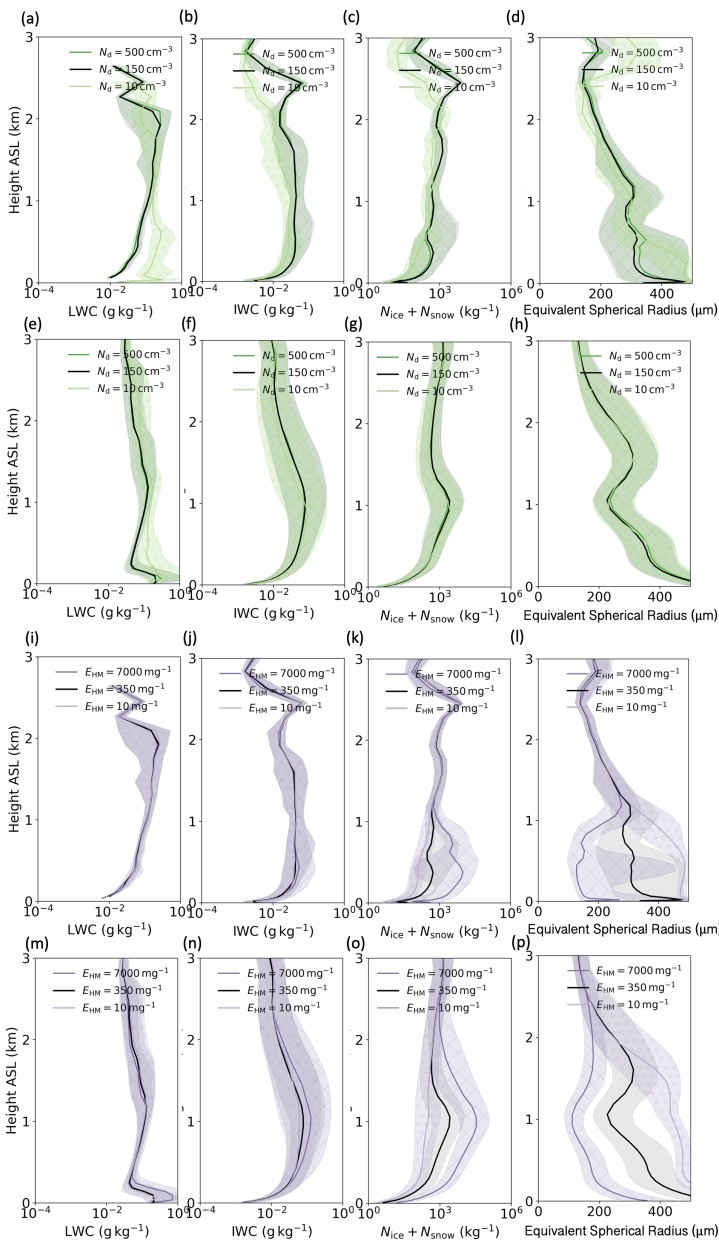

**Figure I2.** Vertical profiles of in-cloud LWC (liquid water content), in-cloud IWC (ice water content), $N_{ice} + N_{snow}$ and equivalent spherical radius for stratocumulus-dominated (a-d, i-l) and cumulus-dominated (e-h, m-p) regions in the 24 October 2022 CAO case with different $N_d$ (a-f) and $E_{HM}$ (g-i).

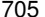

**Figure J1.** Model output compared with satellite retrievals for simulations with different $N_d$ (a-f) and $E_{HM}$ (g-l) on 15 March 2022 : the normalized frequency of all-sky liquid water path (LWP) from AMSR-2 (left column), shortwave radiation at the top-of-atmosphere (middle column) and longwave radiation at the top-of-atmosphere from CERES (right column).

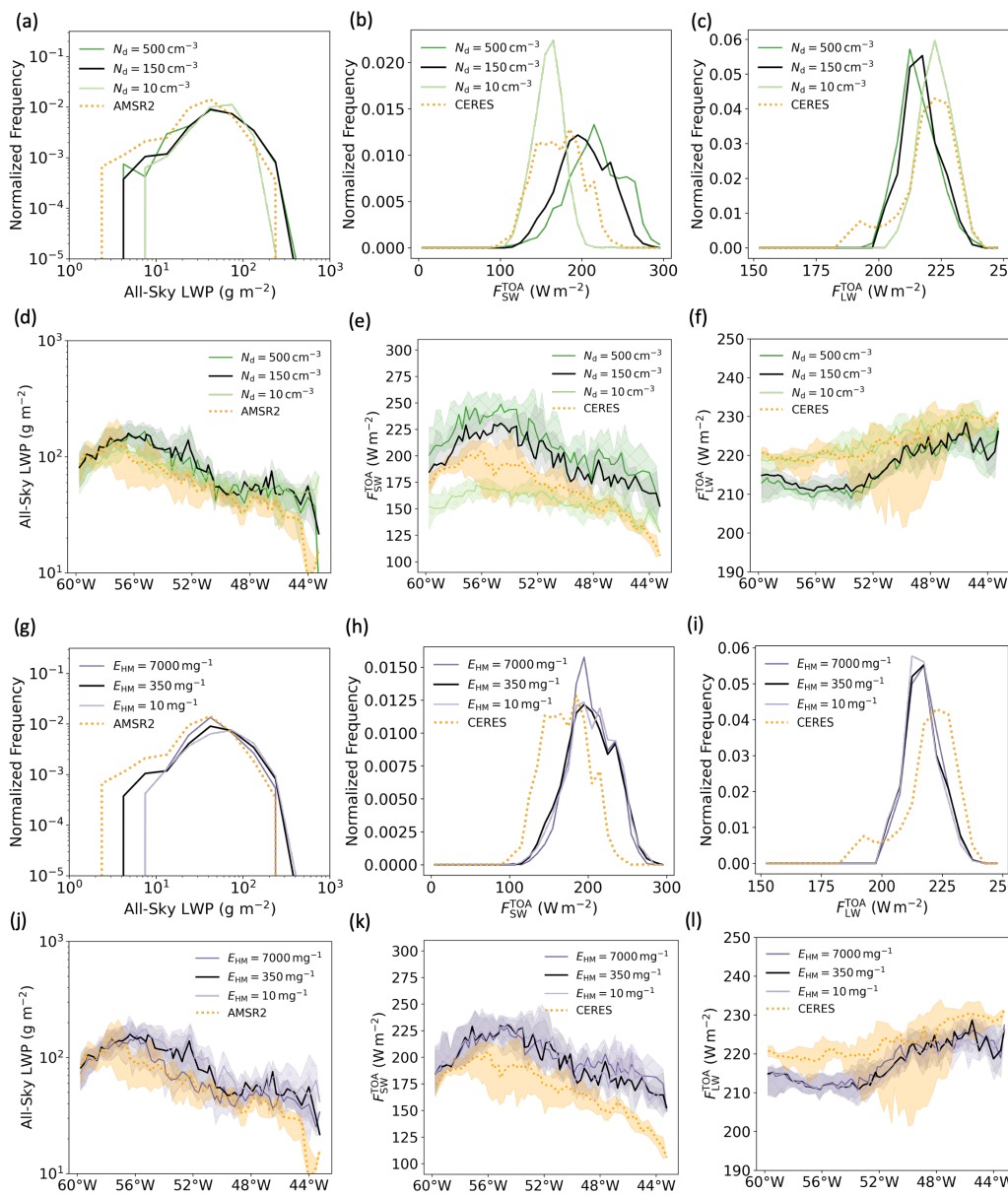

**Figure J2.** Model output compared with satellite retrievals for simulations with different $N_d$ (a-f) and $E_{HM}$ (g-l) on 24 October 2022: the normalized frequency of all-sky liquid water path (LWP) from AMSR-2 (left column), shortwave radiation at the top-of-atmosphere (middle column) and longwave radiation at the top-of-atmosphere from CERES (right column).

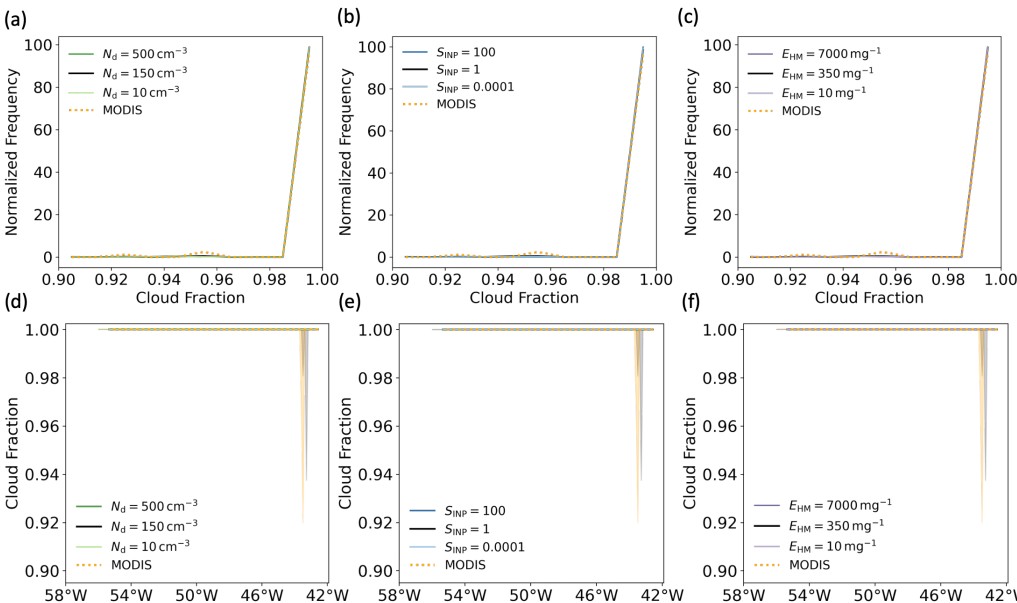

**Figure J3.** Modelled cloud fraction compared to cloud fraction retrieved from MODIS onboard the Aqua satellite with different $N_d$ (a,d), $S_{INP}$ (b,e) and $E_{HM}$ (g-l) on 15 March 2022. Cloud fractions from the model were calculated using the percentage of model grids with cloud cover larger than 20% in each coarser MODIS grid. Lines in Figures (d), (e) and (f) are the cross-section medians and the shaded area indicates the IQR (interquartile) range.

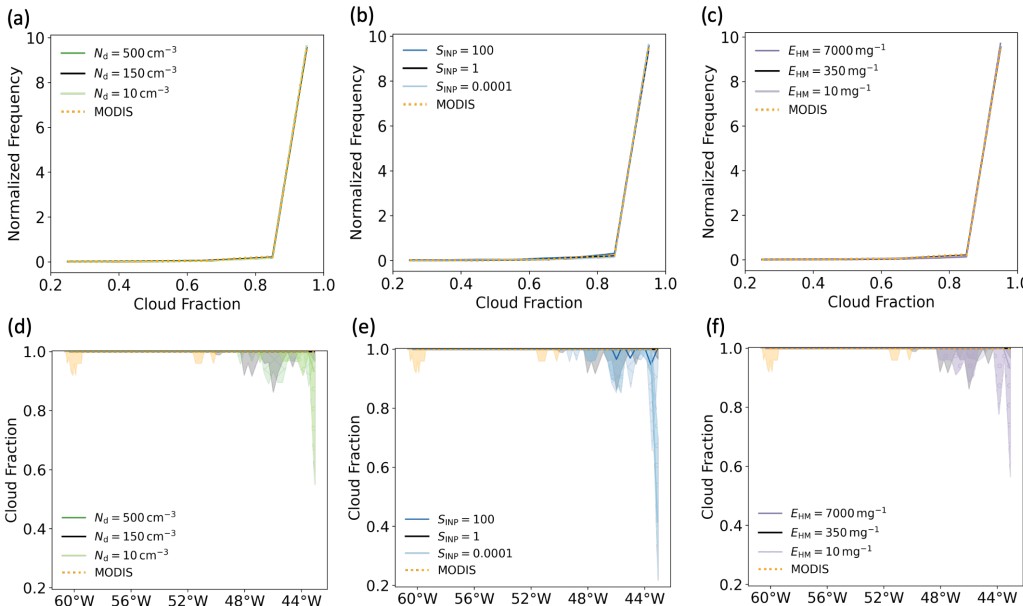

**Figure J4.** Modelled cloud fraction compared to cloud fraction retrieved from MODIS onboard the Aqua satellite with different $N_d$ (a,d), $S_{INP}$ (b,e) and $E_{HM}$ (g-l) on 24 October 2022. Cloud fractions from the model were calculated using the percentage of model grids with cloud cover larger than 20% in each coarser MODIS grid. Lines in Figures (d), (e) and (f) are the cross-section medians and the shaded area indicates the IQR (interquartile) range.

*Author contributions.* XH, PRF, BJM and KSC contributed to the design of this study. XH, PRF and DPG set up and performed the UM-CASIM simulations presented in this paper. XH performed the model output analysis. XH, DPG, FH and KSC performed comparison between model output and satellite retrievals. The original draft was written by XH, and edited by PRF, BJM, DPG, FH and KSC.

*Competing interests.* KSC is an executive editor of Atmospheric Chemistry and Physics.

*Acknowledgements.* We acknowledge the use of Monsoon, a collaborative high-performance computing facility funded by the UK Met Office and NERC (Natural Environment Research Council) for performing our model simulations. We acknowledge the use of JASMIN, the UK collaborative data analysis facility, for model and satellite data analysis. We thank the M-Phase team especially Erin N. Raif and Mark D. Tarn for measuring and providing the INP concentration on board the FAAM Atmospheric Research Aircraft. We acknowledge the use of satellite retrievals from NASA's Earth Observing System Data and Information System (EOSDIS) and the use of reanalysis data products

from the Copernicus Climate Change Service.

*Financial support.* The M-Phase aircraft campaign was supported by the Natural Environment Research Council (NERC) as part of the CloudSense programme (M-Phase: NE/T00648X/1 and NE/T006463/1). XH was supported by the SENSE - Centre for Satellite Data in Environmental Science CDT (Centre for Doctoral Training) (NE/T00939X/1) with a CASE studentship from the UK Met Office.

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
