# Peer review of "Different responses of cold-air outbreak clouds to aerosol and ice production depending on cloud temperature"

_EGUsphere, 2024_

## Author Response (AR1)

**Responses to reviewer comments on the manuscript "Different responses of cold-air outbreak clouds to aerosol and ice production depending on cloud temperature"**

We thank all the reviewers for their time and effort on reviewing our manuscript, and their insightful and constructive comments which have helped us to improve the scientific and presentation quality of our manuscript.

The validation of the control simulations against observations, especially for the cold March CAO case, is one of the major issues from all the reviewers' comments. To address this major issue, we have added some suggested model-observation comparison in *Appendix C* (*C1* for 15 March 2022, *C2* for 24 October 2022) in the revised manuscript for the control simulations of both CAO cases.

Detailed responses to each reviewer are shown below with the major/general comments shown first and the minor comments following. Note that the line numbers and section numbers may become different in the revised manuscript, so the locations of the old and new text are shown according to the original manuscript and the revised manuscript respectively. Long content such as the newly added appendices are also attached at the end of this document as appendices. Please note that the numbering of appendices in this document reflects the order in which they appear in this document, which differs from their order in the revised manuscript.

**Response to comments by reviewer 1**

| | |
|---|---|
| **Reviewer comment (1)** | I am particularly concerned because I don't think the 'stratocumulus' section of the March 15 simulation (the western portion of the simulation domain) shows sufficient skill to allow for the analysis presented – at least not without major caveats. From the RGB imagery (Figure 3a), I see relatively shallow roll clouds that grow quickly, which are common at the start of cold air outbreaks, when dry Arctic air first passes over warmer water. The strong winds and latent heat flux produce these rolls which progress to open MCC. The simulation, on the other hand, shows a thick, very cold stratocumulus, with the CWP two orders of magnitude more than MODIS CWP (Figure 4a and 4b). This is far more than 'within one order of magnitude' stated in the manuscript. The boundary layer in the simulated profile for this region (Figure 11, top row) extends to nearly 3 km. This is a region renowned for multilayer clouds (e.g., Mace et al., 2009 https://doi.org/10.1029/2007JD009755), yet the CONTROL simulation has a single, well-mixed boundary layer with a 3 km thick cloud? I need further evaluation work here.

Are there any aircraft profiles through this region from the M-Phase field campaign? Are there any upper air soundings from the coast of Canada?

At the least, I require an evaluation of cloud top temperature and cloud top height against MODIS products with a discussion focussed on the stratocumulus region of 15 March. In a perfect world, we would have a CALIPSO overpass to tell us the true cloud-top height and structure. |
| Our response: | We would like to first apologize for several mistakes and unclarities in the manuscript that might have led to some of the comments by the reviewer.

(1) The statement "within one order of magnitude" refers to the October case and we did not do any flights for the March case. We did not intend to use the CWP (cloud water path) from MODIS for a quantitative comparison and the statement "within one order of magnitude" was not aimed to explain the CWP comparison with MODIS, but for a validation of TWC (total water content) in the control simulation of the October case against the aircraft measurements. We understand our original phrasing can lead to misunderstanding and we have removed this and added more detailed information for model-observation comparison in *Appendix C*.

(2) We apologize for the mistake that the plots of the actual and potential temperatures in Fig.11 (a and f) were in the wrong order. The original 11(f) should be the profile in the stratocumulus-dominated (Sc) region, and the original 11(a) should be the profile in the cumulus-dominated (Cu) region. We have corrected this mistake |

|  | in the revised version of this manuscript and resolved the boundary layer depth issue suggested by the reviewer.  Please also note that the cloud microphysical properties are in-cloud values not grid-box mean values in Figure 11, therefore the profile can be higher than the boundary layer with the inclusion of a very small amount of relatively high clouds. For grid-box mean cloud profiles, please refer to Fig.D1(a) and D1(b) in *Appendix D* where the total cloud volume fraction and grid-box mean TWC (total water content) are plotted for the March case.

We believe our control simulation for the March case has a realistic simulated boundary layer for the Sc region. Based on the potential temperature profile in Figure 11, the boundary layer in the Sc region is around 2 km and gradually deepens to over 3 km in the Cu region. This is also shown in Fig.D1(a) and D1(b) in *Appendix D*. In addition, the output boundary layer types of the March control simulations (Fig.F1 in *Appendix F*) are dominated by the decoupled stratocumulus layers in the Sc region and becomes cumulus-capped layer dominated in the Cu region.

For the validation of the control March case, we have added the comparison of cloud top temperature from MODIS onboard the Aqua satellite, as well as temperature and IWC (ice water content) profiles from CALIOP onboard CALIPSO in *Appendix C1*. We do not think the cloud top height (CTH) retrieved from MODIS is valid, as it shows the CTH at the beginning of the CAO event almost reaching 5 km. We think the CTT from MODIS is valid and our model shows good agreements with it. |
|---|---|
| Old text: | *Line 193-Line 198* were removed as additional information is shown in *Appendix C* for the validation of control simulations.

Original *Appendix B* was completely rewritten to *Appendix C* in the revised manuscript. |
| New text: | Added *Line 213-Line 216*:
"Note that the CWP comparison is only qualitative here, and quantitative model-observation comparisons of the control simulations with satellite retrievals of cloud top temperature from MODIS, temperature and IWC (ice water content) from CALIOP for the March case, and MPhase aircraft measurements of temperature, cloud water content and liquid water fraction for the October case are shown and discussed in Appendix C."

Added *Appendix C1* for model-observation comparison of the March control simulation in the revised manuscript. Please see the attached *Appendix 1* at the end of this response document for the added content in *Appendix C1*. |

| | |
|---|---|
| **Reviewer comment** (2) | While the Oct case study also needs a more rigorous evaluation, I am more comfortable with the quality of the CONTROL simulation and the ensuing discussion. |
| Our response: | We have added other model-observation comparisons of temperature, cloud water content (TWC, LWC and IWC), and liquid water fraction profiles here in *Appendix C2* to strengthen the evaluation of the control simulation of the October case. For the stratocumulus clouds, the control simulation reproduced the liquid water fraction with a small overestimation of the total water content (profile mean bias of 0.048 $gm^{-3}$); for the cumulus clouds, the control simulation overestimated the liquid water content (0.018 $gm^{-3}$) and underestimated the ice water content (-0.03 $gm^{-3}$) for clouds below 1500m, resulting a higher liquid water fraction in the control simulation. Such bias is also seen in the all-sky LWP comparison in Figure 14. However, the control simulation captured the dominance of liquid in the cumulus clouds here, on which we based our conclusions and findings.

Unfortunately, a rigorous and systematic evaluation of the October case is not available in this paper as we do not have all the aircraft measurement data ready. |
| Old text: | N/A |
| New text: | Added *Appendix C2* for model-observation comparison of the October control simulation in the revised manuscript. Please see the attached *Appendix 1* at the end of this response document for the added content in *Appendix C2*. |
| | |
| **Reviewer comment** (3) | Given the detailed discussion on the SCT, it is worthwhile to establish the environment beyond the un-evaluated cloud top temperature.  It would be worthwhile to consider other aspects of the boundary layer environment and the role they may play in this transition.  Please comment on the SST, downstream SST gradient, the boundary layer stability, the M parameter and the estimated inversion strength (EIS).  As this manuscript reads, one might think that the only thing that matter are the cloud temperature and the microphysics. |
| Our response: | We have added the cross-section mean SST, mean downstream SST gradient, lower tropospheric stability (LTS), the CAO index (M parameter), and EIS as Figure D2 in Appendix D for the control simulations. Please note that SST was prescribed from daily forecasting analyses in our model simulations.

A description of these variables has been added in *Section 3.1*. We also added several sentences near the end of *Section 3.3* to highlight the importance of boundary layer environment for SCT in CAO events, which hopefully will not make the readers think |

| | |
|---|---|
| | only the cloud temperature and the microphysics matter for SCT. |
| Old text: | N/A |
| New text: | *Line 248-Line 255:*
"Meteorological variables and environmental conditions of boundary layer are also shown and compared for the control simulations of these cases as shown in Figure D2 of Appendix D. The SST (sea surface temperature) increases as clouds move eastward, with the SST temperature lower in the March case (2-3 ∘C) compared to the one in the October case (4-7 °C). Note that the SST was prescribed based on daily forecasting analyses in all our model simulations. The March case is a much stronger CAO event with the highest CAO index at 800 hPa (M800) almost reaching 20 K at the western boundary of the subdomain, while the highest M800 is around 1.5 K in the October case. The more unstable boundary layer in the March case is also consistent with a lower LTS (lower tropospheric stability) compared to the one in the October case. Both cases experience an EIS (estimated inversion strength) over 5 K at the western boundary of their subdomain with a decreasing trend to the east."

*Line 370-Line 374*:
"Several other factors may influence SCT during CAO events but they were not examined in this study. These include sea surface temperature (SST), which can impact convection and turbulence, as well as boundary layer stability, inversion layer strength and humidity. Additionally, this work focused on only three cloud microphysics parameters, whereas other microphysical processes could also potentially affect SCT. Future research should aim to explore the influence of these additional factors to gain a more comprehensive understanding of factors controlling SCT in CAO events."

Please see *Appendix 2* at the end of this document for the added Figure D2 in *Appendix D* of the cross-section mean SST, mean downstream SST gradient, lower tropospheric stability (LTS), the CAO index (M parameter), and the EIS. |

**Response to comments by reviewer 2**

| | |
|---|---|
| **Reviewer comment (1)** | One concern I have is with the consistency between the ice water paths and the precipitation rates in both simulations. The IWP in both simulations exceeds 1/2 kg/m2 as the clouds mature yet the precipitation rates seem to be quite small on the order of 5 mm/day in the mean. Because the satellite data cannot constrain these aspects of the simulations, it would be helpful to present other evidence that the relationship between IWP and precipitation rates are reasonable. My concern is that the model is too slow in making precipitation from the small ice that is nucleated in the supercooled environments. In Figures 11 and 12, the authors show vertical profiles of the ice properties such as number concentration but they do not separate cloud ice from precipitation. Some examination of the process rates would be interesting and might establish confidence since they could compare them to the process rates presented in other papers such as Karalis et al. (2022) whom they cite. |
| Our response: | We thank the reviewer for their suggestions on the further validation of the control simulations and please see below for our responses to (1) validating the relationship between IWP and precipitation rates, (2) presenting process rates and compare the process rates with other studies. |

(1) The IWP and precipitation relationship can vary with different cases, and precipitation in mixed-phase clouds can be strongly affected by other factor such as the number concentration of droplets and ice, therefore it is difficult to make a good validation using the relationship between IWP and precipitation. However, we checked the scatter plots from Matrosov (2024) where they plotted radar-derived snowfall rates versus IWP for North Slope of Alaska and MOSAiC (Multidisciplinary drifting Observatory for the Study of Arctic Climate) in Figure 3 with our peak IWP (over~750 gm$^{-2}$, 2.88 for the base-10 logarithm) and peak snowfall rate at surface (over~0.25 mm h$^{-1}$, -0.60 for the base-10 logarithm). For a similar IWP, their results show the snowfall rates ranging from 0.1 mm h$^{-1}$ to 0.3 mm h$^{-1}$, and our model sits within this range and closer to the higher end. Therefore, we suggest the precipitation rate in the control simulation of the March case is reasonable from this comparison.

(2) We agree with the reviewer that presenting and comparing process rates can be interesting and useful. However, process rates can be tricky when making comparisons as they are state-dependent. For example, the ice nucleation rates can be high when there is high INP concentrations with enough liquid water content, but can be low if the INP concentration is unchanged while the majority of the liquid water content is removed by the

high ice concentration from the high INP concentration. Similar reasons applied to the comparison of process rates with other studies. It is also difficult to make good comparisons of process rates when other model studies have completely different cases. Therefore, process rates are not presented and compared to the ones in other studies in this revised manuscript.

We understand that the original manuscript lacked validation of the March case. We have now added model-observation comparison against CALIPSO IWC (ice water content) and some other variables in *Appendix C1* for the March case. Please refer to our responses to the comment (1) from the first reviewer for more information on the evaluation of the March case control simulation against satellite retrievals.

Reference:
Matrosov, S. Y. : Statistical Relations among Solid Precipitation, Atmospheric Moisture and Cloud Parameters in the Arctic, Atmosphere, *15*(1), 132, https://doi.org/10.3390/atmos15010132, 2024.

| Old text: | N/A |
|---|---|
| New text: | N/A |

| **Reviewer comment (2)** | My primary concern is the validity of the March simulation.  This case is very cold and there is very little liquid water in these clouds (~10% of the water path).  Essentially, the model is producing cirrus clouds in the MBL.  Comparison to satellite LWP from both MODIS and AMSR2 show that the simulated LWP is biased low by an order of magnitude.  I question the overall validity of this simulation and whether it is suitable for this paper.  It would help immensely in establishing confidence if the authors could present additional evidence that such ice dominated MBL cloud fields actually exist in nature. |
|---|---|
| Our response: | We understand the concern from the reviewer on the validation of the March case and we have added evaluation of modelled temperature and IWC (ice water content) against CALIOP retrievals in *Appendix C1* to provide additional evidence required by the reviewer.

Temperatures from the control simulation of the March case is consistent with temperatures from CALIOP onboard the CALIPSO satellite as shown in Appendix C1 (for temperature profile in the domain: model median: -30.9 °C, CALIPSO median: -30.2°C). Therefore, the temperatures in the March case were very cold and our control simulation reproduced such cold environment.

With such high IWP (domain mean IWP from the control simulation: 631.85 g$m^{-2}$) and low LWP (domain mean LWP from AMSR-2: 17.31 g$m^{-2}$), it can be seen that the clouds during this March case were dominated by ice in such cold environment. |

| | Although our modelled LWP (domain mean LWP from the control simulation: 7.33 $gm^{-2}$) is biased low for ~10 $gm^{-2}$ when compared to the AMSR-2 LWP, it is quite consistent with the AMSR-2 LWP that there is limited liquid water for clouds in the March case.

We suggest that such bias of LWP is not critical as the LWP is a very small fraction of TWP (total water path, domain mean: 639.22 $gm^{-2}$) in the March case. Please also note that AMSR-2 LWP has a systematic error of 5 $gm^{-2}$ when rain is absent (Wentz, 1997) (uncertainties increase when there is precipitation) and has been shown a positive bias in previous studies (e.g., Seethala and Horvarth, 2010; Painemal et al., 2016). It is also shown that microwave retrievals of LWP onboard aircrafts in mixed-phase clouds is biased high compared to *in-situ* measurements (Klingebiel et al., 2023).

Our modelled IWC generally agrees with the CALIPSO IWC as shown in *Appendix C1*. The medians, interquartile ranges for the modelled IWC are 0.19 $gm^{-3}$, 0.08 $gm^{-3}$-0.39 $gm^{-3}$, and the ones for CALIPSO IWC are 0.17 $gm^{-3}$, 0.08 $gm^{-3}$-0.33 $gm^{-3}$. Therefore, we believe our model can reproduce the ice water properties in the March case. There is unfortunately no direct satellite-retrieved IWP for the March case as far as we know (CloudSat data is not available for the selected March case).

References:
Klingebiel, M. et al.: Variability and properties of liquid-dominated clouds over the ice-free and sea-ice-covered Arctic Ocean, *Atmospheric Chemistry and Physics*, 23(24), 15289–15304, https://doi.org/10.5194/acp-23-15289-2023, 2023.

Painemal, D., T. Greenwald, M. Cadeddu, and P. Minnis.: First extended validation of satellite microwave liquid water path with ship-based observations of marine low clouds, Geophys. Res. Lett., 43, 6563–6570, https://doi:10.1002/2016GL069061, 2016.

Seethala, C. and Horváth, A.: Global assessment of AMSR-E and MODIS cloud liquid water path retrievals in warm oceanic clouds, Journal of Geophysical Research: Atmospheres, 115, https://doi.org/https://doi.org/10.1029/2009JD012662, 2010.

Wentz, F. J.: A well-calibrated ocean algorithm for special sensor microwave / imager, J. Geophys. Res., 102(C4), 8703–8718, https://doi:10.1029/96JC01751, 1997. |
|---|---|
| Old text | N/A |
| New text | We have added evaluation of modelled temperature and IWC against CALIOP retrievals in *Appendix C1* in the manuscript. Please see the attached *Appendix 1* at the end of this response document for the added content in *Appendix C1*. |
| | |

| Reviewer comment (3) | *Line 106*: Not sure what this means. Why would ice and liquid be overlapped at 0.5? Why not some other number like 0.9 since ice falls from liquid clouds in these environments. |
|---|---|
| Our response: | More information has been added to explain this parameter in the *Method* section. Please see the "New text" box below for a detailed description of mixed-phase overlap factor.

The mixed-phase overlap factor ($mpof$) is a model parameter in the CASIM scheme of our model, which controls a function that quantifies the overlap during the run time of the model rather than being a fixed mixed-phase overlap in a model grid box. The degree of overlap of ice and liquid regions of the cloud during run time is controlled by $mpof$ as well as the liquid and ice cloud fractions in a model grid box.

Here is the function for calculating the overlap mixed-phase fraction ($O_{lf}$) in the model, using the mixed-phase overlap factor ($mpof$), liquid cloud fraction ($CF_{liq}$) and ice cloud fraction ($CF_{ice}$):

$$O_{lf} = \max\left(0.0, mpof \times min\left(CF_{liq}, CF_{ice}\right)\right) + \max(0.0, (1 - mpof)(CF_{liq} + CF_{ice} - 1))$$

As an example, when $mpof$ is set to 0.5, the mixed-phase overlap during run time is 1.0 when both liquid and ice cloud fractions are 1.0 (i.e., if both ice and liquid cover the whole grid box then the box must be fully mixed-phase). When liquid and ice cloud fractions are 0.5, the overlap would be 0.25. |
| Old text: | *Line 106-Line 107*:
"When both ice and liquid exist in the same grid box, a mixed-phase overlap fraction is calculated, with a default value of 0.5 along with liquid and ice cloud fractions from the cloud scheme." |
| New text: | *Line 107-Line 115*:
"When both ice and liquid exist in the same grid box, the overlap mixed-phase fraction is calculated, with a fixed mixed-phase overlap factor of 0.5 (Field et al., 2023). The overlap factor controls a function that quantifies the overlap during the run time of the model rather than being a fixed mixed-phase overlap in a model grid box. If the overlap factor is set to 1, then the subgrid liquid and ice cloud are maximally overlapped. If the overlap factor is 0, then the subgrid liquid and ice are not overlapped as long as $CF_{liq}+CF_{ice}<1$, where CF refers to cloud volume fraction. Once the combined cloud fraction goes above 1, there will be overlap. For an overlap factor of 0, the overlap is minimised. Values of the overlap factor in between lead to increasing overlap, but once either the liquid or ice cloud fraction reaches 1, then mixed-phase overlap is maximum whatever the overlap factor is set to. See section A.6 in the |

| | documentation of CASIM implementation in UM from Field (2023) for more information." |
|---|---|
| | |
| **Reviewer comment (4)** | *Line 144:* When referring to quantities that vary vertically in the atmosphere one should not use above and below to indicate magnitude changes of a quantity. For example, when referring to temperature, use warmer and colder since "above" can also refer to higher in the column creating ambiguity in the reader's mind. |
| Our response: | We thank the reviewer for this suggestion. We have changed the description of temperature change using "higher temperature" and "lower temperature". |
| Old text: | *Line 98-Line 99:*
"Heterogeneous ice nucleation is assumed to occur in grid boxes with temperatures below -8 °C, and below -38 °C homogeneous ice nucleation can occur."

*Line 144:*
"$T_0$ is set to -8 °C, meaning there are no INPs at temperatures above -8 °C." |
| New text: | *Line 100-Line 101:*
"Heterogeneous ice nucleation is assumed to occur in grid boxes with temperatures lower than -8 °C, and higher than -38 °C when homogeneous ice nucleation can occur. "

*Line 148:*
"265.15 K (-8 °C) was chosen as the warmest temperature for ice nucleation, meaning there are no INPs at temperature higher than -8 °C." |
| | |
| **Reviewer comment (5)** | What are the units of equation 1? |
| Our response: | Added |
| Old text: | N/A |
| New text: | *Line 144-Line 145:*
"Here we use a scale factor $S_{INP}$ (unitless) to change the INP concentration from the default Cooper parameterization:"

*Line 146:*
"The unit of $N_{INP}(T)$ is $m^{-3}$. The default value of $S_{INP}$ is 1.0. $T_0$ is 273.15 K and T is the ambient temperature (K)." |
| | |
| **Reviewer comment (6)** | SST and stability are relevant to understanding the events but neither are shown. |
| Our response: | We have added the cross-section mean SST, mean downstream SST gradient, lower tropospheric stability (LTS), the CAO index (M parameter), and EIS in Figure D2 of |

| | |
|---|---|
| | *Appendix D* for the control simulations, based on the suggestion from reviewer 1 (comment 3). |
| Old text: | N/A |
| New text: | *Line 248-Line 255:*
"Meteorological variables and environmental conditions of boundary layer are also shown and compared for the control simulations of these cases as shown in Figure D2 of Appendix D. The SST (sea surface temperature) increases as clouds move eastward, with the SST temperature lower in the March case (2-3 ∘C) compared to the one in the October case (4-7 °C). Note that the SST was prescribed based on daily forecasting analyses in all our model simulations. The March case is a much stronger CAO event with the highest CAO index at 800 hPa ($M_{800}$) almost reaching 20 K at the western boundary of the subdomain, while the highest $M_{800}$ is around 1.5 K in the October case. The more unstable boundary layer in the March case is also consistent with a lower LTS (lower tropospheric stability) compared to the one in the October case. Both cases experience an EIS (estimated inversion strength) over 5 K at the western boundary of their subdomain with a decreasing trend to the east."

*Line 370-Line 374*:
"Several other factors may influence SCT during CAO events but they were not examined in this study. These include sea surface temperature (SST), which can impact convection and turbulence, as well as boundary layer stability, inversion layer strength and humidity. Additionally, this work focused on only three cloud microphysics parameters, whereas other microphysical processes could also potentially affect SCT. Future research should aim to explore the influence of these additional factors to gain a more comprehensive understanding of factors controlling SCT in CAO events."

Please see *Appendix 2* at the end of this document for the added Figure D2 in *Appendix D* of the cross-section mean SST, mean downstream SST gradient, lower tropospheric stability (LTS), the CAO index (M parameter), and the EIS. |
| | |
| **Reviewer comment (7)** | Figure 4:  Seems that the LWP comparisons is not that great really.  It is very difficult to gauge what the water paths are to within an order of magnitude with the color bar used. The model does produce cloud streets and cellular clouds, but there seem to be pretty large differences in cloud fraction and water path in the March case. |
| Our response: | The LWP comparison was not aimed for a quantitative comparison and the "within an order of magnitude" is referred to the October case rather than the March case. This has been removed from the manuscript to avoid potential misunderstanding and the details of the validation of control simulations are now added in *Appendix C*. The colour bar used |

| | |
|---|---|
| | was intended to plot the clouds like reality (e.g., clouds with higher CWP are brighter and whiter) rather than to show quantitative values. |
| Old text: | *Line 193-Line 197:*
"Therefore, a preliminary comparison of total condensed water content (TWC) between the modelled and the observed cloud water content during the M-Phase campaign is shown in Appendix B for the October case. In general, our model is doing a reasonably good job of representing the CAO cloud features, with the highest bias within an order of magnitude." |
| New text: | *Line 213-Line 216:*
"Note that the CWP comparison is only qualitative here, and quantitative model-observation comparisons of the control simulations with satellite retrievals (March case) and MPhase aircraft measurements (October case) are shown and discussed in Appendix C." |
| | |
| **Reviewer comment (8)** | Figures 6 and 7 are very different but, according to the caption, are only different by 15 minutes in time. I think there is some mistake here. |
| Our response: | We are sorry that the caption of Figure 7 is wrong, and Figure 7 is for the October case. We have corrected the mistake. |
| Old text: | "… on 15 March at 17:00 UTC: …" |
| New text: | "… on 24 October 2022 at 17:00 UTC,: …" |
| | |
| **Reviewer comment (9)** | Is Figure 8 March or October? |
| Our response: | It's March and we are sorry for this mistake. This has been corrected. |
| Old text: | "…on 15 October…" |
| New text: | "…on 15 March 2022…" |
| | |
| **Reviewer comment (10)** | *Line 385*: I disagree with the contention that the March case LWP comparison to data are reasonable. The figures show they are different by an order of magnitude. |
| Our response: | We agree with the reviewer that there is an underestimation of LWP and therefore the modelled LWP does not agree well with the observed LWP from AMSR-2. However, the bias is less than an order of magnitude, as the domain-mean LWP from the control simulation is 7.3 gm$^{-2}$ and 17.3 gm$^{-2}$ for the domain-mean LWP from the AMSR-2 retrievals.

As we mentioned above, the LWP is a very small fraction of TWP (1.15% using the modelled mean LWP, 2.71% using the AMSR-2 mean LWP), so the cloud is essentially highly glaciated and the bias in LWP is not critical. |

| | |
|---|---|
| Old text: | *Line 385-Line 387:*
"In the March case, the control model shows reasonably good agreements of all-sky LWP, SW and LW fluxes at the top-of-the-atmosphere compared with other sensitivity test simulations. There is a small underestimation (approx. 5 g m$^{-2}$ of all-sky LWP compared to the AMSR-2 retrievals (Figure 13) from approximately 54 °W to 46 °W." |
| New text: | *Line 432-Line 434:*
"In the March case, the control simulation shows reasonably good agreements of SW and LW fluxes at the top-of-the-atmosphere compared with other sensitivity test simulations, however underestimates the all-sky LWP from approximately 54 °W to 46 °W (approx. 10 g m$^{-2}$ lower for the domain-mean LWP compared to the AMSR-2 retrievals (Figure 13)). "

*Line 439-Line 443:*
"Although our model underestimates the LWP, the LWP from AMSR-2 retrievals in fact suggests that the liquid water was very small in this March case (domain mean around 17 gm$^{-2}$). Clouds in the March case were dominated by ice with a high IWP (modelled domain mean around 632 gm$^{-2}$) and the control simulation shows good agreement of IWC against CALIOP retrievals (Appendix C). Therefore, our model agrees with the observations on the liquid-ice partitioning and that the CAO clouds in this March case were dominated by ice, on which we based our conclusions." |
| | |

**Response to comments by reviewer 3**

| Reviewer comment (1) | How common are the Warm/Cold (March/October) like cases?

The manuscript criticizes previous studies for only focusing on one case, and yet only two cases are explored here. These cases have (without sufficient support in my view) been presented as "archetypal". In particular, I think the manuscript should address how common both types might be, or at a minimum, present some evidence that very cold events like the March case are not unusual. |
|---|---|
| Our response: | We agree that we need to show how common these types of clouds are. We have added monthly distribution of cloud top temperature retrieved from MODIS in 2022 over the Labrador Sea region in *Appendix B*. Note that only six months of data are shown as the other months have few CAO events. CAO index at 800 hPa from ERA5 data was used to determine the CAO cloud grids.

We chose these two cases on the basis that they represent end members of the cloud-top temperature range of mixed-phase CAO clouds, while previous studies focused only on one of these, and the point of our paper is that the cases may behave differently. In fact, the March case was selected to represent the colder end of the CTT climatology, while the October case was selected to present the warmer end (Fig.B1 in *Appendix B*). We aimed to study the sensitivities by having two very contrasting cases instead of the most common cases, therefore these two cases nicely fit our purpose.

The word "archetypal" was perhaps not most appropriate, so has been removed. |
| Old text: | The world "archetypal" from *Line 7* is now removed. |
| New text: | Monthly distributions of cloud temperature for low-level, mixed-phase CAO clouds over the Labrador Sea in 2022 have been added into *Appendix B* in the revised manuscript. Please see *Appendix 2* at the end of this document for the added content in *Appendix B*.

*Line 190– Line 196:*
"Monthly distributions for cloud top temperature of low level, mixed-phase clouds during CAO events over the Labrador Sea in 2022 are shown in Appendix B using the ERA5 (ECMWF ReAnalysis version 5) (Hersbach et al., 2020) dataset and CTT retrieved from MODIS. The CTTs in the March case are near the colder end of the shown distributions, while the ones in the October case are more close to the warmer end, which suggests that these two CAO cases are nicely contrasting from each other in terms of CTTs and sit near the boundaries of CTT ranges in CAO clouds over the Labrador Sea. These two cases were chosen on the basis that they represent end members of |

| | the temperature range of mixed-phase CAO clouds. Detailed information for the method of analysis is shown in Appendix B." |
|---|---|
| **Reviewer comment (2)** | The comparison of the control simulation to the observations is quite limited:

A) On line 197 you write "A full model-observation comparison for this case is in preparation and will be shown in a subsequent paper." In this event, why not wait for the "full" comparison to be completed in order to build appropriate confidence that these simulations are reliable and perhaps better establish a nominal control case? |
| Our response: | We have added more comparison of the October case against observations from the Nevzorov probes in *Appendix C2* to strengthen the validation of the control simulation. As this comment is also linked to comment (4) below, please see a detailed response of October case validation for comment (4).

Unfortunately, a rigorous and systematic evaluation against new aircraft measurements of the October case is not possible in this paper as we do not have all the observation data ready yet when this manuscript was written. |
| Old text | Please see the content for comment (4) |
| New text | Please see the content for comment (4) |
| **Reviewer comment (3)** | B) Line 379. Given that events lasted days, why is the analysis restricted to essentially one A-train overpass, and even then, the only microphysical quantity used is LWP from AMSR? What about geostationary datasets? Yes, one has to be careful are higher latitudes and larger solar zenith angles, but these data are far from useless. I note the CERES SW, LW, Albedo data that are used, depend on cloud cover, optical depth, etc. from satellite datasets that are not used here (because it is suggested they are too uncertain). |
| Our response: | One main issue we had when selecting the satellite products to compare with the model output is how to make sure we are comparing like-to-like, as variables from satellite are retrieved using measured radiances and algorithms instead of being measured and observed directly. One way to overcome this issue is to use satellite simulators such as COSP (CFMIP Observation Simulator Package), but unfortunately COSP was not available for high-resolution regional simulations in our model when we performed simulations and the later on analysis. This issue directly limited us to compare cloud optical properties from satellite such as the optical depth and effective radius.

Yes, geostationary dataset can be helpful to some extent for model-observation comparison, but we doubt adding them especially those with high uncertainties will further our |

| | understanding, as we already have better and more accurate products from instruments onboard polar-orbit satellites for comparison (MODIS, CERES, AMSR-2, CALIOP) or *in-situ* observations. However, we understand the concern from the reviewer that the validation of the March case was limited in the original manuscript, therefore we have added evaluation of cloud top temperature (CTT) against MODIS retrievals, and temperature and IWC against CALIOP retrievals in *Appendix C1* for the March case. More model-observation comparison of the October control simulation against the MPhase aircraft measurements have also been added *in Appendix C2*. Please see our response for comment (4) below for more details. |
|---|---|
| Old text: | Please see the content for comment (4) |
| New text: | Please see the content for comment (4) and *Appendix 1* in the end of this document for the added model-observation comparison. |
| | |
| **Reviewer comment (4)** | C) The comparison to in situ data is largely confined to a small Appendix (B) without much detail. In Fig. B1 panel C. What are the symbols? How do the satellite data compare to the aircraft data? In general, TWC data appears to span orders of magnitude. I'm not sure how one can justify the model as having done "well" based on this comparison. Please quantify (with appropriate statistics) and explain why this is "well". Is the position of the aircraft (relative to cloud base) accounted for in any way? In Fig B1 panel B, it appears that perhaps the aircraft was using ramps from which profiles of LWC/IWC and total liquid/ice water path might be obtained. If yes, it looks like there might have been two-cloud-layers on two-of the ramps? Are the later aircraft data from constant-altitude-legs (that might allow comparison for of obs/models at a fixed altitude)? Was there no other microphysical information coming from the aircraft beside data from the Nevzorov probes? Even if yes, why not compare/discuss the relative abundance of liquid vs. ice? |
| Our response: | We have rewritten the original *Appendix B* (it is *now Appendix C2* in the revised manuscript) for the validation of October control simulation against aircraft measurements.

Instead of using temperature-binned plots as in the original manuscript, we have now made vertical profiles for temperature, grid-box mean TWC (total water content), LWC (liquid water content) and IWC (ice water content), as well as the liquid water fraction (for the relative abundance of liquid vs. ice) for two MPhase flights on 24 October 2022 in *Appendix C2*. These two flights (MPhase C322 and C323) were in the stratocumulus-dominated region and cumulus-dominated region respectively.

For the stratocumulus clouds, the control simulation reproduced the liquid water fraction with a small overestimation |

of the total water content (profile mean bias of 0.048 gm$^{-3}$); for the cumulus clouds, the control simulation overestimated the liquid water content (0.018 gm$^{-3}$) and underestimated the ice water content (-0.03 gm$^{-3}$) for clouds below 1500m, resulting a higher liquid water fraction in the control simulation. Such bias is also seen in the all-sky LWP comparison in Figure 14. However, the control simulation captured the dominance of liquid in the cumulus clouds here, on which we based our conclusions and findings.

In terms of how the satellite data compare to the aircraft data, it is beyond the scope our work here and we do not have the answer for the reviewer. However, works are being conducted to compare MPhase measurements (all flights, not limited to this October case) with satellite retrievals and we would like to draw the reviewer's attention to upcoming papers from the MPhase team.

There were other measurements beyond the ones from the Nevzorov probes, but the data is not ready yet.

| | |
|---|---|
| Old text: | *Appendix B* |
| New text: | We have added more evaluation of the control simulation against MPhase aircraft measurements in *Appendix C2* in the revised manuscript. Please refer to *Appendix 1* at the end of this document for the added content. |

| | |
|---|---|
| **Reviewer comment (5)** | *Abstract Line 2.* Not entirely sure who is "our". The authors of this study? The larger scientific community? Perhaps simply change to read "Recent case studies of CAO events suggest that increases …". |
| Our response: | Changed. |
| Old text: | *Line 2-Line3:* "Our current understanding is that increases in ice nucleating particle (INP) concentrations" |
| New text: | *Line 2-Line3:* "Recent case studies of CAO events suggest that increases in ice-nucleating particle (INP) concentrations" |

| | |
|---|---|
| **Reviewer comment (6)** | *Abstract Line 3.* Yes, on a reduction in total water content, but not sure this is true for reflectivity. What radar wavelength? Depending on the wavelength, reflectivity is largely controlled by precipitation particle size or amount (rather than differences in dielectric constant between liquid and ice) such that more precipitation will often lead to a larger reflectivity. |
| Our response: | For the reflectivity here, we are referring to shortwave flux and albedo at the top of the atmosphere. This has been changed to avoid misunderstanding. |

| | |
|---|---|
| Old text: | *Line 3:*
"…cause a reduction in cloud total water content and reflectivity." |
| New text: | *Line 3-Line4:*
"…cause a reduction in cloud total water content and albedo at the top of the atmosphere." |

| **Reviewer comment (7)** | *Line 22.* Perhaps "a key" rather than "the key"? |
|---|---|
| Our response: | Changed. |
| Old text: | *Line 22:*
"…physical representations of clouds are the key reason why models in CMIP6…" |
| New text: | *Line 22:*
"…physical representations of clouds are a key reason why models in CMIP6…" |

| **Reviewer comment (8)** | *Line 26.* Similarly, perhaps "major" rather than "the main". There are a lot of uncertainties. |
|---|---|
| Our response: | Changed. We believe the reviewer was referring to *Line 24* for this comment. |
| Old text: | *Line 24:*
"…is one of the main reasons for radiative flux biases…" |
| New text: | *Line 24:*
"…is one of the major reasons for radiative flux biases…" |

| **Reviewer comment (9)** | *Line 41.* Here and next sentence, perhaps credit authors upfront rather than parenthetically. For example, "Field et al. (2014) found an improvement …" |
|---|---|
| Our response: | Changed. |
| Old text: | *Line 40-Line 41:*
"An improvement of LWP (liquid water path) and radiation bias was achieved by modifying the boundary layer parameterization and by inhibiting heterogeneous ice formation in CAO clouds (Field et al., 2014)."

*Line 41-Line 43:*
"It has also been shown that changes in the INP (ice-nucleating particle) concentration can strongly modulate the freezing behaviour of cloud droplets and the reflectivity of CAO clouds through changing the liquid-ice partitioning in mixed-phase CAO clouds (Vergara-Temprado et al., 2018)" |

| | |
|---|---|
| New text: | *Line 40-Line 41:*
"Field et al. (2014) found that an improvement of LWP (liquid water path) and radiation bias can be achieved by modifying the boundary layer parameterization and by inhibiting heterogeneous ice formation in CAO clouds."

*Line 41-Line 44:*
"It is also found by Vergara-Temprado et al. (2018) that changes in the INP (ice-nucleating particle) concentration can strongly modulate the freezing behaviour of cloud droplets and the albedo of CAO clouds through changing the liquid-ice partitioning in mixed-phase CAO clouds…" |

| | |
|---|---|
| **Reviewer comment (10)** | *Line 45.* In my view, "Stratocumulus-to-cumulus transition (SCT)" is not a process, and is a description (it doesn't "affect the amount of stratocumulus and cumulus clouds in the cloud field). Perhaps simply, "Stratocumulus-to-cumulus transitions (SCT) in CAOs have an important radiative effect." |
| Our response: | Changed. |
| Old text: | *Line 44-Line 47:*
"Stratocumulus-to-cumulus transition (SCT), during which the change of cloud regimes happens, is an important process in CAO clouds as it can affect the amount of stratocumulus and cumulus clouds in the cloud field, hence influencing the radiative effects of the CAO clouds." |
| New text: | *Line 44-Line 45:*
"Stratocumulus-to-cumulus transition (SCT) in CAOs have an important influence on the radiative properties of CAO clouds,…" |

| | |
|---|---|
| **Reviewer comment (11)** | \*\**Line 58 (and others).* Is "Murray and the MPhase Team, 2024" really the best available reference for the campaign? This is just a conference abstract with no links depicting flights, conditions observed, instruments, science plan OR information on how to obtain data, etc. Frankly, a campaign web site might be more useful. |
| Our response: | We have now added a new citation for the INP dataset and description now from Tarn et al., (2025) in addition to the Murray and the MPhase Team (2024) citation. The data paper and relevant dataset of the MPhase campaign which includes detailed information for this campaign are still in preparation, and we are sorry that the campaign website does not show these information either. The conference abstract is still now one of the best sources of information for the general MPhase aircraft campaign. |
| Old text: | N/A |
| New text: | New citation from Tarn et al., (2025) is added to the original Murray and the MPhase Team (2024) citation. |

| | |
|---|---|
| **Reviewer comment** (12) | Figure 1. Please show the domain simulated in Figure 1. More generally, why show most of Europe and parts of North Africa. In my view it would be better to show a narrower region that focuses on region of study so as to provide more detailed picture of the relevant meteorology. |
| Our response: | We agree with the reviewer that a narrower region focusing on the domain of study will be provide more detailed information. Unfortunately, the synoptic charts were directly provided from the UK Met Office, and we could not make further modifications such as selecting the domain of interest or adding other information onto the plots. |
| Old text: | N/A |
| New text: | N/A |
| | |
| **Reviewer comment** (13) | *Line 94*. Fixed Nd may make interpretation easier but it also removes potential feedbacks created by aerosols/CCN being removed via coalescence of cloud droplets (precipitation formation). Surely this is worth a line or two of text and perhaps some considerations as regards future activities discussed in the final section. |
| Our response: | We agree with the reviewer and have added the limitation of using fixed $N_d$ in the Method section. Discussion on future work on using aerosol-derived $N_d$ for a better representation of aerosol-cloud interactions are added in the Discussion and Conclusions section. |
| Old text: | N/A |
| New text: | *Line 94-Line 96*:
 "However, it is worth noting that using fixed in-cloud Nd instead of having aerosols involved can remove potential feedback between aerosols and clouds, for example, precipitation formed in clouds can remove aerosols/CCNs which can enhance the precipitation and further remove the aerosols."

 Line 520-Line 523:
 "In addition, fixed in-cloud $N_d$ was used in this study for the easier interpretation of sensitivity test results, but such a setup can lead to the neglect of potential feedbacks between cloud and aerosols/CCNs. Future work will also include aerosol-derived $N_d$ and cloud processing of aerosols where possible for a better representation of aerosol-cloud interactions." |
| | |
| **Reviewer comment** (14) | *Line 106*. I do not understand what "… a mixed-phase overlap fraction is calculated, with a default value of 0.5 along with liquid and ice cloud fractions from the cloud scheme" is intended to convey. Please expand or rephrase the description, and in particular, please address the implications this has for the results presented in the manuscript. |

| | |
|---|---|
| Our response: | We have now corrected and expanded the description for mixed-phase overlap factor and overlap mixed-phase fraction in the *Method* section. Please also see our response to comment (3) from reviewer 2 above for a detailed explanation of the mixed-phase overlap factor in the model. |
| Old text: | *Line 106-Line 107*:
"When both ice and liquid exist in the same grid box, a mixed-phase overlap fraction is calculated, with a default value of 0.5 along with liquid and ice cloud fractions from the cloud scheme." |
| New text: | *Line 107-Line 115*:
"When both ice and liquid exist in the same grid box, the overlap mixed-phase fraction is calculated, with a fixed mixed-phase overlap factor of 0.5 (Field et al., 2023). The overlap factor controls a function that quantifies the overlap during the run time of the model rather than being a fixed mixed-phase overlap in a model grid box. If the overlap factor is set to 1, then the subgrid liquid and ice cloud are maximally overlapped. If the overlap factor is 0, then the subgrid liquid and ice are not overlapped as long as $CF_{liq}+CF_{ice}<1$, where CF refers to cloud volume fraction. Once the combined cloud fraction goes above 1, there will be overlap. For an overlap factor of 0, the overlap is minimised. Values of the overlap factor in between lead to increasing overlap, but once either the liquid or ice cloud fraction reaches 1, then mixed-phase overlap is maximum whatever the overlap factor is set to. See section A.6 in the documentation of CASIM implementation in UM from Field (2023) for more information." |
| | |
| **Reviewer comment (15)** | *Line 119*. Do you mean equivalent mass? equivalent volume? |
| Our response: | It's the equivalent mass spherical radius and we have added this information in. |
| Old text: | *Line 119:*
"…are calculated using an equivalent spherical radius…" |
| New text: | *Line 127:*
"…are calculated using an equivalent mass spherical radius…" |
| | |
| **Reviewer comment (16)** | *Line 138*. Units? Perhaps give concentrations in #/Liter? |
| Our response: | Units have now been added. |
| Old text: | N/A |
| New text: | *Line 144-Line 145*:
"Here we use a scale factor $S_{INP}$ (unitless) to change the INP concentration from the default Cooper parameterization:"

*Line 146:* |

| | "The unit of $N_{INP}(T)$ is $m^{-3}$. The default value of $S_{INP}$ is 1.0. $T_0$ is 273.15 K and T is the ambient temperature (K)." |
| --- | --- |
| | |
| **Reviewer comment (17)** | *Line 143*. The slope matches, but figure 2a would suggest to me the control case should perhaps be Sinp = 0.01. So, using Sinp values of 0.0001, 0.01, and 1 for sensitivity tests. |
| Our response: | $S_{INP}$ is a scale factor for the original Cooper parameterization and therefore the default $S_{INP}$ = 1.0 is the grey dashed line shown in Figure 2. The scale factor does not change the slope of the INP curve, but scales the INP concentration up and down. The INP concentration as shown in Figure 2 is calculated by multiply the $S_{INP}$ to the INP concentration from Cooper parameterization at a certain temperature. |
| Old text: | N/A |
| New text: | N/A |
| | |
| **Reviewer comment (18)** | *Line 384*. I am not sure this is the best way to interpret the comparison. 5 g/m2 out of what? Presumably less than 10 g/m2 on average? What is the minimum LWP that AMSR can reasonably identify? How good do you expect AMSR LWP to be? My take would be that AMSR-2 shows LWP is small and this is consistent with the model for this case. |
| Our response: | We are sorry for not providing enough information for this value and have now changed to "…(approx. 10 g m−2 lower for the domain-mean LWP compared to the AMSR-2 retrievals) …", as the domain mean LWP for the control simulation is around 7.3 gm$^{-2}$ and 17.3 gm$^{-2}$ for the domain-mean LWP from the AMSR-2 retrievals

AMSR has no retrieval range of LWP but a systematic error of 5 g m$^{-2}$ when rain is absent (Wentz, 1997) (uncertainties increase when there is precipitation) and has been shown a positive bias in previous studies (e.g., Seethala and Horvarth, 2010; Painemal et al., 2016). It is also shown that microwave retrievals of LWP onboard aircrafts in mixed-phase clouds is biased high compared to *in-situ* measurements (Klingebiel et al., 2023).

We agree with the reviewer that the LWP from AMSR-2 shows that the LWP in the March case is very small and is consistent with the model.

| | |
|---|---|
| | with ship-based observations of marine low clouds, Geophys. Res. Lett., 43, 6563–6570, https://doi:10.1002/2016GL069061, 2016.

Seethala, C. and Horváth, A.: Global assessment of AMSR-E and MODIS cloud liquid water path retrievals in warm oceanic clouds, Journal of Geophysical Research: Atmospheres, 115, https://doi.org/https://doi.org/10.1029/2009JD012662, 2010.

Wentz, F. J.: A well-calibrated ocean algorithm for special sensor microwave / imager, J. Geophys. Res., 102(C4), 8703–8718, https://doi:10.1029/96JC01751, 1997. |
| Old text: | *Line 385-Line 387:*
"In the March case, the control model shows reasonably good agreements of all-sky LWP, SW and LW fluxes at the top-of-the-atmosphere compared with other sensitivity test simulations. There is a small underestimation (approx. 5 g m−2 of all-sky LWP compared to the AMSR-2 retrievals (Figure 13) from approximately 54 °W to 46 °W." |
| New text: | *Line 432-Line 434:*
"In the March case, the control model shows reasonably good agreements of SW and LW fluxes at the top-of-the-atmosphere compared with other sensitivity test simulations, however underestimates the all-sky LWP from approximately 54 °W to 46 °W (approx. 10 g m$^{-2}$ lower for the domain-mean LWP compared to the AMSR-2 retrievals (Figure 13)."

*Line 439-Line 443:*
"Although our model underestimates the LWP, the LWP from AMSR-2 retrievals suggests that the liquid water was very limited in this March case (around 17 gm−2) and the clouds in the March case were dominated by ice with a high IWC suggested from the CALIPSO retrievals (Appendix C1). Therefore, our model agrees with the observations on the liquid-ice partitioning and that the CAO clouds in this March case were dominated by ice, on which we based our conclusions." |
| | |
| **Reviewer comment** (19) | *Line 387.* Your write "Small underestimation of SW flux …". Do you mean in the default run? Why do you suspect cloud cover is the issue? |
| Our response: | Yes, this is for the control simulation, and it is mentioned near the end of the sentence.

We suggest this underestimation of SW flux and overestimation of LW flux may come from the bias of cloud cover and IWP (IWP wasn't in the original manuscript and we have added in the revised version), as the SW and LW flux in the March case are strongly controlled by the cloud cover and IWP shown in Figure 6 and there is little influence from |

| | |
|---|---|
| | changing $N_d$ as shown in Appendix I1. Although an overestimation of LW flux may also because of cloud temperature bias, but such will not lead to an underestimation in the SW flux. Therefore, we suggest the bias of cloud cover and IWP may be the reason. |
| Old text: | *Line 390:*
"…which may due to the cloud cover in the control simulation being slightly lower than the observed…" |
| New text: | *Line 437-Line 438:*
"…which may due to the cloud cover and IWP in the control simulation being slightly lower than the observed…" |
| | |
| **Reviewer comment (20)** | **Figure 5 & C. This is all model data, yes? Perhaps show uncertainty in the mean and/or IQR or some measure of spatial variability. Why is observational satellite data not shown … at least for cloud-cover, albedo, SW, LW (which you seem to trust). Also please consider putting March and October lines in same panels (unless you are going to add observations and this makes it hard to see). |
| Our response: | Yes, these are all model data. We have now added the +/- 1 standard deviation to show the spatial variability for all the variables in Figures 5, D1 and D2. The observational data such as LWP, SW, LW and cloud cover (*Appendix I4*) are shown later in Section 3.5 for model-observation comparison of all simulations, as these are of primary importance for the cloud radiative effects and can be directly compared to without using the satellite simulator (which was not available for our model when we performed the simulations). Additional model-observation comparisons for the control simulations are shown in *Appendix C*.

We have carefully considered the reviewer's suggestion on putting March and October lines in same panels, but this will make it difficult to see the trend of LWP and IWP through the development of the CAO field if using the same y axis and scale. A twin axis can be used but may cause extra reading effort than the original two column plots. The included spatial variability (coloured range) may also overlap and makes it difficult to read. We hope these explain the reasons why we chose to stay with the two column plots instead of taking the reviewer's suggestion. |
| Old text: | Figures 5 and C1 |
| New text: | Figures 5, D1 and D2 |
| | |
| **Reviewer comment (21)** | Figure 5e & C1d. How is 5e consistent with C1d for October (**I think perhaps October data in panels C1e and C1d been swapped)? |

| Our response: | We thank the reviewer for their attention on the details of this manuscript. Yes, October data in panels C1e and C1d were swapped and we are sorry for this mistake. This has now been corrected. |
|---|---|
| Old text: | Figure C1 |
| New text: | Figure D1 |
| | |
| **Reviewer comment (22)** | Lines 200 to 217. I think many individual references here to figures 4a to 4e are supposed to be 5a to 5e. |
| Our response: | Corrected. |
| Old text: | Incorrect Figures from *Line* 200 to *Line* 217 |
| New text: | *Corrections in Line* 221-*Line* 235 |
| | |
| **Reviewer comment (23)** | ** *Line 244*. You have already said there is little LWP several times. I presume the simulations of Tornow and Able were for much warmer clouds and contained more Liquid. If yes, I don't know why you are expecting results to be similar. |
| Our response: | Yes, there is little LWP in the March case and it's different from the simulations of Tornow and Able that focused on warmer clouds with more liquid. However, in this sentence, we are stating that the change of LWP, IWP and liquid water fraction from changing INP concentration is consistent with previous studies, that is a higher INP concentration leads to higher IWP and lower LWP, and a lower INP concentration leads to lower IWP and lower IWP. Such influences of changing INP concentration on these variables are indeed consistent with previous studies, no matter the case is warm or cold.

The difference of the March case to the previous warmer case studies is that with a higher IWP from high INP concentration, the removal of cloud water is not enhanced as there is little liquid water to be removed and higher ice crystal number concentration leads to lower snow autoconversion. This difference makes the cloud cover, surface precipitation and albedo responses to changing INP concentration different from the previous studies. We see some similar responses to INP concentrations for LWP, IWP, but opposite responses for cloud cover, surface precipitation and TOA albedo. |
| Old text: | N/A |
| New text: | N/A |
| | |
| **Reviewer comment (24)** | ** *Line 278*. You write, "The responses become complex and some even non-monotonic near the eastern end of the sub-domain". Yes, so what is going? The discussion on the role of SIP just seems to end with "it is complicated", which is not very satisfying or useful. |

| | |
|---|---|
| Our response: | We agree with the reviewer that more explanation should be added to the role of SIP. Please see the added text below for our added content and corrections. |
| Old text: | *Line 284-Line 286*:
"Although there are limited influences from low $E_{HM}$, one might notice that the responses of some cloud properties become complicated and non-monotonic (e.g., the default model output is outside the low and high model output range) near the end of the sub-domain where cumulus clouds dominate." |
| New text: | *Line 310-Line 312:*
"This is because the HM process in the model is dependent on the processes of cloud water accretion onto graupel and snow, while ice is very limited in the stratocumulus-dominated region but a higher IWP is seen after SCT."

*Line 312-Line 314:*
"This is because although the HM process is the source of ice crystals, it is also the sink for graupel and snow which can accrete and remove water through precipitation in the model."

*Line 317-Line 385:*
"Although low $E_{HM}$ has a limited influence compared to the ones from high $E_{HM}$, the response of surface precipitation to $E_{HM}$ becomes complicated and non-monotonic (e.g., the default model output is outside the low and high model output range) near the end of the sub-domain where cumulus clouds dominate. For example, low $E_{HM}$ results in a stronger surface precipitation from 52 °W to 50 °W, but a weaker surface precipitation around 46 °W, compared to the precipitation from the control simulation. This may because precipitation rate is a state-dependent variable, and a low $E_{HM}$ leads to an earlier peak of precipitation when clouds move eastward compared to the one from the control simulation, followed by a lower precipitation rate later as less cloud water exist." |
| **Reviewer comment (25)** | *\*\* Line 280-286*. Somewhat similar to the above comment, the take away on the effect of EHM changes just seems to be "it is complicated". As best I can see there are no conclusions in this manuscript as regards the importance of the HM process. You might look at total number of ice particles and the production rates of ice changes due to SINP and EHM. Are EHM driven changes relatively small or only happening in only part of the domain? |
| Our response: | The influence of changing HM efficiencies becomes stronger in the cumulus-dominated region for the October case, as there is more ice and the HM process is dependent on cloud water accretion onto graupel and snow. The importance of the HM process has been stated in *Line 364-Line 379* in the original manuscript where the fractional changes are presented. We have also added more explanation and statements on the |

| | We agree with the reviewer that it will be interesting to investigate the ice production rates. However, process rates can be tricky when making comparisons as they are state-dependent. For example, the ice nucleation rates can be high when there is high INP concentrations with enough liquid water content, but can be low if the INP concentration is unchanged but the majority of the liquid water content is removed by the high INP concentration. Therefore, we have not added any process rates in this manuscript. |
|---|---|
| Old text: | (Please see the old text section for the above comment 24) |
| New text: | (Please see the new text section for the above comment 24) |
| | |
| **Reviewer comment (26)** | Figure 7. The caption claims these are results for 15 March, but I believe they are for the October Case. |
| Our response: | Yes, this figure is for the October case and we are sorry for this mistake. This has now been corrected. |
| Old text: | "… on 15 March at 17:00 UTC: …" |
| New text: | "… on 24 October 2022 at 17:00 UTC,: …" |
| | |
| **Reviewer comment (27)** | Figure 8. Appears to have the opposite problem. Claims to be for the October case but is for March? Please check your captions carefully. |
| Our response: | Yes, this is for the March case and we are sorry for this mistake. This has now been corrected. |
| Old text: | "…on 15 October…" |
| New text: | "…on 15 March 2022…" |
| | |
| **Reviewer comment (28)** | *Line 383.* When SZA & Hsigma are large this is legitimate concern. Note from their Figure 2, however, this means Hsigma > 10. Is this really the case here? Surely not for the whole domain? |
| Our response: | SZA-dependent bias is not the main reason that we chose not to use MODIS-retrieved LWP for a quantitative comparison here, but the mixed-phase cloud bias as MODIS is unable to identify the mixed-phase nature of mixed-phase clouds. The main problem is that MODIS-retrieved LWP/IWP assumes a homogeneous vertical structure of the clouds, while most of the clouds from the two cases were not. In fact, the MODIS-retrieved LWP is CWP when the cloud top phase is liquid, and the MODIS-retrieved IWP is CWP when the cloud top phase is ice. This is not the actual LWP and IWP (columnar sum of cloud liquid/ ice) we would like to compare with. Using a satellite simulator (with modelled radiances and similar retrieval algorithms) may overcome this problem, but unfortunately COSP was not available for regional simulations when our model simulations were performed for this study. We |

| | |
|---|---|
| | have added "mixed-phase cloud bias" in the original sentence to avoid potential misunderstanding and give further information here for the reason why MODIS-retrieved LWP was not used for a quantitative comparison in our work. |
| Old text: | *Line 383-Line 384:*
"MODIS-retrieved LWP is not used for quantitative comparison here due to its uncertainties in high-latitude mixed-phase clouds (Khanal and Wang, 2018)." |
| New text: | *Line 430-Line 431:*
"MODIS-retrieved LWP is not used for quantitative comparison here due to its high mixed-phase cloud bias (Khanal and Wang, 2018)." |
| | |
| **Reviewer comment (29)** | *Line 391.* You write "The simulation with a high SINP in the October case agrees well with all satellite retrievals in general (Figure 14). There is a small overestimation of all-sky LWP …".  I don't think I agree with this assessment. For LWP the default and small Sinp simulations look good on the western boundary but all simulations look poor in the east and only high SNIP compares well near 52W (in the middle).  A similar comment as regards SW might be made (all look bad from 48W to 44 W with error of 25 to 50 watts in the SW). As per earlier general comment, instead of using qualitative "agrees well", perhaps quantify apparent error (with appropriate statistics) and explain why you consider this good agreement. |
| Our response: | We agree with the reviewer here that the high SINP simulation only show good agreement in the middle instead of the whole domain. Please see below for the corrections and modifications of the model-observation comparison for the October case. |
| Old text: | Several corrections and medications have been made to the paragraph from *Line 391* to *Line 400*. |
| New text: | *Line 444-Line 450:*
" The simulation with a high $S_{INP}$ in the October case agrees with all satellite retrievals from approximately 56 °W to 46 °W (Figure 14), but strongly underestimates the LWP for the region from 60 °W to 56 °W and overestimates the LWP for the eastern end of the region. The simulations with default and low $S_{INP}$ reproduce the LWP for the region from 60 °W to 56 °W, but overestimates the LWP for the rest of the region. Such overestimation of LWP in the cumulus-dominated region for the control simulation may come from the incorrect liquid-ice partitioning near the cloud top as shown in Figure C5 when model output are compared to MPhase C323 measurements. Similar biases are seen for SW flux which may be the results of LWP bias from the simulations." |

| | |
|---|---|
| | *Line 451-Line 453:*
"Based on the INP measurements from the M-Phase aircraft campaign, it is known that the measured INP concentrations in this October case are within the range of INP concentrations from default $S_{INP}$ and low $S_{INP}$, but both simulations show clear overestimation of LWP and SW flux here."

*Line 456-Line 457:*
"…the INP concentrations are temperature dependent but not directly derived from the background aerosols, and potentially missing variations of INPs through CAO cloud development." |
| **Reviewer comment (30)** | *Line 476 (and others).* Shouldn't a change in cloud phase driven by aerosols be called a "cloud-phase adjustment" rather than a feedback? |
| Our response: | Yes, a change in cloud phase driven by aerosols is a cloud adjustment, but *line 476* refers to the future change of INP concentration and its influence on the radiative properties of future clouds. If INP concentration increases in the warming climate (e.g., due to potentially higher dust emissions with less snow and ice cover), the ice-dominated clouds can become brighter and therefore reflect more SW radiation at the top-of-the-atmosphere, causing a negative cloud-phase feedback. This is a type of feedback as the warming climate induces the change of INP concentration, and then the change of cloud radiative properties, which eventually lead to the change of the warming. |
| Old text: | *Line 476-Line 477*:
"…and is potentially a new mechanism of negative cloud-phase feedback in ice-dominated CAO clouds if INP concentrations increase in the future." |
| New text: | *Line 535-Line 538:*
"…and is potentially a new mechanism of negative cloud-phase feedback in ice-dominated CAO clouds if INP concentrations increase in the future due to the warming climate, additional to the original three cloud-phase feedback mechanisms suggested by Murray et al. (2021)." |

**Minor changes made besides the response to reviewer comments**

| | |
|---|---|
| Description and reasons of change | We have added titles for all the appendices in this revised manuscript which were missing in the original manuscript. |
| Old text: | Untitled appendices |
| New text: | The added titles for appendices are:
- Appendix A: Retrieval time and selected model timepoint for satellite data used in this study
- Appendix B: Monthly distribution of cloud top temperature over the Labrador Sea in 2022
- Appendix C: Model-observation comparison for control simulations
- Appendix D: Supplementary figures for Figure 5
- Appendix E: Supplementary figures for Figures 6 and 7
- Appendix F: Boundary layer type fractions for all the simulations
- Appendix G: Results of model simulations with different precipitation, evaporation and sublimation setup for the March case
- Appendix H: Vertical profiles of in-cloud properties for simulations with different $N_d$ and $E_{HM}$
- Appendix I: Supplementary figures for model output compared to satellite retrievals |

| | |
|---|---|
| Description and reasons of change | We have corrected the description of the global model grid spacing. |
| Old text: | *Line 82:*
"…grid spacing near the equator…" |
| New text: | *Line 81:*
"…grid spacing near the midlatitudes…" |

| | |
|---|---|
| Description and reasons of change | We have added the sources of CALIPSO dataset and MPhase INP concentrations in the Data availability section. |
| Old text: | N/A |
| New text: | *Line 535-Line 536:*
"CALIPSO temperature and IWC can be found from https://www-calipso.larc.nasa.gov/products/. The INP concentrations can be found from https://doi.org/10.5281/zenodo.14781199." |

**Appendix 1:**

**Added content in Appendix C in the revised manuscript**

**Appendix C: Model-observation comparison for control simulations**

[revised manuscript text omitted]

---

## Author Response (AR2)

**Responses to reviewer comments on the manuscript "Different responses of cold-air outbreak clouds to aerosol and ice production depending on cloud temperature"**

We thank all the reviewers for their time and effort on reviewing our revised manuscript. Please see below for our responses to the reviewers' comments and minor changes made besides the response to reviewer comments.

**Response to comments by reviewer 1**

| Reviewer comment (1) | That said, I still have reservations about the 15 March case study that should be addressed. Specifically, I have reservations about the change in the simulation behaviour around 54W as seen in Figure 5 and beyond. |
|---|---|
| | In the revised manuscript it is stated: The model struggles with reproducing the cloud streets in March, and a better representation of the cloud streets requires much higher computational resources with finer grid spacing to conduct the sensitivity test, hence are not further investigated here. |
| | Yet the simulations are able to generate fine-scale cloud streets to the east of the 54W line, so I would think the resolution is adequate. Rather I suspect the issue is due to the coarse resolution fields generated by the global model ("N216 ~60 km resolution") coming in through the western boundary at 60W. What is the wind speed along the western boundary? How far would you expect the boundary effects to propagate into the nested domain? Will it reach the sub-domain? Will it reach 54W? Please comment on this directly in the manuscript. |
| Our response: | We thank the reviewer for raising this question. |
| | We realised that the nested model domains for both cases have not been shown in the manuscript, which may lead to misunderstanding of the location of the sub-domains within the nested model domains. The bigger domains shown in Figure 3 and Figure 4 in the manuscript are not the nested model domains, but domains designed based on the availability and coverage of the satellite retrievals and they are smaller than the actual nested model domains. |
| | We have now added the maps showing the nested model domains and the subdomains for analysis in a new Appendix (Appendix C in the revised manuscript). The ordering of the appendices after Appendix C has also been updated in the revised manuscript. |

| | We believe that all the sub-domain for analysis in the March case is sufficiently far from the western boundary of the nested model domain. The boundary effects do not reach the sub-domain or regions around 54W. Please see the new Appendix C for a more detailed discussion and our answers to the reviewer's questions.

We have also added the text to explain why the cloud streets are not resolved in the early stratus region in our model in the manuscript. Please see the "New text" below for the added content in the main content and Appendix 1 in the end of this document for the added content in the new Appendix C. |
|---|---|
| Old text: | *Line 208-Line 210:*
"The model struggles with reproducing the cloud streets in March, and a better representation of the cloud streets requires much higher computational resources with finer grid spacing to conduct the sensitivity test, hence are not further investigated here." |
| New text: | *Line 83-Line 84:*
"The nested model domains are shown in Appendix C."

*Line 208-Line 214:*
"In the March case, the model struggles with reproducing the fine structures of cloud streets to the east of 54 ºW in the sub-domain. This is because the effective horizontal resolution (between 5 times to 10 times of the grid spacing) in our model cannot fully resolve the cloud streets at the beginning of the CAO event. With the clouds moving into the convective region and the boundary layer getting depended (to the west of 54 ºW), the scales of cloud street grow and can then be resolved in the model. A better representation of the cloud streets at the beginning of the CAO event requires higher model resolution (Field et al., 2017a) and therefore much higher computational resources to conduct the sensitivity test, hence are not further investigated here."

*Line 227–Line 230:*
"The locations of the sub-domains in the nested model domains are shown in Appendix C. The whole sub-domain in the March case and most of the sub-domain (except the small northwestern part) of the October case are sufficiently distant from the boundaries of the nested model domain, hence not affected by the boundary effects from fields entering from the global model. A detailed discussion on the boundary effects is shown in Appendix C."

*Added content in Appendix C (shown in Appendix 1 in the end of this document)* |

| | |
|---|---|
| **Reviewer comment (2)** | One small query for Figure 5, is the figure for a specific time or for the duration of the simulation? I think it's the later, but it would be best to state this explicitly. |
| Our response: | The times for plots in Figure 5 and figures in Appendix E (originally Appendix D in the previous manuscript) are for a specific time of each case. This choice is to make sure the consistency with the other plots (corresponding to the CERES measurement times).

We have now added the specific time for each case in the corresponding plots. |
| Old text: | N/A |
| New text: | *The content below are added to the captions of Figures 5, E1 and E1:*
"The time points selected are 16:45 UTC for the March case and 17:00 UTC for the October case, which are consistent with the corresponding CERES measurement times." |
| **Reviewer comment (3)** | Figure 1 legend:  2022 instead of 2024 |
| Our response: | We are sorry that the reviewer's two typo corrections from the first peer-review report were missed in our previous responses. Both changes have been made. |
| Old text: | "Figure 1. The UK Met Office surface analysis charts at 1200 UTC on (a) 15 March 2024 and (b) 24 October 2024." |
| New text: | "Figure 1. The UK Met Office surface analysis charts at 1200 UTC on (a) 15 March 2022 and (b) 24 October 2022." |
| **Reviewer comment (4)** | Figure 8 legend:  24 October |
| Our response: | Corrected. Please note that this was corrected in the previous manuscript and therefore is not shown in the tracked change for the revised manuscript here from this round of review. |
| Old text: | "… on 15 October, …" |
| New text: | "… on 15 March 2022, …" |

**Response to comments by reviewer 3**

| | |
|---|---|
| **Reviewer comment** (1) | While perhaps not strictly necessary, in my view it is good practice to give the product names and versions numbers of the datasets used, and I encourage the authors to add such to the section on data availability. |
| Our response: | We have now added missing product names and version numbers of the datasets used in the data availability section, as well as adding the missing version numbers in the Method section and Appendix D (the original Appendix C for the previous manuscript). |
| Old text: | *Line 544-Line 549:* "MODIS data including the RGB imagery and in-cloud cloud water path can be found from https://modis.gsfc.nasa.gov/. All-sky liquid water path from AMSR-2 can be found from https://www.remss.com/missions/amsr/. Shortwave and longwave radiation flux at the top-of-the-atmosphere can be found from https://ceres.larc.nasa.gov/data/. CALIPSO temperature and IWC can be found from https://www-calipso.larc.nasa.gov/products/. The INP concentrations can be found from https://doi.org/10.5281/zenodo.14781199. Observations during the M-Phase aircraft campaign can be found from https://catalogue.ceda.ac.uk/uuid/2040b17716fd49f2ac8b0b35c 773d609/ on the CEDA Archive. Model data used in this work is available at https://doi.org/10.5281/zenodo.14536461." |
| New text: | *data availability (Line 552-Line 566):* "The satellite data products used in this study include: MODIS Level 1B Calibrated Radiances Product (Collection 6.1) onboard the Aqua satellite for RSB composites with bands 1,3 and 4 (https://ladsweb.modaps.eosdis.nasa.gov/missions-and-measurements/products/MYD021KM) with geolocation data from the Geolocation 1km (https://ladsweb.modaps.eosdis.nasa.gov/missions-and-measurements/products/MYD03); MODIS Atmosphere Level 2 Cloud Production (Collection 6.1) onboard the Aqua satellite for cloud water path, cloud cover and cloud-top temperature (https://ladsweb.modaps.eosdis.nasa.gov/missions-and-measurements/products/MYD06_L2); CERES SSF Level 2 product (Edition 4A) onboard the Aqua satellite for TOA shortwave flux and longwave flux (https://ceres.larc.nasa.gov/data/); columnar cloud liquid water (version 8.2) for all-sky liquid water path from AMSR-2 onboard GCOM-W (https://www.remss.com/missions/amsr/); Temperature from Level-2 1km Cloud Layer Data (version 4-51, https://asdc.larc.nasa.gov/project/CALIPSO/CAL_LID_L2_01km CLay-Standard-V4-51_V4-51) and ice water content from Level-2 5km Cloud Profile Data (version4-51, https://asdc.larc.nasa.gov/project/CALIPSO/CAL_LID_L2_05km CPro-Standard-V4-51_V4-51) from CALIPSO. The ERA5 data used include the surface skin and surface pressure from ERA5 |

<table>
<tr>
<td></td>
<td>hourly data on single levels (https://cds.climate.copernicus.eu/datasets/reanalysis-era5-single-levels), and temperature and pressure at 800 hPa from ERA5 hourly data on pressure levels (https://cds.climate.copernicus.eu/datasets/reanalysis-era5-pressure-levels). The INP data from the MPhase aircraft campaign were obtained from: https://zenodo.org/records/14781199. The measurements during the MPhase aircraft campaign can be obtained from https://catalogue.ceda.ac.uk/uuid/6d7971a92d154bb29af3167df b6f5a7e/. The model data used for analysis can be found from https://zenodo.org/records/14536461. "

*Method (Line 175-Line 182):*
Several changes made to add version numbers for the satellite products.

*Appendix D (Line 615-Line 616):*
Several changes made to add version numbers for the satellite products.</td>
</tr>
</table>

**Minor changes made besides the response to reviewer comments**

| **Description of change** | Added the missing "financial support" section. |
|---|---|
| Old text: | N/A |
| New text: | *Financial support:*
"The M-Phase aircraft campaign was supported by the Natural Environment Research Council (NERC) as part of the CloudSense programme (M-Phase: NE/T00648X/1 and NE/T006463/1). XH was supported by the SENSE - Centre for Satellite Data in Environmental Science CDT (Centre for Doctoral Training) (NE/T00939X/1) with a CASE studentship from the UK Met Office." |

**Appendix 1:**

**Added content in Appendix C in the revised manuscript**

**Appendix C: Maps for the nested model domains and the sub-domains for analysis**

Figure C1 shows the nested model domains and the sub-domains for analysis of both cases. The sub-domains were chosen to be away from the boundaries of the nested model domains to avoid the boundary effects. In both cases, due to the winds and airmasses from northwest direction, the fields generated from the coarser global model require some time to spin up once entering the nested model domain.

We examined whether the boundary effects can reach the sub-domains by using simple calculations of how long it takes the airmasses to reach the western boundary of each sub-domain, and how long it takes for the boundary layer overturning once the fields entering the nested model domain.

The distances were estimated based on the mean direction of wind. For the March case, the distance between the middle of the western boundary of the nested model domain to the middle of the western boundary of the sub-domain is around 430 km. The mean wind speed from surface to 2 km height above sea level is 13.0 m s$^{-1}$ (46.8 km hr$^{-1}$).  Therefore, it takes around 9 hours for the airmasses to reach the sub-domain in the March case. For the October case, the distance between the northwest point of the nested model domain to the middle of the western boundary of the sub-domain is around 500 km. The mean wind speed from surface to 2 km height above sea level is 16.5 m s-1 (59.4 km hr$^{-1}$), so it takes around 8.5 hours for the airmasses to reach the sub-domain in the October case.

The timescale for boundary layer overturning in the nested model domain is calculated as $L/\sigma_w$, where L is the boundary layer depth (~1000 m at the beginning of the CAO events) and $\sigma_w$ is the standard deviation of the w component of wind. For the March case, the mean $\sigma_w$ below 1000 m between the western boundaries of the nested model domain and the sub-domain is around 0.67 ms$^{-1}$, which results in a 0.4 h overturning timescale. For the October case, the mean $\sigma_w$ below 1000m between the northwestern end of the nested model domain and the western boundary of the sub-domain is around 0.59 ms$^{-1}$, resulting in a 0.5 h overturning timescale.

Therefore, the boundary layers for both cases should be well evolved (>10x overturns) by the time the air reaches the sub-domain boundaries used for analysis. Some airmass travelling into northwestern part of the sub-domain may have less time to spin up and be affected by the boundary effects, however we choose to keep this part of the sub-domain for capturing more earlier stage CAO clouds as the clouds broke up into cumulus clouds very soon in the October case.

For the whole sub-domain in the March case and most of the sub-domain (except the small northwestern part) of the October case, the time and distance required for the airmasses to reach the sub-domains are sufficient to avoid the boundary effects propagate into the sub-domain.

[Figure]

Figure C1. Nested model domains (black) and the selected sub-domains for analysis (blue) for (a) 15 March 2022, (b) 24 October 2022.